# Muscarinic acetylcholine receptor signaling generates OFF selectivity in a simple visual circuit

Bo Qin[1], Tim-Henning Humberg [2,4], Anna Kim[1,4], Hyong S. Kim[1], Jacob Short[1], Fengqiu Diao[3], Benjamin H. White[3], Simon G. Sprecher [2] & Quan Yuan [1]

ON and OFF selectivity in visual processing is encoded by parallel pathways that respond to either light increments or decrements. Despite lacking the anatomical features to support split channels, *Drosophila* larvae effectively perform visually-guided behaviors. To understand principles guiding visual computation in this simple circuit, we focus on investigating the physiological properties and behavioral relevance of larval visual interneurons. We find that the ON vs. OFF discrimination in the larval visual circuit emerges through light-elicited cholinergic signaling that depolarizes a cholinergic interneuron (cha-lOLP) and hyperpolarizes a glutamatergic interneuron (glu-lOLP). Genetic studies further indicate that muscarinic acetylcholine receptor (mAchR)/Gαo signaling produces the sign-inversion required for OFF detection in glu-lOLP, the disruption of which strongly impacts both physiological responses of downstream projection neurons and dark-induced pausing behavior. Together, our studies identify the molecular and circuit mechanisms underlying ON vs. OFF discrimination in the *Drosophila* larval visual system.

[1] National Institute of Neurological Disorders and Stroke, National Institutes of Health, Bethesda, MD 20892, USA. [2] Department of Biology, University of Fribourg, 1700 Fribourg, Switzerland. [3] National Institute of Mental Health, National Institutes of Health, Bethesda, MD 20892, USA. [4] These authors contributed equally: Tim-Henning Humberg, Anna Kim. Correspondence and requests for materials should be addressed to Q.Y. (email: quan.yuan@nih.gov)

O N and OFF selectivity, the differential neuronal responses elicited by signal increments or decrements, is an essential component of visual computation and a fundamental property of visual systems across species[1–3]. Extensive studies of adult *Drosophila* optic ganglia and vertebrate retinae suggest that the construction principles of ON and OFF selective pathways are shared among visual systems, albeit with circuit-specific implementations[4–6]. Anatomically, dedicated neuronal pathways for ON vs. OFF responses are key features in visual circuit construction. Specific synaptic contacts are precisely built and maintained in laminar and columnar structures during development to ensure proper segregation of signals for parallel processing[4,7]. Molecularly, light stimuli elicit opposite responses in ON and OFF pathways through signaling events mediated by differentially expressed neurotransmitter receptors in target neurons postsynaptic to the photoreceptor cells (PRs). This has been clearly demonstrated in the mammalian retina, where light-induced changes in glutamatergic transmission activate ON-bipolar cells via metabotropic metabotropic glutamate receptor 6 (mGluR6) signaling and inhibit OFF-bipolar cells through the actions of ionotropic AMPA or kainate receptors[8,9]. In the adult *Drosophila* visual system, functional imaging indicates that ON vs. OFF selectivity emerges from visual interneurons in the medulla[10–13]. However, despite recent efforts in transcriptome profiling and genetic analyses[14,15], the molecular machinery mediating signal transformation within the ON and OFF pathways has not yet been clearly identified.

Unlike the ~6000 PRs in the adult visual system, larval *Drosophila* eyes consist of only 12 PRs on each side[4,16]. Larval PRs make synaptic connections with a pair of visual local interneurons (VLNs) and approximately ten visual projection neurons (VPNs) in the larval optic neuropil (LON) (Fig. 1a). VPNs relay signals to higher brain regions that process multiple sensory modalities[17]. Despite this simple anatomy, larvae rely on vision for negative phototaxis, social clustering, and form associative memories based on visual cues[18–23]. How the larval visual circuit effectively processes information and supports visually guided behaviors is not understood.

Recent connectome studies mapped synaptic interactions within the LON in the first instar larval brain[17], revealing two separate visual pathways using either blue-tuned Rhodopsin 5 (Rh5-PRs) or green-tuned Rhodopsin 6 (Rh6-PRs). Rh5-PRs project to the proximal layer of the LON (LONp) and form direct synaptic connections with all VPNs, whereas Rh6-PRs project to the distal layer of the LON (LONd) and predominantly target one cholinergic (cha-lOLP) and one glutamatergic (glu-lOLP) local interneurons. The two PR pathways then converge at the level of VPNs (Fig. 1a).

Theses connectome studies also revealed potential functions for cha- and glu-lOLP. The pair of lOLPs, together with one of the VPNs, the pOLP, are the earliest differentiated neurons in the larval optic lobe and are thus collectively known as optic lobe pioneer neurons (OLPs)[24–26]. Besides relaying visual information from Rh6-PRs to downstream VPNs, the lOLPs also form synaptic connections with each other and receive neuromodulatory inputs from serotonergic and octopaminergic neurons, suggesting that they may act as ON and OFF detectors[17] (Fig. 1a). This proposal is further supported by recent studies on the role of the Rh6-PR/lOLP pathway in larval movement detection and social clustering behaviors[27]. However, it remains unclear how the lOLPs support differential coding for ON and OFF signals without anatomical separation at either the input or output level.

In this study, we investigated the lOLPs' physiological properties and determined the molecular machinery underlying their information processing abilities. Our functional imaging studies revealed differential physiological responses towards light increments and decrements in cha-lOLP and glu-lOLP, indicating their functions in detecting ON and OFF signals. Furthermore, we found that light-induced inhibition on glu-lOLP is mediated by mAchR-B/Gαo signaling, which generates the sign inversion required for the OFF response and encodes temporal information between the cholinergic and glutamatergic transmissions received by downstream VPNs. Lastly, genetic manipulations of glu-lOLP strongly modified the physiological responses of VPNs and eliminated dark-induced pausing behaviors. Together, our studies identify specific cellular and molecular pathways that mediate OFF detection in *Drosophila* larvae, reveal functional interactions among key components of the larval visual system, and establish a circuit mechanism for ON vs. OFF discrimination in this simple circuit.

## Results

**Identification of enhancer Gal4 lines for the OLPs**. To perform physiological and genetic studies on the lOLPs, we first screened the enhancer Gal4 collection produced by the Janelia Farm Fly-Light Project for driver lines specifically labeling OLPs based on their anatomical features[28,29].

We selected three Gal4 enhancer lines: R72E03, R84E12, and R72A10, and determined which OLPs were labeled by each line using anti-ChAT and anti-VGluT antibody staining (Fig. 1a–c, Supplementary Figs. 1, 2)[24–26]. R72E03-Gal4 (lOLP$^{glu}$-Gal4) labels glu-lOLP only, R84E12-Gal4 (lOLP-Gal4) labels cha- and glu-lOLP, and R72A10-Gal4 (OLP-Gal4) labels both lOLPs and the pOLP. We also tested a R72A10-LexA line, which showed the same expression pattern as the Gal4 version (Fig. 1b, Supplementary Fig. 1)[17]. Single-cell labeling using the FLP-out technique and the R84E12-Gal4 enhancer indicate that cha-lOLP and glu-lOLP have similar projection patterns and that their termini are largely contained within the LON region (Supplementary Fig. 2).

**Light elicits differential calcium responses in the OLPs**. Next, we examined the OLPs' physiological properties using optical recordings. Since OLPs are direct synaptic targets of PRs, we expected to observe light-evoked calcium responses in these neurons using a larval eye–brain explant protocol established in our previous studies[30]. This approach allows us to deliver temporally controlled light simulations using either the 488 or 561 nm laser while detecting calcium transients via cell-specific expression of GCaMP6f through two-photon imaging at both the soma and terminal regions (Fig. 1d, Supplementary Fig. 3a)[31].

Calcium imaging using lOLP$^{glu}$-Gal4 and lOLP-Gal4 revealed distinct light-elicited physiological responses in the two lOLPs. Upon light stimulation, a 100 ms light pulse delivered by the 561 nm laser, glu-lOLP exhibited a small reduction in GCaMP signal followed by a delayed calcium transient, whereas cha-lOLP responded to light with a large and fast calcium rise (Fig. 1d, e). Calcium transients obtained from the terminal region of glu-lOLP displayed similarly biphasic waveforms as those in the somas, although with higher amplitudes and shorter latencies (Fig. 1d). In addition, termini recordings of both lOLPs produced two distinct peaks that clearly reflect the temporal difference in the light-induced calcium responses of the two lOLPs (Fig. 1e). Using R72A10-LexA enhancer-driven LexAop-GCaMP6f expression, we also obtained comparable results for the two lOLPs and characterized the profile of the light response in pOLP, which displayed the same initial reduction followed by a delayed calcium rise as the glu-lOLP response (Fig. 2a, c, d, Supplementary Fig. 3a, 4). Lastly, we validated the consistency of our glu-lOLP data sets by quantifying the three different enhancer lines and obtaining similar results (Supplementary Fig. 4).

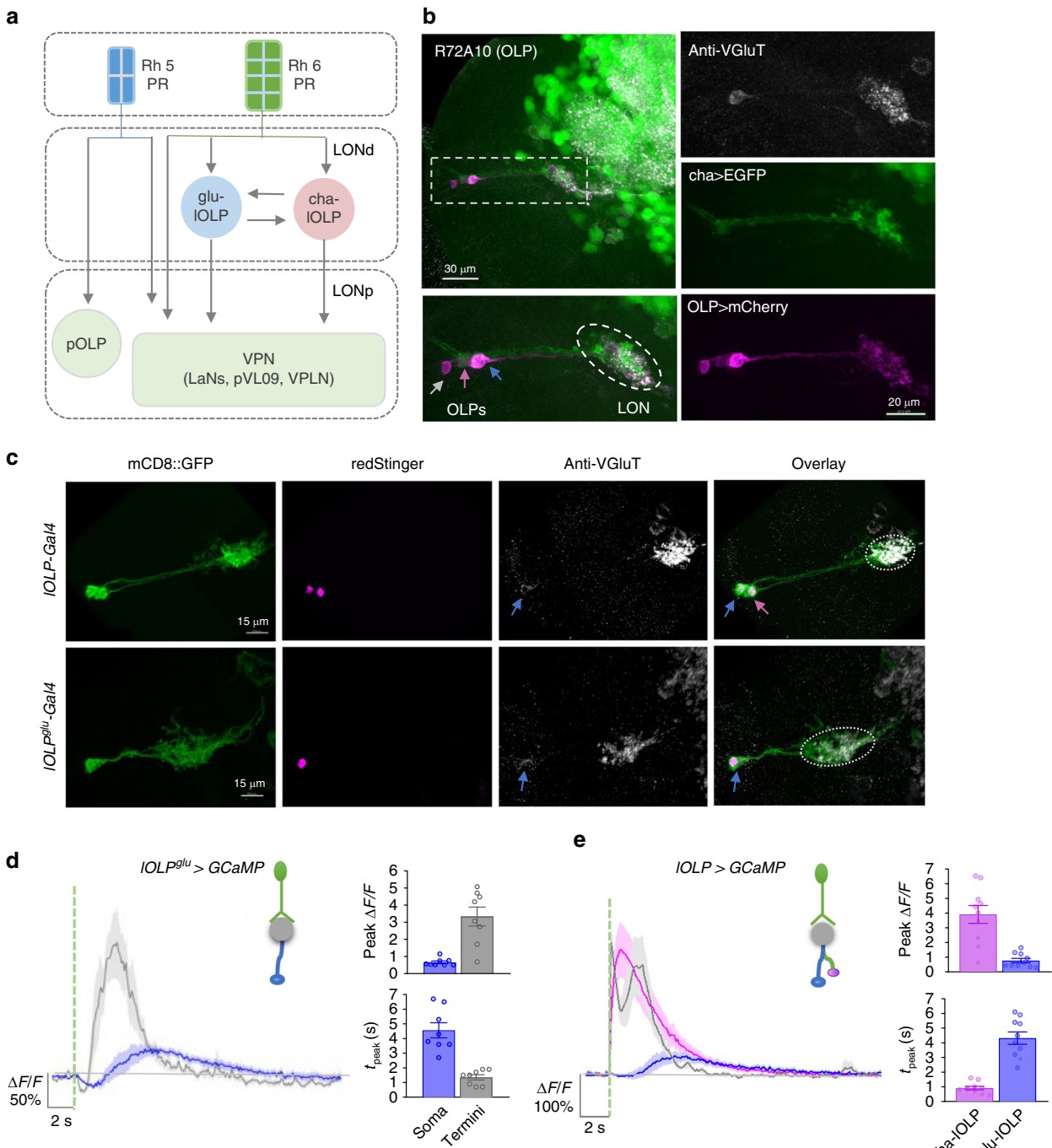

**Fig. 1** Distinct light-elicited calcium responses in larval visual interneurons. **a** Circuit diagram of the *Drosophila* larval visual system. Rh5-expressing photoreceptor neurons (Rh5-PRs) project to the proximal layer of the LON (LONp) and transmit visual signals into the brain via direct synaptic connections with visual projection neurons (VPNs). Rh6-PRs project to the distal layer of the LON (LONd) and predominantly synapse onto two local interneurons, one cholinergic (cha-lOLP) and one glutamatergic (glu-lOLP), which then connect to the VPNs. Gray arrows indicate the unknown effects of light input on OLPs and most VPNs, as well as the undetermined interactions between the lOLPs. **b** Enhancer screens identified enhancer elements that label three OLPs. R72A10-LexA-driven LexAop-mCherry expression (magenta) reveals three somas near the lateral edge of the brain lobe, including the VGluT-positive glu-lOLP (blue arrow), the ChAT-positive cha-lOLP (pink arrow), and the projection OLP (pOLP, gray arrow). The LON region is marked by a dashed oval. **c** Enhancer Gal4 lines specifically labeling two local OLPs (lOLP-Gal4) and the single glu-lOLP (lOLP$^{glu}$-Gal4) were identified. Representative confocal images of larval brains expressing mCD8::GFP and RedStinger driven by enhancer Gal4 lines are shown. Glu-lOLP is positive for anti-VGluT staining in the soma (blue arrows) and terminal processes (dashed circles) that project to the LON. Scale bars = 15 μm. **d**, **e** Calcium imaging experiments reveal differential physiological responses to light in two lOLPs. **d** Delayed calcium transients in glu-lOLP are observed using lOLP$^{glu}$-Gal4 driving GCaMP6f. The calcium transients obtained at the terminal region (termini) show reduced latency and increased amplitude compared to the ones from the soma. n = 8. **e** Light pulses induce fast calcium transients in cha-lOLP (magenta) and slow transients in glu-lOLP (blue). The calcium transient generated at the terminal region is in gray. The average traces of GCaMP6f driven by lOLP-Gal4 and the quantifications of peak value and peak time of changed intensity (ΔF/F) are shown. n = 10. The dashed green line represents a 100 ms light pulse at 561 nm. Shaded areas on traces and error bars on quantifications represent SEM

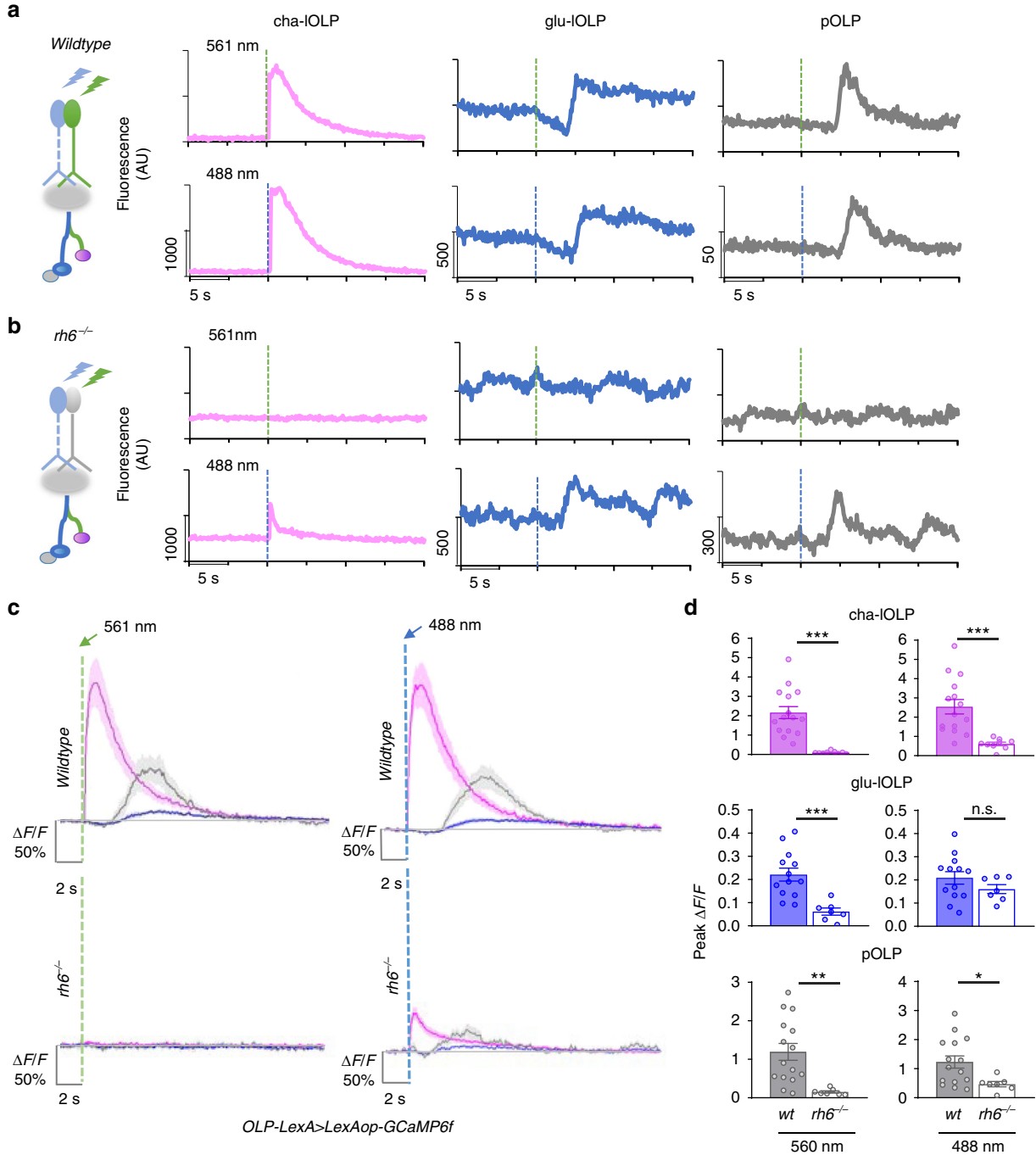

**Fig. 2** OLPs receive presynaptic inputs predominantly from Rh6-PRs. **a**, **b** The contribution of Rh5- and Rh6-PRs to light-evoked calcium responses in OLPs as revealed by stimulation at different wavelengths in wild-type and Rh6 mutants. Left: schematic diagram illustrating the stimulation scheme used in calcium imaging experiments. Green or blue light pulses (dashed lines, green: 561 nm, blue: 488 nm) activate Rh5- or Rh6-PRs and elicit OLP-LexA-driven GCaMP6f signals in the somas of OLPs. Right: Representative raw traces of OLP > GCaMP6f collected from wild-type and Rh6 mutants ($rh6^{-/-}$). Magenta: cha-lOLP; blue: glu-lOLP; gray: pOLP. **c**, **d** OLPs are functionally connected to Rh6-PRs in the third instar larval brain. Light pulses (dashed lines, green: 561 nm, blue: 488 nm) induced fast calcium transients in cha-lOLP (magenta) and slow transients in glu-lOLP (blue) and pOLP (gray). Compared to wild-type controls, OLPs in Rh6 mutants showed no response towards green light (561 nm) stimulation and dampened responses toward blue light (488 nm) stimulation except for glu-lOLP, which remained equally responsive. The **c** average traces and **d** quantification of peak value of changed intensity ($\Delta F/F$) are shown. Shaded areas on traces and error bars on quantifications represent SEM. Wild-type control: cha-lOLP, $n = 15$; glu-lOLP, $n = 13$; pOLP, $n = 15$. Rh6 mutant ($rh6^{-/-}$): cha-lOLP, $n = 9$; glu-lOLP, $n = 7$; pOLP, $n = 7$. cha-OLP, 561 nm: $p < 0.0001$, $t = 5.102$, df = 22; cha-OLP, 488 nm: $p = 0.0007$, $t = 3.929$, df = 22; glu-OLP, 561 nm: $p = 0.0009$, $t = 3.977$, df = 18; glu-OLP, 488 nm: $p = 0.2362$, $t = 1.225$, df = 18; pOLP, 561 nm: $p = 0.0044$, $t = 3.207$, df = 20; pOLP, 488 nm: $p = 0.0261$, $t = 2.402$, df = 20. Statistical significance was determined by Student's $t$ test. $p \geq 0.05$ was considered not significant (n.s.), ***$p < 0.001$, **$p < 0.01$, and *$p < 0.05$

Calcium imaging studies of the OLPs reveal distinct light-evoked response profiles. Notably, calcium transients obtained from cha- and glu-lOLP resemble the ones observed in adult fly visual interneurons that belong to either the ON or OFF pathways, respectively, suggesting potential functional similarities between lOLPs and the interneurons in the adult visual ganglia[10,32].

**OLPs receive presynaptic inputs predominantly from Rh6-PRs.** Connectome studies indicate that, in the first instar larval brain, the majority of lOLPs' PR inputs come from Rh6-PRs, while the pOLP receives inputs directly from Rh5-PRs[17] (Fig. 1a). To establish functional connectivity between subtypes of PRs and OLPs, we performed calcium imaging with light stimulations at either 488 or 561 nm (Supplementary Fig. 3a). Previous studies indicated that Rh6 detects light within the 400–600 nm range, rendering them sensitive to light stimulations at both 488 and 561 nm, whereas Rh5 detects light from 350 to 500 nm and responds to blue light at 488 nm[33]. These features, in combination with a loss-of-function Rh6 mutant ($rh6^{-/-}$)[34], allowed us to examine the specific contributions of Rh5- and Rh6-PRs to the OLPs' light responses.

In wild-type larvae, 488 and 561 nm light stimulations elicit almost identical responses from the OLPs (Fig. 2a, c, d), while responses to 561 nm light were eliminated in Rh6 mutants, demonstrating that green light-evoked responses in OLPs are solely generated by visual transduction in Rh6-PRs (Fig. 2b–d). To test the functional connectivity between Rh5-PRs and OLPs, we performed experiments using 488 nm light stimulations in Rh6 mutants, where blue light-elicited responses are exclusively generated by Rh5-PRs. Interestingly, compared to wild-type controls, blue light-induced calcium responses in cha-lOLP and pOLP were significantly reduced in Rh6 mutants, whereas there was no significant difference in glu-lOLP's response (Fig. 2b–d). These findings demonstrate that cha-lOLP and pOLP receive most of their light inputs from Rh6-PRs. In contrast, glu-lOLP has strong functional connections to both Rh5- and Rh6-PRs.

The functional connectivity revealed by calcium imaging at the third instar larval stage largely agrees with the wiring diagram produced in the first instar larval brain[17], suggesting that Rh6-PR/lOLP connectivity is preserved during larval development and can be detected through functional analyses. However, we also found connections that were not indicated in the connectome study. Specifically, that glu-lOLP receives inputs from both Rh5- and Rh6-PRs and that pOLP is mainly driven by Rh6-PR input. These differences may be attributed either to developmental changes in circuit connectivity or physiological interactions that are not directly reflected by anatomical connections, highlighting the importance of complementing connectome analyses with physiological studies.

**Light hyperpolarizes glu-lOLP and depolarizes cha-OLP.** To measure light-induced calcium and voltage responses in the lOLPs, we examined changes in membrane potential using the genetically encoded voltage sensor Arclight while recording calcium transients with the red calcium indicator RCaMP[35,36]. By matching calcium profiles with voltage changes, we found that light pulses induce depolarization and fast calcium transients in cha-lOLP, but hyperpolarization and biphasic calcium transients in glu-lOLP (Fig. 3a, b). RCaMP recordings obtained calcium transients with similar waveforms, but reduced amplitudes compared to GCaMP recordings (Fig. 3b, Supplementary Fig. 8).

We next tested how the lOLPs respond to light increments and decrements by monitoring calcium responses during onsets and offsets of extended light exposures. Although two-photon recordings of GCaMP6f provided the best image quality, extended light exposures are incompatible with the sensitive light detector. Therefore, in the following experiments, we used RCaMP as the calcium indicator, which can be imaged using a low-intensity confocal laser tuned to 561 nm, reducing the photobleaching effects on both the calcium sensor and the photoreceptors. Additionally, this protocol allowed for the alteration of light cycles and delivering dark pulses by tuning the 488 nm laser during imaging sessions (Supplementary Fig. 5a).

RCaMP recordings showed that cha-lOLP only responded to the light onset of an extended light exposure with a fast calcium transient, demonstrating its specific response to light increments. In contrast, glu-lOLP responded to the light offset with an immediate calcium rise, suggesting that glu-lOLP is activated by the light decrements (Fig. 3c).

We performed additional experiments to examine the differential responses of glu-lOLP toward light increments and decrements by subjecting the preparation to contrast-matched 100 ms light or dark pulses (~11.7 $\mu$W/cm$^2$). A light pulse induces a biphasic calcium transient as indicated by a small and noticeable reduction followed by a delayed calcium rise, whereas a dark pulse, or a brief reduction in light intensity following an extended light exposure, generates an immediate calcium rise in glu-OLPs. Compared to the delayed calcium rise induced by light pulses, this dark-induced OFF response has a similar amplitude, but significantly shorter latency (Fig. 3d). Similar recordings indicate that cha-lOLP does not respond to dark pulses and only generates the fast ON response to light pulses (Fig. 4d, e).

Our recordings using voltage and calcium indicators demonstrate that the ON and OFF selectivity in the larval visual system emerges at the level of the lOLPs. We show that cha-lOLP specifically responds to light increments and is ON selective, while glu-lOLP responds to light decrements and displays OFF selectivity.

**mAchR-B mediates light-induced inhibition of glu-OLP.** Our study demonstrates that light stimulations depolarize cha-lOLP and hyperpolarize glu-lOLP. These physiological responses are likely mediated by differentially expressed acetylcholine receptors (AchRs) in the lOLPs that respond to acetylcholine release from the PRs[37,38]. Sign inversion, which transforms the light response in the PRs into an OFF response in glu-lOLP, is particularly critical for generating ON and OFF selectivity. Therefore, we sought to identify the receptor that produces this sign inversion and mediates the light-induced inhibition in glu-lOLP.

While ionotropic nicotinic AchRs (nAchRs) are generally associated with neuronal activation, subtypes of muscarinic AchRs (mAchRs) can be either excitatory or inhibitory depending on the G protein coupled with the receptors. Studies in mammalian mAchRs indicate that the excitatory M1/3 types are coupled to G$\alpha$q/11, whereas the inhibitory M2/4 types are coupled to G$\alpha$i/o[37]. The *Drosophila* genome contains three mAchRs, with types A and C coupling to G$\alpha$q/11 and type B coupling to G$\alpha$i/o[39,40]. Additionally, R72E03-Gal4, the enhancer Gal4 line labeling glu-lOLP, was generated using an upstream enhancer element identified in the *Drosophila* mAchR-B gene[28,29], suggesting its expression in glu-lOLP.

With mAchR-B as a likely candidate for mediating light-induced inhibition in glu-lOLP, we examined its expression pattern using a gene-trap line with a Gal4-DBD element inserted into the 5′-untranslated region region of the mAchR-B gene by the MiMIC transposon-mediated cassette exchange technique[41,42] (Fig. 4a). Enhancer-driven mAchR-B EGFP expression revealed its extensive distribution in the third instar

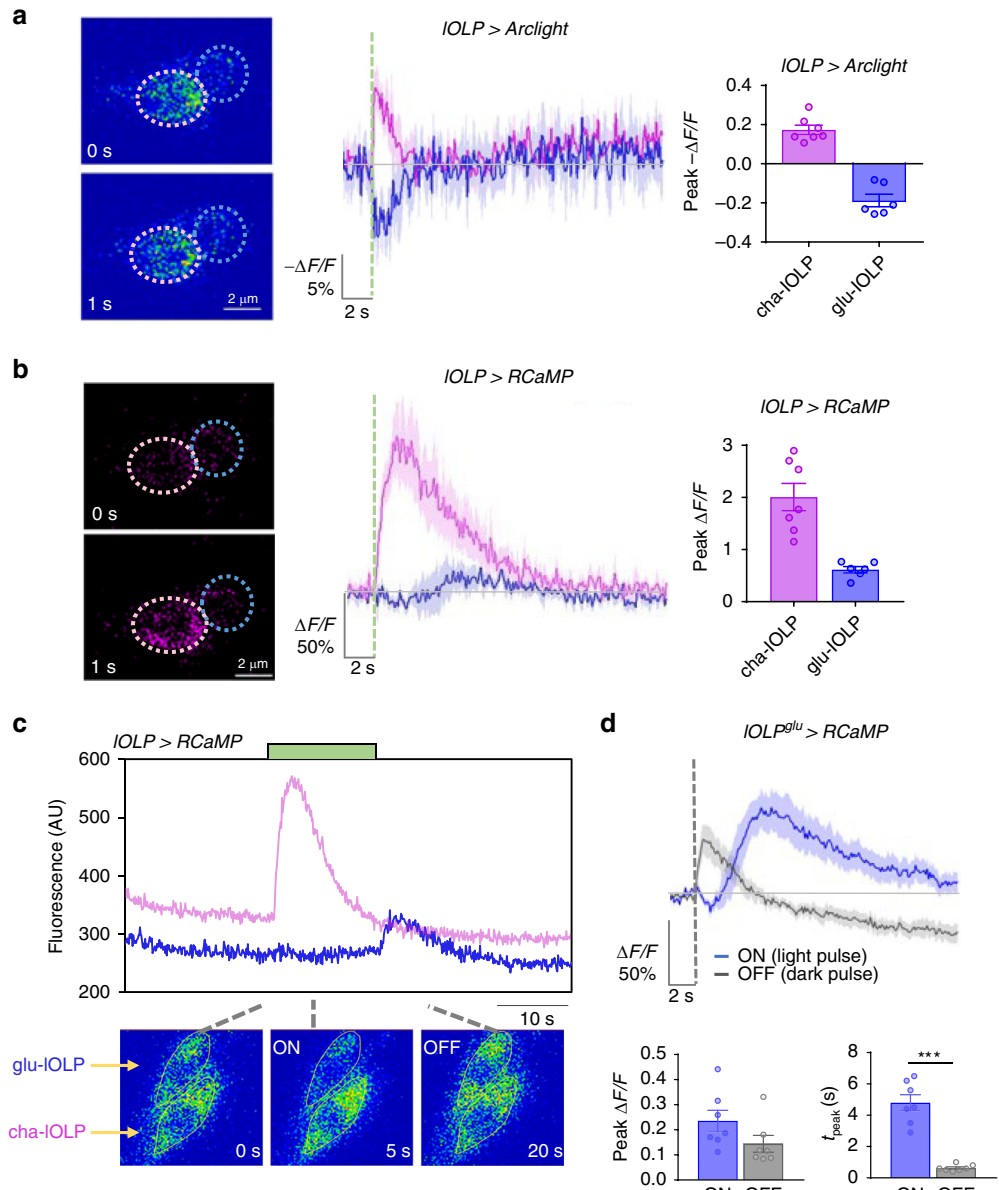

**Fig. 3** Light activates cha-lOLP and inhibits glu-lOLP. **a**, **b** Optical recordings using the voltage sensor Arclight together with the calcium sensor RCaMP reveal light-induced depolarization and fast calcium transients in cha-lOLP (magenta) as well as hyperpolarization and delayed calcium transients in glu-lOLP (blue). Representative frames from the recordings (left), averaged traces (middle), and the quantification of peak values of the changed intensity ($\Delta F/F$) (right) are shown. Scale bars and time are as indicated. Somatic regions used for quantification are marked by dashed circles. The dashed green line represents a 100 ms light pulse. cha-lOLP, $n = 7$; glu-lOLP, $n = 6$. **c** cha-lOLP exhibits ON responses, while glu-lOLP exhibits OFF responses. A representative raw trace from the lOLP > RCaMP recording is shown (top). The sample was subjected to an extended (12.5 s) light stimulation (green bar). cha-lOLP responded to the light onset, but not to the light offset. In contrast, the light onset induced a small reduction of calcium signal in glu-lOLP, while the light offset produced a rapid calcium rise. Representative frames of the recording are shown (bottom). **d** ON and OFF signals generate calcium transients with different temporal profiles in glu-lOLP. Average traces of calcium transients generated by recordings of lOLP$^{glu}$-Gal4 driving RCaMP are shown, demonstrating the slow calcium response to the light pulse (ON response, blue) and the fast calcium response to the dark pulse (OFF response, gray). The response amplitudes were not significantly different. The average traces (top) and the quantification of peak value and peak time of changed intensity ($\Delta F/F$) (bottom) are as shown. $n = 7$ in both groups. ON: $p = 0.1205$; OFF: $p < 0.001$. Shaded areas on traces and error bars on quantifications represent SEM. The dashed line represents a 100 ms light or dark pulse. Statistical significance was determined by Student's $t$ test. $p \geq 0.05$ was considered not significant, ***$p < 0.001$

larval brain. Immunohistochemical studies using anti-ChAT and anti-VGluT antibodies confirmed that the mAchR-B receptor expresses in glu-lOLP, but not in cha-lOLP (Fig. 4b, c).

Next, we performed transgenic RNA interference (RNAi) knockdown experiments targeting mAchR-B and recorded the lOLPs' response to 100 ms light vs. dark pulses using RCaMP to examine mAchR-B's function in mediating glu-lOLP's

physiological responses. Consistent with our earlier observations in wild-type controls, cha-lOLP only responded to the light pulse and generated a fast calcium transient, whereas glu-lOLP responded to both light and dark pulses with delayed and rapid calcium rises, respectively. Strikingly, mAchR-B knockdown eliminated both light and dark pulse-induced calcium transients in glu-lOLP, indicating mAchR-B as the mediator for light-

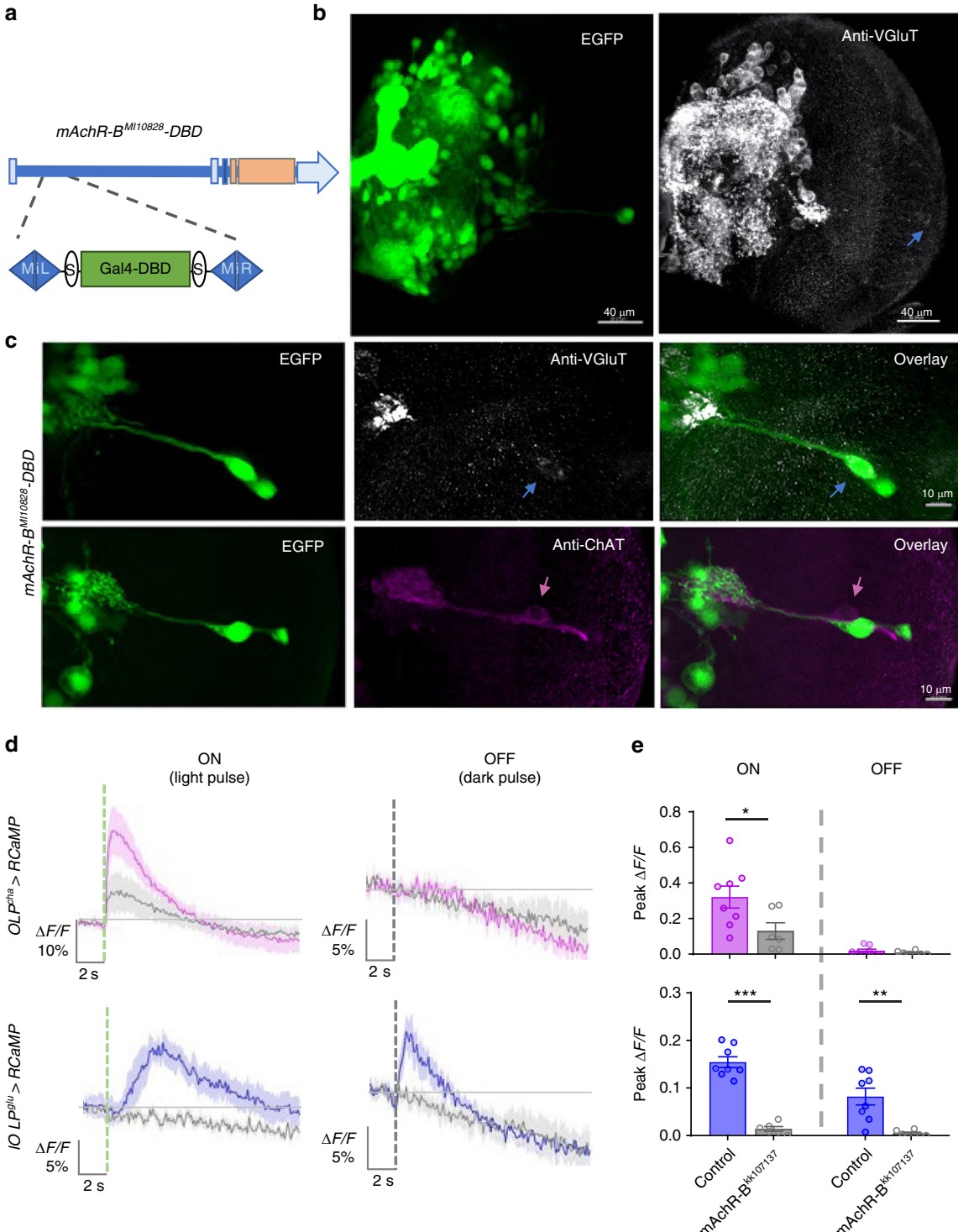

**Fig. 4 mAchR-B mediates light-induced inhibition of glu-OLP. a** Schematic diagram illustrating the insertion of a Gal4-DBD element into the 5′-UTR region of the mAchR-B gene. Orange bar: coding exons. Light blue bar: introns. **b** The mAchR-B enhancer line reveals broad expression of the receptor in the third instar larval brain. Representative projected confocal images with EGFP expression driven by the mAchR-B enhancer (green) and anti-VGluT staining (gray) are shown. Blue arrow: glu-lOLP. **c** mAchR-B expresses in glu-lOLP but not cha-lOLP. The mAchR-B enhancer-driven EGFP signal colocalizes with the VGluT-positive glu-lOLP (blue arrow), but not with the ChAT-positive cha-lOLP (pink arrow). Representative projected confocal images are shown. Scale bars are as indicated. **d, e** Expression of mAchR-B$^{RNAi}$ dampens cha-lOLP's ON response and eliminates both glu-lOLP's ON and OFF responses. The dashed green and gray lines indicate the 100 ms light or dark pulse, respectively. The genotypes are as indicated. The **d** average traces of the changes in lOLP > RCaMP signals and **e** quantification of peak values of changed intensity (Δ$F/F$) are shown. Shaded areas on traces and error bars on quantifications represent SEM. Control, $n = 8$; mAchR-B$^{KK107137}$, $n = 6$. cha-lOLP, ON: $p = 0.0388$, $t = 2.320$, df = 12; cha-lOLP, OFF: $p = 0.3201$, $t = 1.037$, df = 12; glu-lOLP, ON: $p < 0.0001$, $t = 10.09$, df = 12; glu-lOLP, OFF: $p = 0.0028$, $t = 3.736$, df = 12. Statistical significance was determined by Student's $t$ test. $p \geq 0.05$ was considered not significant, ***$p < 0.001$, **$p < 0.01$, and *$p < 0.05$

induced inhibition of glu-lOLP (Fig. 4d, e). Knockdown of mAchR-B also significantly dampened the light responses in cha-lOLP (Fig. 4d, e), suggesting that eliminating the inhibition of glu-lOLP impacts cha-lOLP's light response. However, because the knockdown of mAchR-B was performed in both lOLPs, further evidence is needed to elucidate the interaction between the lOLPs.

To confirm mAchR-B's function, we performed additional experiments examining light-induced calcium transients using two-photon recordings of GCaMP6f driven by lOLP[glu]-Gal4, which showed significantly reduced responses in glu-lOLP with mAchR-B knockdown (Supplementary Fig. 6), supporting the critical role of the receptor in mediating light-induced inhibition on glu-lOLP.

**Blocking Gαo signaling alters glu-lOLP's calcium responses.** We next examined an RNAi line targeting Gαo, the G protein subunit coupled to mAchR-B, to determine if it mediates mAchR-B signaling in glu-lOLP. Knocking down Gαo completely eliminated the dark pulse-induced OFF response (Supplementary Fig. 5b, c). Unexpectedly, knocking down Gαo also generated a distinct phenotype in glu-lOLP, producing an immediate calcium rise upon light stimulation rather than the typical biphasic response (Fig. 5a, b). Blocking Gαo activity in glu-lOLP by Pertussis toxin (PTX) expression, which specifically inhibits Gαo in *Drosophila*[43], also eliminated the initial calcium reduction and accelerated the light-induced calcium rise without significantly affecting its amplitude, an effect observable at both the soma and terminal regions of glu-lOLP (Fig. 5c, d).

This immediate, light-induced calcium increase revealed by disrupting Gαo signaling suggests that, besides mAchR-B/Gαo-mediated inhibition, light induces additional physiological events that lead to calcium increases in glu-lOLP. These events are masked by the initial inhibition and are only observed when mAchR-B/Gαo signaling is strongly affected. The mAchR-B[RNAi] line (mAchR-B[KK107137]) from early experiments was less effective in knocking down receptor activity and produced an unnoticeable effect. To resolve the discrepancy between the mAchR-B- and Gαo-knockdown phenotypes, we examined another RNAi line targeting mAchR-B (mAchR-B[HMS05691]) and observed a light-induced immediate calcium rise with significantly reduced amplitude, similar to those in Gαo-knockdown experiments (Supplementary Fig. 6b). By comparing outcomes generated by blocking mAchR-B/Gαo signaling (Fig. 5a–d, Supplementary Fig. 6), we conclude that the extent and timing of glu-lOLP's activation is regulated by mAchR-B/Gαo signaling.

Additionally, we performed Arclight recordings that revealed dramatic changes in glu-lOLP's voltage responses due to PTX expression. In the control group, we observed a biphasic voltage response in glu-lOLP induced by light stimulation, which produced a large hyperpolarization event followed by a small depolarization (Fig. 5e, f). This response is temporally correlated with the biphasic calcium transients observed in the terminal region of glu-lOLP (Fig. 5c, d). Strikingly, the expression of PTX in glu-lOLP switched the light-induced hyperpolarization to a depolarization, consistent with eliminating the initial reduction of the calcium and producing an immediate calcium rise in glu-lOLP (Fig. 5e, f).

Our genetic studies confirm the role of mAchR-B/Gαo signaling in mediating light-induced inhibition of glu-lOLP and reveal the complexity of glu-lOLP's light responses, which contain multiple signaling events that cooperatively regulate the direction and timing of the neuron's physiological output. Importantly, we found that PTX expression in glu-lOLP eliminates its OFF response while accelerating the light-induced calcium rise,

effectively transforming glu-lOLP into an ON-selective cell. Instead of transmitting light decrements, glu-lOLP expressing PTX transmits light increments to downstream VPNs, potentially disrupting the separation of the ON and OFF channels.

**glu-lOLP regulates light responses in cha-lOLP and VPNs.** To examine how glu-lOLP interacts with cha-lOLP and the downstream projection neurons, we expressed PTX in glu-lOLP using lOLP[glu]-Gal4 and monitored the light-induced calcium responses in all three OLPs using OLP-LexA-driven expression of GCaMP6f. Consistent with our earlier observations, PTX expression eliminated the light-induced calcium reduction and accelerated the delayed calcium rise in glu-lOLP without affecting its amplitude. Importantly, this fast activation of glu-lOLP led to significant reductions in light-induced calcium responses in cha-lOLP (Fig. 6a, b), suggesting that glu-lOLP acts as an inhibitory input to cha-lOLP and that disrupting the temporal separation between the interneurons' light responses affects cha-lOLP's response to light. The direct synaptic interactions between the two lOLPs demonstrated by the connectome study[17], the inhibitory effect of cholinergic inputs from both photoreceptors and cha-lOLP on glu-lOLP (Fig. 3a, b), and the dampened light response in cha-lOLP generated by accelerated activation of glu-lOLP (Fig. 6a, b) support a model of reciprocal inhibitory interactions between glu-lOLP and cha-lOLP.

Blocking Gαo signaling in glu-lOLP also revealed close interactions between pOLP and glu-lOLP. PTX expression in glu-lOLP significantly reduced the latency of light-induced calcium rise in pOLP without affecting its amplitude (Fig. 6a, b). Due to the matching temporal profiles both with and without the PTX expression in glu-lOLP, we concluded that the light-induced calcium increase in pOLP is driven by glu-lOLP's activities (Figs. 2a, b, 6a, b, Supplementary Fig. 3b). Because the connectome study did not find direct synaptic interactions between the pair, this effect may be indirect, although the close physical proximity between glu-lOLP and pOLP also suggests interactions via gap junctions[17].

Next, we examined how altering glu-lOLP kinetics affected larval ventral lateral neurons (PDF-LaNvs or LNvs), an additional group of VPNs. LNvs regulate the circadian rhythm in both larval and adult *Drosophila*[44,45]. Besides receiving synaptic inputs from the lOLPs, LNvs are also contacted directly by both Rh5- and Rh6-PRs (Fig. 6c, Supplementary Fig. 7)[17] and are activated by cholinergic inputs through nAchR signaling[46]. Additionally, previous studies demonstrated that glutamatergic inputs inhibit larval LNvs through the glutamate-gated chloride channel GluCl⁻ [47].

Using an LNv-specific enhancer Pdf-LexA, we expressed GCaMP6f in LNvs and recorded robust light-elicited calcium responses in the LNvs' axon terminal region[30] (Fig. 6d). Importantly, expressing PTX in glu-lOLP significantly reduced both the amplitude and the duration of these calcium transients (Fig. 6d–f), suggesting that glu-lOLP also provides inhibitory inputs onto the LNvs and that changing the temporal profile of glu-lOLP's activation influences LNvs' light responses.

**Light elicits a delayed glutamate release from glu-lOLP.** To determine the specificity and physiological relevance of the delayed calcium rise in glu-lOLP, we examined glutamate transients on LNv dendrites from glu-lOLP using a genetically encoded glutamate sensor iGluSnFR[48].

Upon light stimulation, iGluSnFR signals in the LNv dendrite region exhibit a biphasic pattern with a rapid increase in fluorescence followed by a wide peak 2 s after stimulation (Fig. 7a–c). While the fast peak of the glutamate transient is likely generated by dorsal neuron 1 (DN1), a previously identified glutamatergic input to LNvs[47], the delayed peak has a latency that

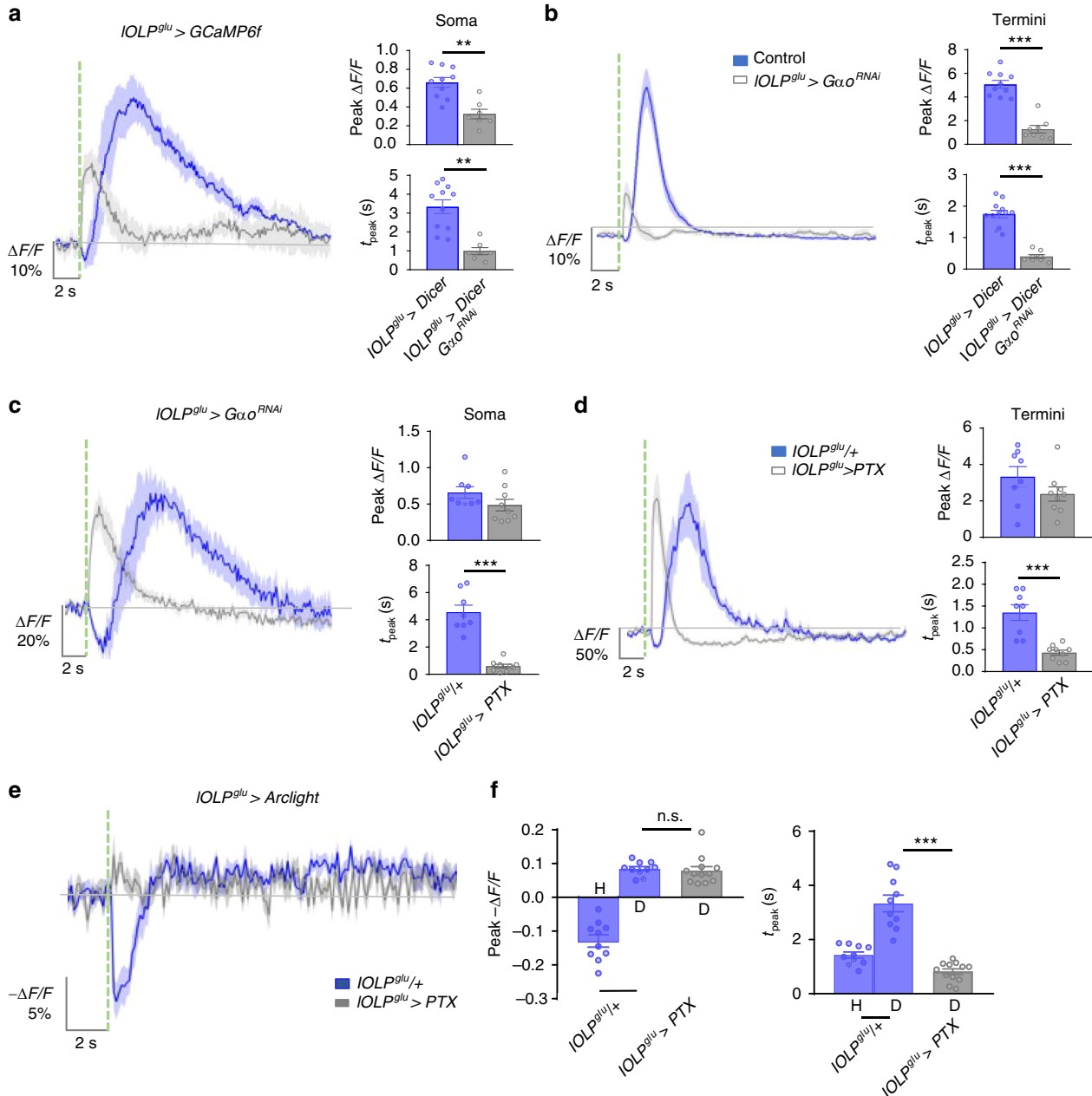

**Fig. 5** Gαo signaling regulates light-evoked responses in glu-lOLP. **a, b** RNAi knockdown of Gαo reduces the amplitude and latency of the calcium rise in glu-lOLP. Average traces of the changes in GCaMP signals (left) and the quantifications of peak value and peak time (right) of changed intensity (ΔF/F) are shown. IOLP$^{glu}$ > Dicer, $n = 10$; IOLP$^{glu}$ > Dicer, Gαo$^{RNAi}$, soma: $n = 7$; termini: $n = 8$. Soma—peak value: $p = 0.0005$; peak time: $p = 0.0002$. Termini—peak value: $p < 0.0001$; peak time: $p < 0.0001$. Statistical significance was determined by Student's $t$ test. **c, d** Expression of the Gαo inhibitor PTX accelerates the light-induced calcium rise in glu-lOLP without affecting its amplitude. IOLP$^{glu}$ > GCaMP6f signals were collected at the soma and termini of glu-lOLPs. Average traces of the changes in GCaMP signals (left) and the quantifications of the peak value and peak time (right) of changed intensity (ΔF/F) are shown. IOLP$^{glu}$/+, $n = 8$; IOLP$^{glu}$ > PTX, $n = 9$. Soma—peak value: $p = 0.145$; peak time: $p < 0.0001$. Termini—peak value: $p = 0.1723$; peak time: $p = 0.0001$. Statistical significance was determined by Student's $t$ test. **e, f** PTX expression transforms light-induced hyperpolarization into depolarization in glu-lOLP. Light-evoked voltage changes in glu-lOLP measured by Arclight expression driven by IOLP$^{glu}$-Gal4 exhibits a biphasic response, a large hyperpolarization (H) followed by a small depolarization (D), in the control group. PTX expression eliminates the hyperpolarization and reveals a depolarization. Average traces of changes in Arclight signals (left) and the quantifications of the peak value and peak time (right) of changed intensity (−ΔF/F) are shown. IOLP$^{glu}$/+, $n = 10$, IOLP$^{glu}$ > PTX, $n = 12$. Peak value: ANOVA: $p < 0.0001$, $F = 66.92$, df = 35; IOLP$^{glu}$/+-IOLP$^{glu}$ > PTX: $p = 0.9883$. Peak time: ANOVA: $p < 0.001$, $F = 42.32$, df = 35; IOLP$^{glu}$/+-IOLP$^{glu}$ > PTX: $p < 0.0001$. Shaded areas on traces and error bars on quantifications represent SEM. The dashed green line represents a 100 ms light pulse at 561 nm. Statistical significance was determined by one-way ANOVA with post hoc Tukey's multiple comparison's test. n.s.: $p \geq 0.05$ was considered not significant, **$p < 0.01$, ***$p < 0.001$

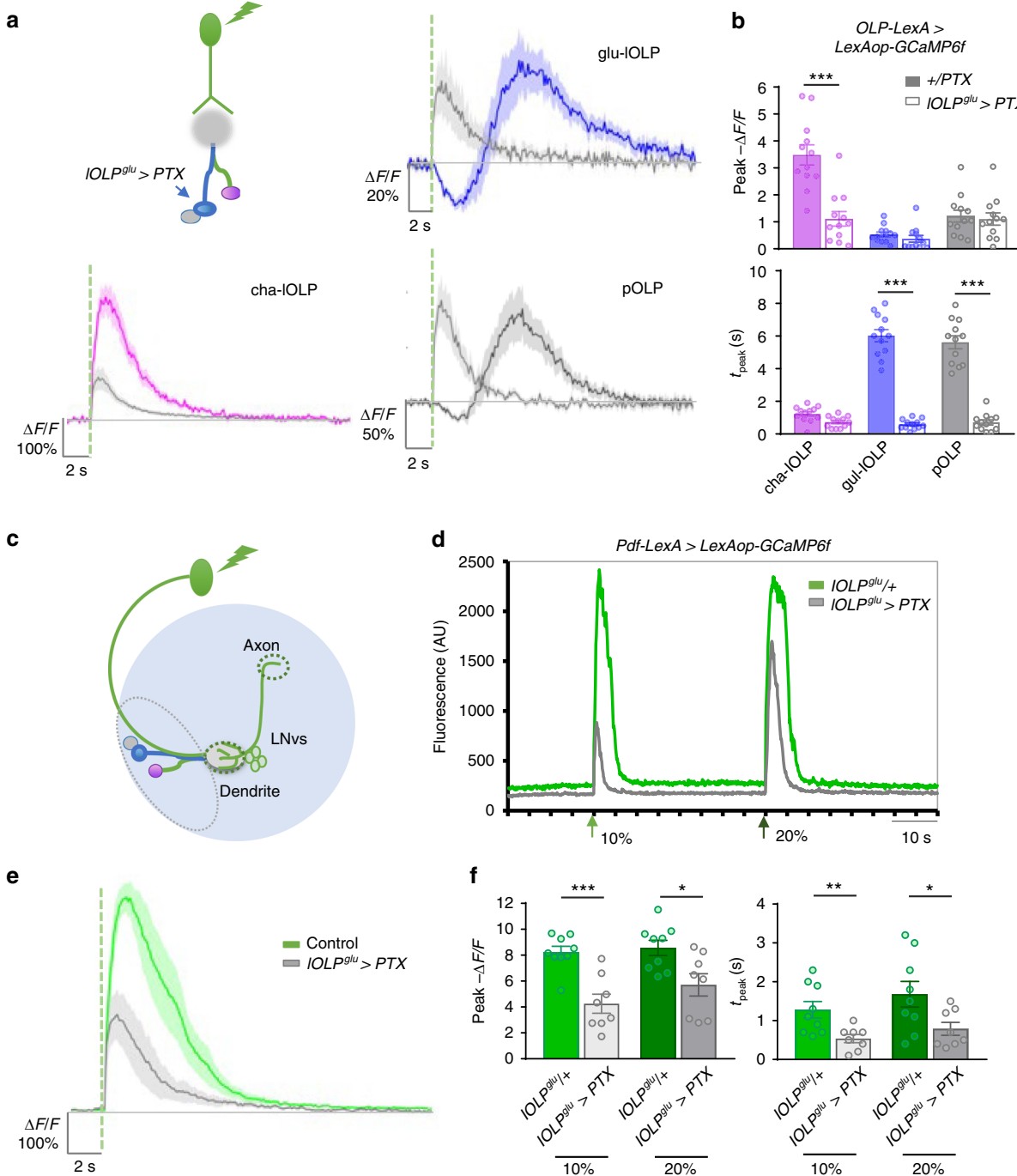

**Fig. 6** Glu-lOLP regulates light responses in cha-lOLP, pOLP, and LNvs. **a**, **b** glu-lOLP inhibits cha-lOLP and activates pOLP. **a** Schematic diagram illustrating the experimental design, with PTX expression restricted to glu-lOLP while light responses in the three OLPs are reported by OLP-LexA-driven LexAop-GCaMP6f. PTX expression accelerates glu-lOLP's and pOLP's activations and dampens the response in cha-lOLP. Average traces of changes in GCaMP signals are shown. The dashed green line represents a 100 ms light pulse at 561 nm. Shaded areas on traces represent SEM. **b** Quantifications of the peak value and peak time of changed intensity ($\Delta F/F$) of GCaMP6f are shown. $n = 12$ in all groups. Peak values—cha-lOLP: $p < 0.001$; glu-OLP: $p = 0.9967$; pOLP: $p = 0.9995$. Peak times: cha-lOLP: $p = 0.6956$; glu-lOLP: $p < 0.001$; pOLP: $p < 0.001$. Error bars represent SEM. Statistical significance was determined by one-way ANOVA with post hoc Tukey's multiple comparison's test. **c** Schematic diagram showing the optical recording of light-induced responses in LNvs. Pdf-LexA-driven GCaMP6f signals are recorded in the axon terminal region (dashed circle). **d** A representative raw trace is shown. 100 ms light stimulations (green arrows) were delivered with either 10% or 20% laser power and induced robust calcium increases in LNvs. Compared to the controls, PTX expression in glu-lOLP (lOLP$^{glu}$ > PTX) leads to dampened responses with reduced durations. **e**, **f** PTX expression in glu-lOLP reduced the light-induced calcium response in LNvs. Average traces (**e**) and quantifications (**f**) of the peak value (left) and peak time (right) of changed intensity ($\Delta F/F$) of GCaMP6f signals are shown. The dashed green line represents a 100 ms light stimulation at 10% intensity. Two different intensities of light stimulation generated similar results. Control: $n = 9$; lOLP$^{glu}$ > PTX: $n = 8$. Peak values: 10%: $p = 0.0003$, $t = 4.758$, df = 15; 20%: $p = 0.0136$, $t = 2.795$, df = 15. Peak times: 10%: $p = 0.0090$, $t = 2.998$, df = 15; 20%: $p = 0.0354$, $t = 2.311$, df = 15. Statistical significance was determined by Student's $t$ test. $*p < 0.05$, $**p < 0.01$, and $***p < 0.001$

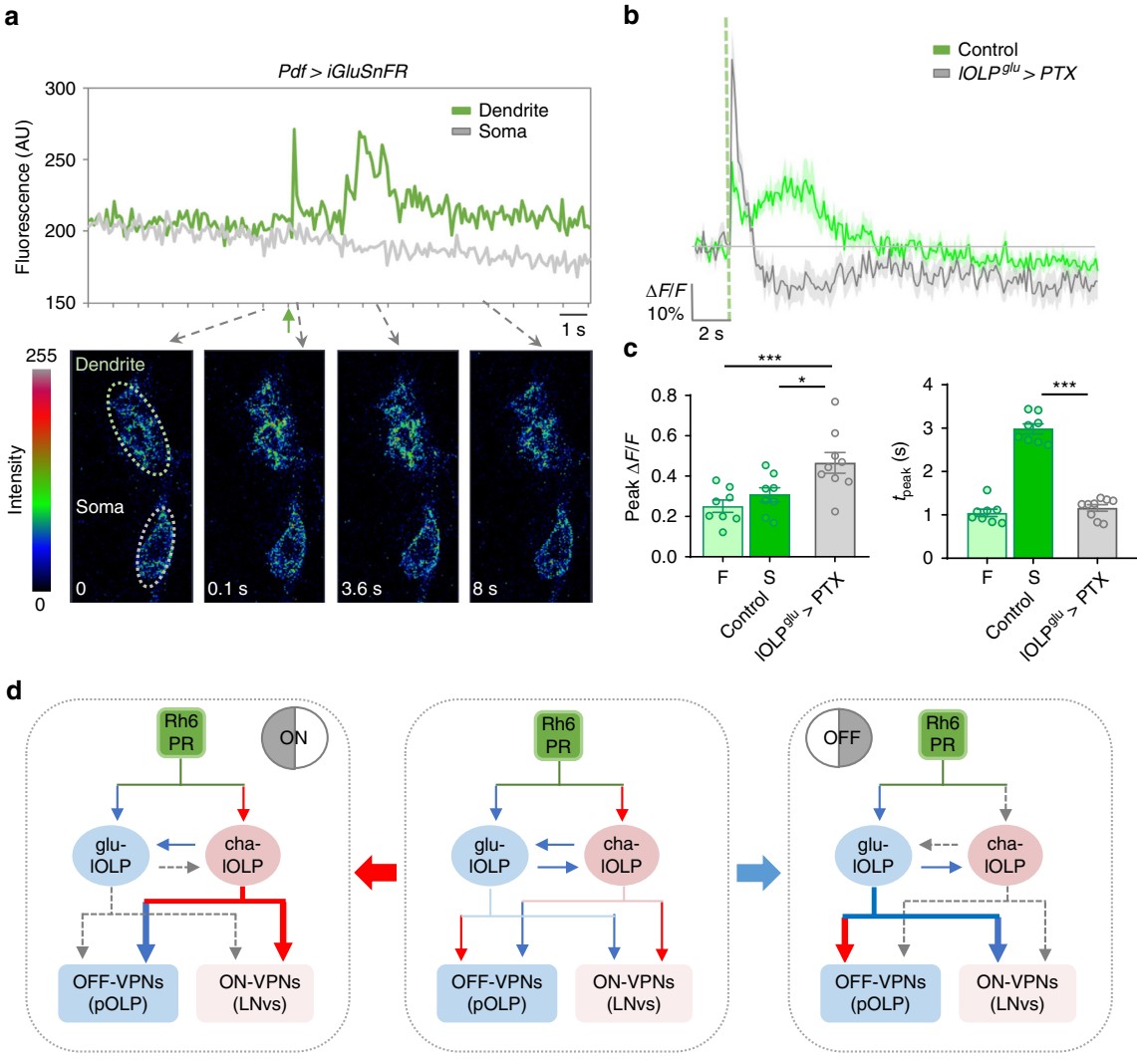

**Fig. 7** Light elicits delayed glutamate release from glu-lOLP. **a** Top: A representative raw trace of the light-induced glutamate transient generated by iGluSnFR recording on LNv dendrites. Bottom: representative frames from the same recording show increased iGluSnFR signals in the LNvs' dendritic region but not the soma. The green arrow indicates the light pulse. **b, c** A light pulse (green dashed line) induces a biphasic release of glutamate onto the LNv dendrites. PTX expression in glu-lOLP eliminates the slow phase of the glutamate transient. **b** Average traces of the glutamate transients. **c** Quantification of changed intensity ($\Delta F/F$) in iGluSnFR signals on LNv dendrites. The fast phase (F) and slow phase (S) have different latencies and similar amplitudes. PTX expression in glu-lOLP eliminates the slow phase and generates one fast transient with an increased amplitude compared to the controls. Control: $n = 8$; lOLP$^{glu}$ > PTX: $n = 9$. Peak value: ANOVA: $p = 0.0033$, $F = 7.473$, df = 22; F/lOLP$^{glu}$ > PTX: $p = 0.0034$; S/lOLP$^{glu}$ > PTX: $p = 0.0318$. Peak time: $p < 0.0001$, $F = 134.6$, df = 22; F/lOLP$^{glu}$ > PTX: $p = 0.6569$; S/lOLP$^{glu}$ > PTX: $p < 0.0001$. Shaded areas on traces and error bars on quantifications represent SEM. Statistical significance determined by one-way ANOVA with post hoc Tukey's multiple comparison's test. *$p < 0.05$, ***$p < 0.001$. **d** A proposed model illustrating the emergence of ON and OFF selectivity in the larval visual circuit. Middle: lOLPs detect and transmit the ON and OFF signals in the larval visual circuit. Light induces acetylcholine release from Rh6-PRs, which activates cha-OLP and inhibits glu-lOLP through differentially expressed AChRs. During an ON response (left panel), the cholinergic transmission is dominant, activating ON-VPNs and suppressing OFF-VPNs. During an OFF response (right panel), glu-lOLP activates OFF-VPNs and suppresses ON-VPNs. The synaptic interactions are labeled blue for inhibitory and red for excitatory

matches the light-induced calcium response in glu-lOLP. Importantly, PTX expression in glu-lOLP eliminates this slow peak and produces only a single fast glutamate transient on LNv dendrites (Fig. 7a–c), indicating glu-lOLP as the source of the delayed glutamate transient and a major glutamatergic input to the LNvs. These results also indicate that the temporal features of glu-lOLP's activity are preserved and transmitted to downstream VPNs through timed glutamate release.

Together, our results show that altering the temporal profile of glu-lOLP's activation strongly influences light responses in both visual interneurons and projection neurons, supporting the functional significance of the temporal control of glutamatergic

transmission in the larval visual circuit. In addition, our studies also validated the reciprocal interactions between cha- and glu-lOLP and demonstrated the ability of glu-lOLP to elicit distinct physiological responses in different types of VPNs.

**glu-lOLP is required for dark-induced behavioral responses**. To illustrate the potential roles for lOLPs in transmitting ON and OFF signals from the PRs to the VPNs, we propose a model with three components based on the connectivity map and our findings. First, the pair of lOLPs act as ON and OFF detectors and exhibit distinct responses to light increments and decrements.

The sign inversion required for OFF detection in glu-OLP is mediated by the mAchR-B receptor. Second, while cha-lOLP displays clear ON selectivity, glu-lOLP shows a biphasic response to light. Its OFF selectivity emerges from the temporal control of its activity by mAchR-B/Gαo signaling. Third, extending our findings in the LNvs and pOLP to the rest of the VPNs, we propose that, although downstream VPNs receive both cholinergic and glutamatergic inputs, there are specific groups of ON (ON-VPNs) vs. OFF-responsive VPNs (OFF-VPNs) that are functionally separated by their molecular compositions. ON-VPNs, such as LNvs, are activated by cholinergic signaling and inhibited by glutamatergic signaling while OFF-VPNs, such as pOLP, behave oppositely. Although additional physiological studies on other VPNs are needed to validate this model, this functional separation of VPNs is a plausible solution to preserving and transmitting the ON and OFF signals at the level of VPNs given the lack of anatomical segregation of ON and OFF pathways (Fig. 7d).

This model suggests that an ON response is dominated by cholinergic transmissions from cholinergic PRs and cha-lOLP, while inhibition of glu-lOLP via mAchR-B/Gαo signaling ensures that only the ON-VPNs are active. During an OFF response, with no cholinergic input, the glu-lOLP is solely responsible for activating OFF-VPNs. During behavioral regulation, cha-lOLP likely modulates the strength and duration of the light-induced response and glu-lOLP is essential for initiating dark-induced behavioral responses (Fig. 7d).

To identify the functional role of glu-lOLP, we performed behavioral experiments to quantitatively analyze larval responses towards dark-light and light-dark transitions during negative phototaxis. Previous studies indicated that, upon encountering a reduction in light intensity at a light-dark boundary, larvae increase their pausing frequencies. On the other hand, upon sensing an increase in light intensity at a dark-light boundary, larvae increase their turning frequencies[20].

Behavioral tests in Rh6 mutants showed that phototransduction mediated by Rh6-PRs is necessary for dark-induced pausing. In addition, genetic manipulations of glu-lOLP, including the expression of the cell death genes *rpr* and *hid*, the Gαo inhibitor PTX, and the RNAi transgene targeting the mAchR-B receptor all generated significant reductions of dark-induced pausing behavior, whereas corresponding Gal4 and UAS control larvae showed robust dark-induced pausing (Fig. 8a, b). These results indicate that either the ablation of glu-lOLP or the blocking of mAchR-B/Gαo signaling affects the dark-induced behavioral response, supporting the critical functions of glu-lOLP and mAchR-B/Gαo signaling in mediating OFF detection.

In contrast, although Rh6 mutants also exhibit deficits in light-induced increases in turning frequency, this behavioral response to light was largely unaffected by genetic manipulations of glu-lOLP. This result demonstrates that glu-lOLP is not involved in regulating larval responses towards a dark-light transition and that altering glu-lOLP's activation does not change the visual circuit's basic light responsiveness (Fig. 8c, d).

Although further experiments are needed to address the behavioral relevance of cha- and glu-lOLP in regulating other visually guided behaviors, our studies measuring dark-induced pausing behavior indicate that glu-lOLP mediates OFF detection in the larval visual circuit, consistent with our model.

## Discussion

The *Drosophila* larval visual circuit, with its small number of components and complete wiring diagram, provides a powerful model to study how specific synaptic interactions support visual computation. Built on knowledge obtained from connectome and behavioral analyses, our physiological and genetic studies revealed unique computational strategies utilized by this simple circuit for processing complex outputs. Specifically, our results indicate that ON vs. OFF discrimination emerges at the level of the lOLPs, a pair of second-order visual interneurons. In addition, we demonstrate the essential role of glu-lOLP, a single glutamatergic interneuron, in mediating OFF detection at both the cellular and behavior levels and identify mAchR-B/Gαo signaling as the molecular machinery regulating its physiological properties.

Functional imaging studies using genetically encoded calcium and voltage indicators provide us with valuable information regarding the physiological properties of synaptic interactions among larval visual interneurons and projection neurons. However, our optical recording approaches have certain technical limitations, including the kinetics and sensitivities of the voltage and calcium sensors, as well as our imaging and visual stimulation protocols. In addition, although glu-lOLP displays a biphasic response towards the light stimulation, we quantified calcium reductions and increases for only the initial set of physiological characterizations (Supplementary Fig. 4). Compared to the delayed calcium rise, the light-induced calcium reductions have low amplitudes and high variabilities, possibly due to the half-wave rectification of the intracellular calcium previously described in adult visual interneurons[13,29]. For the genetic experiments, we then focused on evaluating the activation of glu-lOLP, which is reflected by the increase of intracellular calcium signals that lead to neurotransmitter release.

To process light and dark information in parallel, both mammalian and adult fly visual systems utilize anatomical segregation to reinforce split ON and OFF pathways[49]. In the larval visual circuit, however, almost all VPNs receive direct inputs from both cha-lOLP and glu-lOLP as well as the Rh5-PRs[17]. Therefore, the response signs of the VPNs cannot be predicted by their anatomical connectivity to ON and OFF detectors. Based on the cumulative evidence obtained through genetic, anatomical, and physiological studies, we propose that temporal control of inhibition potentially contributes to ON vs. OFF discrimination in larvae. While cha-lOLP displays clear ON selectivity, the OFF selectivity in glu-lOLP is strengthened by the extended suppression of its light response by mAchR-B/Gαo signaling. This temporal control may also produce a window of heightened responsiveness in cha-lOLP and ON-VPNs towards light signals, similar to the case in mammalian sensory systems where the temporal delay of input-evoked inhibition relative to excitation sharpens the tuning to preferred stimuli (reviewed in ref. [50]). Together, the temporal separation between cholinergic and glutamatergic transmission could reinforce the functional segregation in the VPNs and lead to distinct transmissions of ON and OFF signals. Although further functional validations are needed, temporal control of inhibition provides an elegant solution that may be of general use in similar contexts where parallel processing is achieved without anatomically split pathways.

The connectome study identified ten larval VPNs which receive both direct and filtered inputs from two types of PRs and transmit visual information to higher brain regions, including four LNvs (PDF-LaNs), five LaN, nc-LaN1, and two pVL09, VPLN, and pOLP[17]. Based on our studies on LNvs and pOLP, we expect to observe the functional diversity in VPNs generated by differential expression of neurotransmitter receptors or molecules involved in electric coupling. Besides basic ON vs. OFF discrimination, VPNs are also involved in a variety of visually guided behaviors[19,20,51]. The temporal regulation of their glutamatergic and cholinergic inputs as well as the local computation within the LON are among potential cellular mechanisms that increase the VPNs' capability to process complex visual information. Further physiological and molecular studies of the VPNs and behavioral

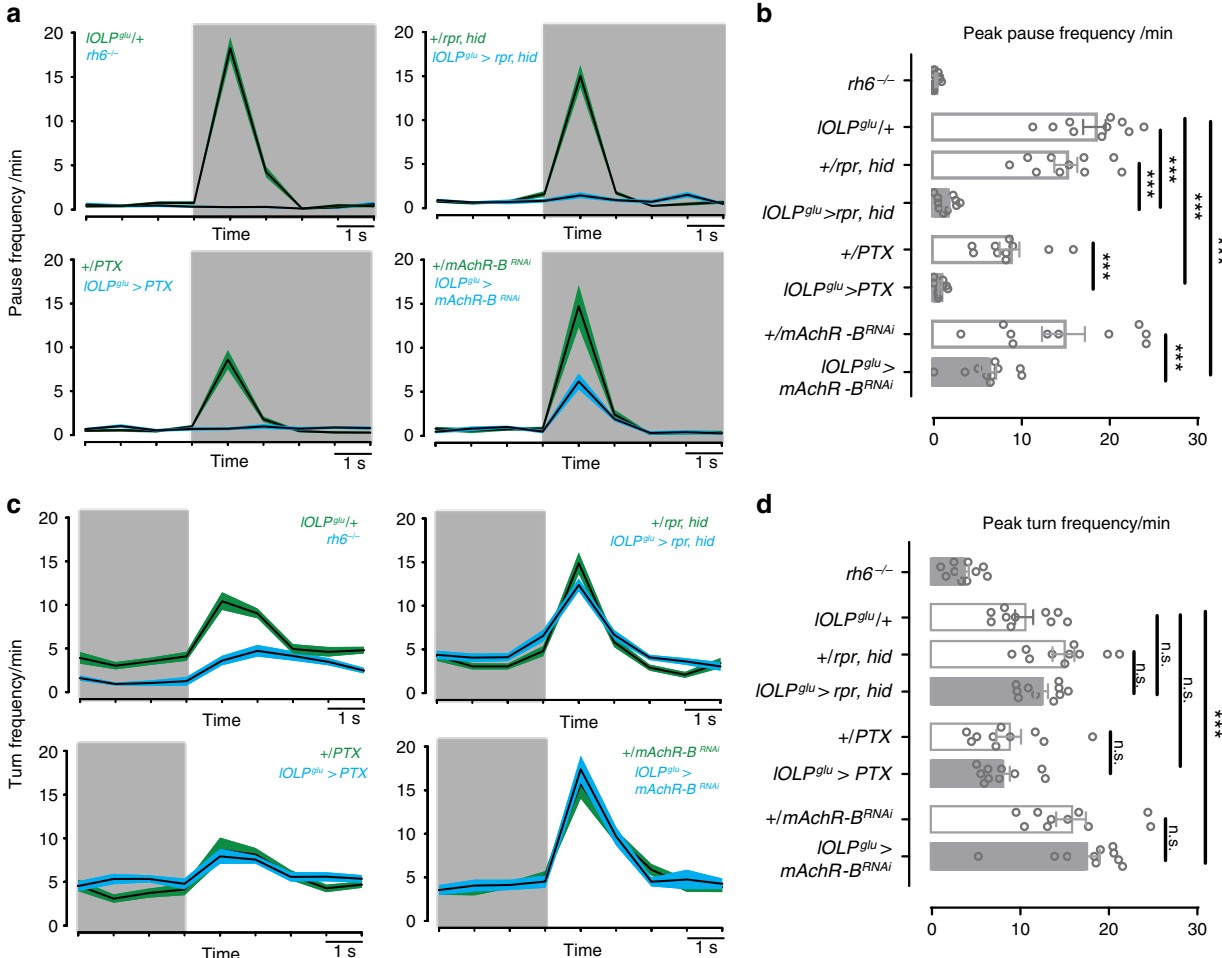

**Fig. 8** Glu-lOLP is required for dark-induced pausing behavior. **a**, **b** Genetic manipulations of glu-lOLP affect dark-induced pausing behavior in larvae. **a** Plots of average pause frequency are shown. The transition from light to dark is indicated by the shade of the area. **b** Quantification of dark-induced pause frequency reveals the critical role of Rh6-PRs and glu-lOLP in this behavioral response. Statistical significance determined by one-way ANOVA: $p < 2e − 16$, $F = 35.6$, df $= 7$, 72 followed by post hoc Dunnetts's multiple comparison's test: lOLP$^{glu}$/+-lOLP$^{glu}$ > rpr, hid: $p < 1e − 04$, $t = 9.907$; +/rpr,hid-lOLP$^{glu}$ > rpr, hid: $p < 1e − 04$, $t = 7.990$; lOLP$^{glu}$/+- lOLP$^{glu}$ > PTX: $p < 1e − 04$, $t = 10.337$; +/PTX-lOLP$^{glu}$ > PTX: $p = 0.000108$, $t = 4.648$; lOLP$^{glu}$/+-lOLP$^{glu}$ > mAChR-B$^{RNAi}$: $p < 1e − 04$, $t = 7.120$; +/mAChR-B$^{RNAi}$-lOLP$^{glu}$ > mAChR-B$^{RNAi}$: $p < 1e − 04$, $t = 5.044$. ***$P < 0.001$. $n = 10$ for each genotype. **c**, **d** The light-induced increase in turning frequency is reduced in Rh6 mutants but unaffected by glu-lOLP manipulations. **c** Plots of average turn frequency are shown. The transition from dark to light is indicated by the shade of the area. **d** Quantifications of the light-induced turn frequency reveals that glu-lOLP does not influence the behavioral responses induced by the dark to light transition. Statistical significance determined by one-way ANOVA: $p < 7.16e − 12$, $F = 14.93$, df $= 7$, 72 followed by post hoc Dunnetts's multiple comparison's test: lOLP$^{glu}$/+-lOLP$^{glu}$ > rpr, hid: $p = 0.753$, $t = −1.159$; +/rpr,hid-lOLP$^{glu}$ > rpr, hid: $p = 0.538$, $t = 1.472$; lOLP$^{glu}$/+-lOLP$^{glu}$ > PTX: $p = 0.530$, $t = 1.483$; +/PTX-lOLP$^{glu}$ > PTX: $p = 0.996$, $t = 0.445$; lOLP$^{glu}$/+-lOLP$^{glu}$ > mAChR-B$^{RNAi}$: $p < 0.001$, $t = −4.134$; +/mAChR-B$^{RNAi}$-lOLP$^{glu}$ > mAChR-B$^{RNAi}$: $p = 0.849$, $t = −0.996$. n.s., $p > 0.05$, ***$p < 0.001$. $n = 10$ for each genotype

experiments targeting specific visual tasks are needed to elucidate their specific functions.

Besides the similarities observed between larval lOLPs and the visual interneurons in the adult fly visual ganglia, we can also draw an analogy between lOLPs and interneurons in mammalian retinae based on their roles in visual processing. Cha-lOLP and glu-lOLP carry sign-conserving or sign-inverting functions and activate ON- or OFF-VPNs, respectively, performing similar functions as bipolar cells in mammalian retinae[52]. At the same time, lOLPs also provide inhibitory inputs to either ON- or OFF-VPNs and thus exhibit the characteristics of inhibitory amacrine cells[53]. The dual role of lOLPs is the key feature of larval ON and OFF selectivity, which likely evolved to fulfill the need for parallel processing using limited cellular resources.

Lastly, our studies reveal signaling pathways shared between mammalian retinae and the larval visual circuit. Although the two systems are constructed using different neurochemicals, Gαo

signaling is responsible for producing sign inversion in both glu-lOLP and the ON-bipolar cell[54]. In mGluR6-expressing ON-bipolar cells, light increments trigger Gαo deactivation, the opening of TrpM1 channels, and depolarization. In larval glu-lOLP, how light induces voltage and calcium responses via mAChR-B signaling has yet to be determined. Gαo is known to have functional interactions with a diverse group of signaling molecules including potassium and calcium channels that could directly link the light-elicited physiological changes in glu-lOLP[55]. Genetic and physiological studies in the larval visual circuit will facilitate the discovery of these target molecules and contribute to the mechanistic understanding of visual computation.

## Methods

**Fly strains.** The following lines were used (in the order of appearance in figures): 1. GMR72A10-LexA, Bloomington Stock Center (BDSC): 54191; 2. LexAop-

mCherry, BDSC: 52272; 3. ChAT-Gal4, UAS-EGFP, BDSC: 6793; 4. GMR84E12-Gal4, (no longer available at BDSC); 5. GMR72E03-Gal4, BDSC: 47445; 6. UAS-mCD8::GFP, BDSC: 5136; 7. UAS-RedStinger, BDSC: 8547; 8. UAS-GCaMP6f, BDSC: 42747; 9. Lexop-GCaMP6f, BDSC: 44277; 10. $rh6^1$; 11. UAS-ArcLight, BDSC: 51057; 12. UAS-RCaMP, BDSC: 51928; 13. mAchR-B$^{MI10828}$-Gal4-DBD; 14. Tub-dVP16AD, UAS-EYFP; 15. UAS-Dcr-2, BDSC: 24651; 16. mAchR-B$^{RNAi}$, Vienna *Drosophila* Resource Center (VDRC): KK107137; 17. Gαo$^{RNAi}$: HMS01129, BDSC: 34653; 18. mAchR-B$^{RNAi}$, BDSC: 67775; 19. UAS-PTX; 20. Pdf-LexA.

Stock #10 is a gift from Dr. Claude Desplan. Stock #19 is a gift from Dr. Gregg Roman. Stock #20 is a gift from Dr. Michael Rosbash. The rest of the lines were from BDSC or VDRC.

Stock #13 was generated using the MI10828 MiMIC insertion in the first intron of the mAchR-B gene. A gene-trap cassette containing the Gal4-DBD sequence in place of the original Gal4 sequence was inserted into MI10828 using ΦC31 technology by Rainbow Transgenic Flies (Camarillo, CA)[41,42,56]. Stock #14 is as described[42].

**Fly culture**. Fly stocks are maintained using the standard cornmeal medium in humidity-controlled 25 °C incubators with a 12-h light:12-h dark schedule. Light intensity in the incubator is around ~1000 lx. All immunohistochemistry studies and optical imaging were performed using wandering third instar larvae.

**Immunohistochemistry**. Larval brains were collected from wandering third instar larvae and fixed in 4% paraformaldehyde/phosphate-buffered saline (PBS) at room temperature for 30 min, followed by washing in PBST (0.3% Triton X-100 in PBS) and incubating in the primary antibody overnight at 4 °C. Brains were then washed with PBST and incubated in the secondary antibody at room temperature for 1 h before final washes in PBST and mounting on the slide with antifade mounting solution. Primary antibodies used were rabbit anti-GFP antibody (Abcam, Ab6556, 1:200), mouse anti-ChAT (DSHB, ChAT4B1, 1:10), and rabbit anti-VGluT (a gift from Dr. DiAntonio, 1:5000). Secondary antibodies used were goat anti-rabbit Alexa 633 (Invitrogen, A-21070) and donkey anti-mouse CY3 (Jackson Immu-noResearch Labs, 715165150). Whole-mount brain samples were treated and mounted on slides using the SlowFade Antifade kit (Life Technologies, S2828).

**Confocal and two-photon imaging**. Fixed samples were imaged on a Zeiss 700 confocal microscope with a ×40 oil objective. Serial optical sections were obtained from whole-mount larval brains with a typical resolution of 1024 μm × 1024 μm × 0.5 μm. Two-photon imaging of genetically encoded sensors, including GCaMP6f and Arclight, was performed on a Zeiss LSM780 confocal microscope equipped with a Coherent Vision II multiphoton laser. Time-lapse live imaging series were acquired at 100 ms per frame for 1000 frames using a ×40 water objective with the two-photon laser tuned to 920 nm. Typical resolution for a single optical section is 256 μm × 96 μm with 3× optical zoom. RCaMP signals were collected with similar optical and temporal resolutions, using either the two-photon laser tuned to 1040 nm (Fig. 3b) or the confocal laser at 561 nm (Fig. 3c, d, 4d, Supplementary Figs. 5b and 8).

**Visual stimulation**. All optical recordings, except for the experiments described above, were collected using the two-photon laser tuned to 1040 for RCaMP, or 920 nm for GCaMP6f. The preparation was stimulated by 100 ms light pulses. The blue light stimulation at 488 nm or the green light stimulation at 561 nm is produced by an Argon multiline laser set at 488 nm or a DPSS-561 nm laser, respectively. Both lasers are incorporated into the LSM780 confocal microscope and can be controlled by the photobleaching program in the Zen software. The spectral sensitivity of *Drosophila* Rh5 and Rh6 has been previously established[33]. Rh6 detects light within the 400–600 nm range and its maximal spectral sensitivity is ~437 nm, while Rh5 detects light from 350 to 500 nm and its maximal spectral sensitivity is ~508 nm.

The intensity of the light stimulation was adjusted by the power setting of the laser. As measured by a light meter (Thorlabs, Germany, Model: PM100D) equipped with a light sensor (Thorlabs, Germany, Model: S170C), the output was ~39 μW/cm$^2$ for the 561 nm laser and 11.7 μW/cm$^2$ for the 488 nm laser at 20% laser power. At 10% laser power, the output was around 21.5 μW/cm$^2$ for the 561 nm laser and 5.9 μW/cm$^2$ for the 488 nm laser. During a 1000 frame recording collected at 100 ms per frame, two separate light pulses of different wavelengths (488 nm vs. 561 nm) or different intensities (10% vs. 20% laser power) were delivered at the 200th and 600th frames (Supplementary Fig. 3a).

To study the responses of lOLPs to the onset and offset of extended light exposures (Fig. 3c), we collected RCaMP signals using the 561 nm confocal laser with the power setting of 0.5%, while tuning the light cycle using the 488 nm laser with the power setting of 5%. The laser power output during the light exposure was ~3.9 μW/cm$^2$. When the 488 nm laser was turned off, the output was reduced to ~1 μW/cm$^2$.

To measure the ON response using confocal recording of RCaMP (Figs. 3d, 4d, Supplementary Fig. 6) (the response of lOLPs to light pulses), we collected RCaMP signals using the 561 nm confocal laser with the power setting of 0.5–1%, while stimulating the preparation using a 100 ms light pulse generated by the 488 nm laser with the power setting of 20%. The laser power during the recording was ~1–2 μW/cm$^2$ and increased to ~12.5 μW/cm$^2$ with the light pulse.

To measure the OFF response (Figs. 3d, 4d, Supplementary Fig. 5) (the response of lOLP towards light decrements), we recorded RCaMP signals using the confocal laser at 561 nm with the power setting of 5% plus additional illumination using the confocal laser at 488 nm with the power setting of 2%, which produced an output of ~11.7 μW/cm$^2$. The 100 ms dark pulse was delivered by the photobleaching program with no laser activated and therefore produced a reduction of light intensity from ~11.7 to ~0 μW/cm$^2$.

**Larval eye–brain explant preparation for live imaging**. Optical recordings were performed on explant preparations collected during the subjective day between ZT1 and ZT8 (ZT: zeitgeber time in a 12:12 h light-dark cycle; lights-on at ZT0, lights-off at ZT12). Procedures for dissection and preparation of larval brain explants were as described[30]. The eye–brain explant containing the Bolwig's organ, the Bolwig's nerve, eye discs, and the larval brain were dissected in PBS. The explant was carefully separated from the rest of the larval tissue without damaging the optic nerve or brain lobes, transferred into an external saline solution (120 mM NaCl, 4 mM MgCl$_2$, 3 mM KCl, 10 mM NaHCO$_3$, 10 mM glucose, 10 mM sucrose, 5 mM TES, 10 mM HEPES, 2 mM Ca$^{2+}$, pH 7.2), and maintained in a chamber between the slide and cover glass during the recording sessions.

**Imaging data analysis**. Time-lapse imaging series were first processed using the Zen software (Zen-black 2011, Zeiss, Germany). Regions of interest (ROIs) around individual soma or the terminal processes were manually selected for each sample. Examples of raw images of optical recordings using OLP > GCaMP6f with the ROI selection are shown in Supplementary Fig. 3b. A txt. file containing the intensity value of each ROI for individual frames within the time series was generated by the Zen software and exported to be further processed in MATLAB. No averaging, normalization, or bleaching correction was performed on the imaging data set.

The quantification and graphing of the imaging data were performed using a custom-written MATLAB script. Specifically, the average fluorescence intensity of the 20 frames prior to the stimulation was computed as $F_0$. The change of fluorescence intensity after the stimulation was computed as $(F_t - F_0)/F_0$ ($\Delta F/F$). For each sample, the peak amplitude, defined as the highest value of $\Delta F/F$ within the 80 frames after the stimulation, and the peak time, defined as the time point when peak $\Delta F/F$ is achieved, were computed and used for statistical analyses. Most traces in figures were generated by plotting the average $\Delta F/F$ of individual samples ± standard error of the mean for each frame for the duration of 20 s or 200 frames using a customized MATLAB script. Results presented in Figs. 2a, b, 3c, 6d and Supplementary Fig. 3 are plotted with Microsoft Excel using the raw fluorescence intensity data.

**Behavioral experiments**. Preparation and performance of behavioral experiments was during the day under red light conditions. Larvae were removed from food vials and cleaned with water. For each experiment, 30 early third instar larvae were collected with a fine brush and dark adapted for at least 10 min before the start of the experiments. The larvae were placed in the middle of the testing plate made of a Petri dish (BD Falcon BioDishXL, BD Biosciences) of size 24.5 × 24.5 cm that was filled with 2% agarose (Agarose Standard, Roth). Experiments were performed in a black box illuminated with red LEDs (623 nm, Conrad). A camera (acA2500-14gm, Basler AG, Germany) equipped with a Fujinon lens (Fujinon HF12.5HA-1B 12.5 mm/1.4, Fujifilm, Switzerland) and a red bandpass filter (BP635, Midwest Optical Systems, USA) was placed on top of the arena and recorded the larval behavior for 11 min at the rate of 13 frames/s. The first min of each experiment was not used for the analysis to allow the larvae to adapt to the testing plate.

During the recording period, an ON/OFF light cycle was delivered to the larvae on the testing arena by a light source made of blue and green LEDs (PT-120, Luminus, Billerica, MA, USA). The LED lights illuminated the testing plate from the top at a height of 45 cm. The intensity was 378 μW/cm$^2$, with peaks at 455 nm (11.9 μW/cm$^2$) and 522 nm (3.7 μW/cm$^2$) with half-widths of 9 and 14 nm, respectively. An Arduino running a customized script was used to switch the LEDs off for 1 min and on for 1 min, repeating 5 times per experiment. For image acquisition and larval behavior analyses, customized software developed in LabVIEW and the MAGATAnalyzer were used, respectively[20,57]. MATLAB and R Studio scripts were used for further analysis, statistics, and graphing.

**Behavioral data analyses**. The definitions and thresholds of the behavioral parameters were as described[19,20,23]. A run was defined as an event of forward locomotion with larval head and body aligned. A turn or a pause was defined as an event of slow or no forward locomotion. The speed threshold was determined for each larva individually. An event is marked as turn/pause in cases where larval velocity is slower than the average speed directly before and after a turn/pause. The head and body were aligned during a pause and not aligned during a turn. In other words, turns possess at least one head sweep, whereas pauses do not possess head sweeps. An event is marked as a head sweep in cases where the body bend angle was >20°. A head sweep ends when the body bend angle is again lower than 10°. An accepted head sweep is followed by a run and a rejected head sweep is followed by another head sweep. We calculated the pause frequency per min per animal by determining the number of pauses during a 1 s time window, multiplying this value by 60 and dividing it by the number of larvae present in the field of view of the

camera during the respective time window. The turn frequency per min per animal was calculated in the same way.

**Statistical analysis**. Statistical analyses for optical recordings were performed using the GraphPad Prism. The two-tailed unpaired Student's *t* test was used to compare data in two groups with equal or unequal sample numbers. For data containing multiple groups, one-way analysis of variance (ANOVA) was used with post hoc Tukey's multiple comparison test. Data in figures are presented as mean ± SEM. $p \geq 0.05$ was considered not significant (n.s.): *$p < 0.05$, **$p < 0.01$, and ***$p < 0.001$.

For behavioral experiments, the statistic functions "aov" and "glht(multcomp)" in R Studio were used for statistical analyses. One-way ANOVA followed by Dunnett's multiple comparison test was performed. $p \geq 0.05$ was considered not significant (n.s.), ***$p < 0.001$. Exact *n* values, degrees of freedom, *F* values, *t* values, and *p* values are provided in the figure legends.

**Reporting summary**. Further information on research design is available in the Nature Research Reporting Summary linked to this article.

## Data availability

All data supporting the findings in this study are available from the corresponding author upon reasonable request. The source data underlying Figs. 1d, e, 2d, 3a, b, d, 4e, 5a–d, f, 6a, d, 7c, 8c, e, and Supplementary Figs. 4a, b, 5c and 6a, b are provided as a Source Data file.

## Code availability

Custom-written MATLAB scripts for calcium imaging data analyses are available from the corresponding author upon request.

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

## Acknowledgements

We thank Mark Stopfer and Ralph Nelson for helpful discussions and comments on manuscripts. We also thank C. Desplan, M. Rosbash, and G. Roman for the mutant and transgenic *Drosophila* lines and A. DiAntonio for anti-VGluT antibody. This work was supported by the Intramural Research Programs of NINDS and NIMH, NIH Grant number: NIH/ZIA-NS003137 (to Q.Y.) and NIH/ZIA MH002800 (to B.H.W.), and by the Swiss National Science Foundation (grant number 31003A_169993) (to S.G.S.).

## Author contributions

B.Q. and Q.Y. designed the experiment and performed data collection for optical recordings. T.-H.H. performed the behavioral experiments and quantifications. A.K., H.S.K. and J.S. performed experiments and data analyses. F.D. generated the mAchR-B-Gal4-DBD transgenic fly. B.H.W. and S.G.S. provided advice and supervision. B.Q., A.K. and Q.Y. wrote the manuscript.

## Additional information

**Competing interests:** The authors declare no competing interests.

