## [Peer Review File · Nature Communications]

Editorial Note: An additional Reviewer #3 was consulted during the third round of review.

Reviewers' Comments:

Reviewer #1:

Remarks to the Author:

This manuscript addresses the mechanism of On/Off segregation of visual signals in *Drosophila* larvae. The authors used genetic methods to express calcium and voltage sensors in the glutamatergic and cholinergic interneurons of the larval visual system, and found that these two types of neurons exhibit distinct patterns of responses to On and Off stimuli. They further performed functional imaging in the postsynaptic neurons of these interneurons, and measured light-dependent behaviors in mutant larvae with altered light responses in glutamatergic interneurons. They propose that the On/Off selectivity of *Drosophila* larvae is mediated by the temporal control of inhibitory glu-IOLP activation.

Overall, understanding On/Off selectivity in this simple circuit can add to our knowledge of fundamental principles of sensory processing. The combination of fly genetics, functional imaging and behavior is a state-of-the-art approach to study the functional circuitry in *Drosophila* larvae. The differential responses of glu-IOLPs and cha-IOLPs are very interesting. However, the results in this manuscript are difficult to interpret due to several technical and conceptual issues.

1. The calcium imaging experiments in this study were done with several different methods. The imaging light source is either the 2pt laser (920 nm or 1040 nm), or visible laser (561 nm), while the visual stimuli are also different. The authors need to clarify the rationales of using these different conditions. Does the 561 nm laser used for RCaMP imaging cause photobleaching of photoreceptors? In Figures 4d and 4f, glu-oLOP calcium responses are measured with GCaMP for On responses but with RCaMP for Off responses (e.g. Figures 4d and 4f). Why? How is the kinetics of RCaMP compared to GCaMP6f?

2. It is confusing that the On response of the glu-IOLP showed an initial hyperpolarization/decrease of calcium signal, followed by a prominent calcium increase. But the Arclight signal did not show a biphasic response (e.g. figures 5a and 5c). Please explain why.

3. The spectral sensitivities of glu-, cha-, and pOLPs in Figure 2 are confusing. Significant reduction of 488 nm light-evoked responses in cha-IOLPs and pOLPs in rh6 mutants indicates that Rh6 photoreceptors are strongly activated by 488 nm light. However, the 561 nm responses of the glu-IOLPs in rh6 mutants were eliminated but the 488 nm responses were not affected. This result is hard to understand. It will also be helpful to show spectral sensitivity curves of Rh5 and Rh6 photoreceptors and the emission spectra of the 488 and 561 nm lasers used in this paper.

4. The On response of the glu-IOLP shows a biphasic curve in figures 4 and 5, but not in Figure 3c. Please explain this discrepancy.

5. It will be very helpful if the authors provide a more detailed explanation of their proposed model, and how this model fits the data presented in this manuscript. For example, the On response of the glu-IOLP is altered by PTX in a very specific manner: from a biphasic response to a fast On response similar to that of cha-IOLPs. How does this manipulation impact VPN On/Off selectivity according to the model, and can this prediction be experimentally tested?

Minor points:

Page 4: "Behavioral studies suggest that the spectral differentiation in PRs does not code for color discrimination, but rather generates independent processing for spatial and temporal information during larval navigation." Please clarify this sentence. Do the authors mean that one PR type is responsible for spatial information, and the other type is responsible for temporal information? What do the red arrows represent in Figure 1?

Reviewer #2:

Remarks to the Author:

In the present study, Qin et al study putative ON and OFF pathway neurons in the visual system of the

Drosophila larvae and claim that a temporal control of inhibition generates this selectivity. Using voltage recordings, the authors show that there is a physiological split onto two neurons that are ON and OFF-selective, respectively, and that this is mediated through a metabotropic AChR receptor. If this split is indeed relevant for animal behavior remains unclear, as these neurons converge onto downstream neurons, and behavioral experiments lack specificity. There is no solid evidence for the claim that there is reciprocal interaction between these two neurons / pathways, or that there even is a temporal shift, as I will outline below. While the data suggest some interesting aspects of ON and OFF pathway selectivity in the Drosophila larvae, the authors fail to point these out and concentrate on likely very unspecific aspects of the calcium signal. The study in its current form should not be published.

In my view, the authors are dramatically misinterpreting their data.

The switch in Arlight signal polarity between cha-IOLP and glu-IOLP is convincing (Fig. 3a). This indeed suggest that these neurons might ON and OFF signals respectively.

The problem starts with the interpretation of calcium responses. To me, the RCaMP and GCaMP responses in general reflect the Arlight signal and show an increase in dF/F in cha-IOLP and a smaller initial decrease in glu-IOLP that is not discussed. This decrease likely reflects the hyperpolarization seen when using Arlight. This decrease in fluorescence appears to be specific, and is visible in Figures 3b, 3d, 4f, 5a, 5e, so every time the authors are measuring calcium responses in glu-IOLP. The fact that this decrease in calcium is smaller than the hyperpolarization as measured with Arlight is not surprising, because the calcium indicator can be rectifying. That is especially likely here because baseline (F_0) appears to be very low, and will hardly allow measuring strong decreases in calcium. In the calcium measurements in glu-IOLP, the author solely concentrate on a slow increase in calcium, which they interpret as light decrements activating glu-IOLP. This is happening at time scales that are far from physiological (several seconds after a 100 ms light flash, and after the above mentioned decrease in calcium that has the same timing as the increase in calcium in cha-IOLP). This could just be an unspecific rebound of the calcium signal, or if specific, be triggered by network properties downstream of these neurons, but has nothing to do with the initial light response of these neurons. Furthermore, it is not visible in the response to a prolonged ON stimulus (Figure 3C), further arguing for an actual OFF response associated with the disappearance of the stimulus (even a 100 ms OFF flash has an ON and an OFF component!). All quantifications of the glu-IOLP signal are focusing on this, likely unspecific, signal component, and all claims on temporal differences are based on this.

My interpretation of the data is further corroborated by the findings upon manipulation. For example, the increase in calcium in glu-IOLP upon knockdown Galphao has the same dynamics as the smaller decrease in calcium in the control, arguing that it is really just required to switch response polarity! Similarly (and very obvious) in Figure 5, PTX expression just switches the polarity but leaves the timing of the peak unaltered.

Taken together, despite interesting aspects in the data, there is no evidence for a delay of light-induced responses!

Calcium signals in the somata of monopolar neurons are not a good reflection of the real dynamics within the cell. This is not only known from the literature (see for example Yang et al (2016) Cell 166: 245-57, Figure 5C), but only well visible in the current manuscript: Figure 1e shows how different the temporal dynamics are at the terminal (which is where calcium matters as controls neurotransmitter release) compared to the cell body. This might be okay for statements about response polarity, but to draw conclusions about the temporal dynamics of a somatic calcium response is a big stretch. And even if the authors wanted to argue that this was meaningful, one can clearly see the initial fast decrease in calcium at the terminal.

When looking at the contribution of Rh6-expressing neurons (Figure 2), the description of the results

says that glu-IOLP “remained equally responsive” (figure legend). That is not correct. Glu-IOLP responses are gone when stimulated with 561 nm light (also well visible in their quantification), and the response to 488 nm light triggers an initial decrease in calcium, that is well visible in the individual traces, and this component is also gone in rh6 mutants.

... That is actually fully in line with the idea that glu-IOLPs receive major input from Rh6, that this drives an inhibitory response in glu-IOLP, and that the later response is some overall network effect.

The introduction fails to mention many important aspects of ON and OFF selectivity, and is full of errors. To just give a few examples:

- In a general introduction on ON and OFF selectivity, seminal studies such as the work on the split of ON and OFF channels in the olfactory system of *C. elegans* (Chalasan et al. 2007), or the use of glutamate gated chloride channels in fish should be mentioned
- nicotinic and muscarinic acetylcholine receptors have not at all been implicated in ON and OFF detection in adult flies. Shinomiya et al. (cited as [10]) show that neurons of the OFF pathway (after the ON and OFF split happened!) are cholinergic, and that downstream direction-selective cells express nAChRs and mAChRs respectively. Both Shinomiya et al. and Strother et al. [11] argue that this could produce a time delay, and thus contributes to the extraction of direction-selective and not ON and OFF signals.
- It is unclear if ON and OFF pathways are conserved or convergent structure, and while some papers discuss that they might have a common origin, others do not use the word “conserved” at all (e.g. [6]).

Together, this part of the paper is very unsatisfying.

The authors “conclude that calcium responses in pOLP are driven by the activities in glu-IOLP” (p.15). There is no evidence for that. In flies this can be tested by combining genetic silencing with physiological measurements.

There is no evidence for reciprocal inhibitory interactions between cholinergic and glutamatergic transmission as claimed.

When recording in neurons downstream while expressing PTX in glu-IOLP, the authors interpret a drop in signal as “temporally delayed feedforward glutamatergic inhibition from glu-IOLP”. What argues against the more straightforward idea that glutamate is excitatory and they are just taking away an excitatory input?

The behavioral experiments lack specificity. That manipulation of this “OFF-neuron” not affect light induced behavior? Does cha-IOLP not affect this dark induced behavior? So far, nothing argues against the simple idea that there is no behaviorally relevant ON and OFF pathway split, but the neurons guiding these behaviors are just lacking visual drive.

Summarizing this major criticism, it is interesting that the authors see different response polarities in two neurons downstream of photoreceptors, and their results on mAChRs are compelling. I would encourage the authors to follow up on this, using either Arlight recordings, calcium recordings at the terminal, paying attention to decreases in calcium, or using recently published indicators that are better at capturing decreases in calcium (Zhao et al 2018 Scientific Reports 8). Furthermore, a more ethological visual stimulus might be critical in allowing them to distinguish ON and OFF steps relative to an intermediate contrast.

further comments:

- It is not clear what a light pulse of “~30uW” means for the fly. Light intensities should be calibrated and given in photon / area*time. How do the authors know that what they are using is a

physiologically relevant stimulus?

- it is not mentioned in the text or the figure legend, how long the pulses are that are delivered in figures 1 and 2. The responses do not reflect a typical visual neuron's impulse response, as they last seconds!
- Whenever calcium or voltage signals are shown, they should be shown as dF/F and not in arbitrary fluorescent units. This is not only standard, but also important to distinguish decreases and increases in the response (see for example glu-IOLP responses in Figure 2a,b)
- P.5: The claim that glutamatergic transmission is "generally inhibitory" at Drosophila central synapses, is wrong. Although glutamate can be inhibitory, there are many ionotropic glutamate receptors, that are broadly expressed in the central nervous system, and pass excitatory signals (that would also explain there findings in Figure 6)
- if the Gal4 lines are used for behavior, it needs to be shown how specific the Gal4 line outside of the larval optic neuropil.
- In Figure 2, the pOLP response is happening $\sim 5s$ after the visual stimulus in the individual trace, but $\sim 2s$ after the stimulus in the averages.
- The small circuit schematic in the imaging figures is very confusing. The green neuron appears to be Rh6-PRs. Then why do they talk to the terminal, and why are the cell bodies at the other end of the cell?
- It should probably be mentioned that the mAChR is not only expressed in glu-IOLP but also in another neuron in Figure 4C

Point-by-point response to reviewers' comments

Manuscript: NCOMMS-18-25662B; "Temporal control of inhibition *via* muscarinic acetylcholine receptor signaling generates ON and OFF selectivity in a simple visual circuit" by Qin et al.

We greatly appreciate the reviewers' careful evaluations of our work. The reviewers found the questions we are studying important and our findings interesting while raising valid points on technical and conceptual issues. The reviewers did not recommend specific experiments but offered constructive suggestions for us to modify the writing and clarify our conclusions. In this revision, we addressed technical issues by providing additional data and modified the text to improve the clarity and accuracy of our statements. In addition, to resolve the conceptual issues raised by reviewers and emphasize the main finding of this study, we adjusted the structure of the manuscript through extensive revisions of the main text and reorganizations the figures. We believe that we have addressed all reviewers' comments and our manuscript has been significantly improved. We hope that the editor and reviewers find this revised version now suitable for publication in Nature Communications. The following is a summary of major changes made in this revision followed by our point-by-point response to reviewers' comments.

Major changes included in the revised manuscript:

1. We performed RCaMP recordings of the ON and OFF responses in IOLPs, to address the questions raised by Reviewer1. The results are included in Fig. 4d-e and described on P13. The data presented previously in Fig. 4d-e is moved to Fig. S5b-c and Fig. S6a.
2. We performed knock-down experiment using a new RNAi line targeting mAChR-B to strengthen our conclusions and address the questions raised by Reviewer 2. The results are included in Fig. S6 and described on P15.
3. We provided quantifications for the trough and the peak of the light-induced biphasic calcium response in glu-IOLP in Fig. S4, as suggested by Reviewer 2.
4. We included schematic illustrations of the protocols we used for visual stimulation and calcium imaging in Fig. S3 and S5, as suggested by Reviewer 1.
5. We included comparison between GCaMP vs. RCaMP and RCaMP recorded using two-photon laser vs. confocal laser in Fig. S8 to address technical points raised by Reviewer 1.
6. To address the conceptual issues raised by the reviewers and to emphasize our main findings, we reorganized data presentation in Fig. 4-7 and removed results generated using the glutamate sensor. We also modified the title and edited the main text following the suggestions from the reviewers.

Reviewer #1 (Remarks to the Author):

This manuscript addresses the mechanism of On/Off segregation of visual signals in Drosophila larvae. The authors used genetic methods to express calcium and voltage sensors in the glutamatergic and cholinergic interneurons of the larval visual system, and found that these two types of neurons exhibit distinct patterns of responses to On and Off stimuli. They further performed functional imaging in the postsynaptic neurons of these interneurons, and measured light-dependent behaviors in mutant larvae with altered light responses in glutamatergic interneurons. They propose that the On/Off selectivity of Drosophila larvae is mediated by the temporal control of inhibitory glu-IOLP activation. Overall, understanding On/Off selectivity in this simple circuit can add to our knowledge of fundamental principles of sensory processing. The combination of fly genetics, functional imaging and behavior is a state-of-the-art approach to study the functional circuitry in Drosophila larvae. The differential responses of glu-IOLPs and cha-IOLPs are very interesting. However, the results in this manuscript are difficult to interpret due to several technical and conceptual issues.

Response:

We greatly appreciate the reviewer's careful evaluation of our work and positive comments on our findings. The reviewer raised valid points related to technical and conceptual issues, which are particularly helpful in guiding our revision. By modifying the text and our data presentation, as well as providing new data, we believe we address all issues raised by Reviewer #1. Please see below for the detailed description of changes we made to address specific comments.

1. The calcium imaging experiments in this study were done with several different methods. The imaging light source is either the 2pt laser (920 nm or 1040 nm), or visible laser (561 nm), while the visual stimuli are also different. The authors need to clarify the rationales of using these different conditions.

Response:

We agree with the reviewer about the need for additional justifications for our experimental paradigms and approaches. In the revision, we elaborate upon the rationale in the main text and provide justifications for using specific indicators and light sources in each type of data collection. We included the following statements in the main text on **Page 7, 10 and 11**.

Our previous studies established a protocol to record light-elicited physiological responses in the target neurons of PRs using larval eye-brain explants, in which the Bolwig's organ, the larval light sensing organ that contains the PRs, the optic nerve and the brain lobe are kept intact¹. This approach allows us to deliver temporally controlled light simulations using either the 488 or 561 nm laser while detecting calcium transients with cell-specific expression of GCaMP6f through two-photon imaging (Fig. S3a)². In addition, to test for the compartmentalization of light-evoked calcium responses, we recorded GCaMP signals in both soma and terminal regions (Fig. 1d).

Additionally, this set of experiments validated our imaging protocols and indicated that, at the intensities we tested, both blue (488 nm) and green (561 nm) light stimulations elicit robust calcium responses in the OLPs (Fig. 2a, c, d). Therefore, light stimulations

delivered at either wavelength can be used to study light-evoked responses in the IOLPs and downstream VPNs.

To measure light-induced calcium and voltage responses within the same IOLP neurons, we examined the change in membrane potential using the genetically-encoded voltage sensor Arlight while recording the calcium transient with the red calcium indicator RCaMP^{3,4}. By matching calcium profiles with voltage changes, we found that light pulses induce depolarization and fast calcium transients in cha-IOLP but hyperpolarization and slow calcium transients in glu-IOLP (Fig. 3a, b). We compared the RCaMP recordings with the previous GCaMP recordings and found that although the amplitude of calcium transients reported by RCaMP recordings was reduced compared to the results obtained by GCaMP recordings, the waveforms of IOLPs' calcium responses remain the same (Fig. 3b, Fig. S8).

We next tested how the IOLPs respond to light increments and decrements by monitoring calcium responses during onsets and offsets of extended light exposures. Although the two-photon recordings of GCaMP6f provide the best image quality, it is incompatible with the extended light exposure due to the light sensitivity of the detector. Therefore, in the following experiments, we used RCaMP as the calcium indicator, which can be imaged using a confocal laser tuned to 561 nm with low intensity, reducing the effects of photobleaching on both the calcium sensor and the photoreceptors. Additionally, this protocol allowed us to alter the light cycles or deliver a dark pulse by tuning the 488 nm laser during imaging sessions (Fig. S5a).

In addition, in the supplementary data, we include schematic diagrams to illustrate the stimulation and recording paradigms for different experiments (**Fig. S3, S5a**) and provide additional data comparing results obtained from GCaMP6f and RCaMP using different light sources (**Fig. S8**). Briefly, to optimize the image quality and reduce phototoxicity, we used the two-photon laser tuned to 920 nm for GCaMP and Arlight recordings. The light stimulations were delivered by the confocal laser tuned to 561 nm for visual stimulation in most of the experiments (**Fig. 1, 2, 3a, 5, 6, Fig. S3b, S6, S8**). The exceptions are: 1. In Figure 3b, we recorded the RCaMP using 2photon laser tuned to 1040 nm, the optimal wavelength for RCaMP demonstrated by previous studies⁴. 2. For Figure 3c, d and 4d, we recorded RCaMP using 561 nm confocal laser to detect the OFF response. Detailed descriptions of the methods we used for visual stimulation and image collections are included in the Methods.

It is worth noting that a large part of the optic recording setup for the larval visual system is established in this study and described in this manuscript. We used a commercial system, Zeiss LSM780 equipped with a Coherent two-photon laser for image collection and light stimulation. Although this setup offers limited flexibility, our protocols for imaging and light stimulation are easily replicable using the parameters provided in the Methods. Furthermore, we acknowledge technical limitations in the Discussion (**P21**). We hope the reviewer finds the additional justifications in the main text and the new data included in the supplementary figures sufficient to address this issue.

2. Does the 561 nm laser used for RCaMP imaging cause photobleaching of photoreceptors? In Figures 4d and 4f, glu-oLOP calcium responses are measured with GCaMP for On responses but with RCaMP for Off responses (e.g. Figures 4d and 4f). Why? How is the kinetics of RCaMP compared to GCaMP6f?

Response:

We thank the reviewer for raising the valid point. As mentioned in our response to the reviewer's comment #1, we used RCaMP to test the OFF response that requires extended light exposure, which is incompatible with our two-photon setup. To address the reviewer's concern, we provided new data from a set of experiments measuring the ON and OFF responses in parallel using RCaMP (**Fig. 4d, e**). The conclusion remains the same; cha-IOLP only shows ON response, and glu-IOLP displays both ON and OFF response with different temporal profiles. The description of this data set is included in the main text on **P13**.

Next, to examine mAChR-B's function in mediating glu-IOLP's physiological responses, we performed knock-down experiments using a transgenic RNAi line targeting mAChR-B and recorded the IOLPs' response to 100 ms light vs. dark pulses using RCaMP. Consistent with our earlier observations in wild-type controls, cha-IOLP only responded to the light pulse and generated a fast calcium transient, whereas glu-IOLP responded to both light and dark pulses with either a slow or a fast calcium transient. Strikingly, mAChR-B knockdown eliminated both the light and dark pulse-induced calcium transients in glu-IOLP, indicating mAChR-B as the receptor mediating the light-induced inhibition of glu-IOLP (Fig. 4d, e).

As the reviewer pointed out, photobleaching can be observed in RCaMP recordings using the 561 nm laser. To provide justification, we included the following statement in the main text on **P11**:

Although the two-photon recordings of GCaMP6f provide the best image quality, it is incompatible with the extended light exposure due to the light sensitivity of the detector. Therefore, in the following experiments, we used RCaMP as the calcium indicator, which can be imaged using a confocal laser tuned to 561 nm with low intensity, reducing the effects of photobleaching on both the calcium sensor and the photoreceptors. Additionally, this protocol allowed us to alter the light cycles or deliver a dark pulse by tuning the 488 nm laser during imaging sessions (Fig. S5a).

To specifically address the reviewer's question about the kinetics of RCaMP compared to GCaMP, in the new **Fig. S8**, we included comparisons between light-induced calcium transients reported by GCaMP vs. RCaMP, as well as the RCaMP transient imaged using the two-photon laser vs. confocal laser. With the same stimulation protocol, there are clear differences in the amplitude of the light-induced calcium responses. However, the waveform and the temporal profiles of the calcium transients are similar. We also included this statement in the main text on **P10**.

We compared the RCaMP recordings with the previous GCaMP recordings and found that although the amplitude of calcium transients reported by RCaMP recordings was reduced compared to the results obtained by GCaMP recordings, the waveforms of IOLPs' calcium responses remain the same (Fig. 3b, Fig. S8).

3. It is confusing that the On response of the glu-IOLP showed an initial hyperpolarization/decrease of calcium signal, followed by a prominent calcium increase. But the Arclight signal did not show a biphasic response (.e.g figures 5a and 5c). Please explain why.

Response:

We thank the reviewer for raising this valid point. We indeed observed a biphasic response of the Arclight in glu-IOLP, shown in **Fig. 5e** and quantified in **Fig. 5f**. We did not describe this result adequately in the previous version of the manuscript. In this revision, we included a statement related to this observation on **P15**:

Additionally, we performed voltage recordings using Arclight and revealed a dramatic change in glu-IOLP's voltage responses caused by PTX expression. In the control group, we observed a biphasic voltage response in glu-IOLP induced by light stimulation, which produced a large hyperpolarization event followed by a small depolarization (Fig. 5e, f). This response is temporally correlated with the biphasic calcium transients that we observed previously which showed a small reduction in calcium concentration followed by a slow rise (Fig. 5c, d).

4. The spectral sensitivities of glu-, cha-, and pOLPs in Figure 2 are confusing. Significant reduction of 488 nm light-evoked responses in cha-IOLPs and pOLPs in rh6 mutants indicates that Rh6 photoreceptors are strongly activated by 488 nm light. However, the 561 nm responses of the glu-IOLPs in rh6 mutants were eliminated but the 488 nm responses were not affected. This result is hard to understand. It will also be helpful to show spectral sensitivity curves of Rh5 and Rh6 photoreceptors and the emission spectra of the 488 and 561 nm lasers used in this paper.

Response:

We appreciate the reviewer's suggestions. The results shown in **Fig. 2** provide critical evidence for IOLPs receiving Rh6-PR inputs and validate our data collection and stimulation protocols. We agree with the reviewer that it is important to provide the spectral sensitivity information on Rh5 and 6 and the emission spectra of the 488 and 561 nm lasers we used in the stimulation. The descriptions are included in the Methods (**Page 35-36**). To improve the clarity, we modified the description in the main text and included the following statement (**P8, 9**)

To establish the functional connectivity between subtypes of PRs and OLPs, we performed calcium imaging experiments with light stimulations delivered at 488 nm (blue) and 561 nm (green) wavelengths (Fig. S3). Previous studies indicated that Rh6 detects light within the 400-600 nm range and its maximal spectral sensitivity is ~437 nm, while Rh5 detects light from 350-500 nm and its maximal spectral sensitivity is ~508 nm. Therefore, Rh6-PRs are sensitive to light stimulations at both 488 nm and 561 nm, whereas Rh5-PRs only respond to blue light at 488 nm⁵. These features, in combination with a loss-of-function Rh6 mutant (rh6⁻)⁶, allowed us to examine the specific contributions of Rh5- and Rh6-PRs to the OLPs' light responses.

In wildtype larvae, 488 and 561 nm light stimulations elicit almost identical responses from the OLPs (Fig. 2a, c, d). In contrast, all responses to green light (561 nm) were eliminated in the Rh6 mutant (*rh6⁻*), demonstrating that green light-evoked responses in OLPs are solely generated by visual transduction in Rh6-PRs (Fig. 2b- d). To test the functional connectivity between Rh5-PRs and OLPs, we performed experiments using blue light (488nm) stimulations in the Rh6 mutant background, where blue light-elicited responses are exclusively generated by Rh5-PRs. Interestingly, compared to the wildtype control, blue light-induced calcium responses in *cha*-IOLP and *p*OLP were significantly reduced in Rh6 mutants, suggesting that Rh6-PR is the primary mediator of light inputs into these two neurons. On the other hand, the Rh6 mutation produced no significant difference in *glu*-IOLP's blue light response (Fig. 2b- d), indicating that *glu*-IOLP receives both Rh5- and Rh6-PR inputs.

Below (right) are the original data from the reference and the screen shots (left) displaying the laser emission spectra on our setup:

Emission spectra of confocal lasers on LSM780

Published spectra sensitivity of Rh5 and Rh6 (Salcedo et al., 1999), Fig. 3

5. The On response of the *glu*-IOLP shows a biphasic curve in figures 4 and 5, but not in Figure 3c. Please explain this discrepancy.

Response:

We thank the reviewer for raising the valid point. There is a clear difference in the waveforms produced by short light exposure vs. long light exposure. We believe this is due to the sustained inhibition generated by mAChR-B signaling during the extended light exposure. We modified the main text and provided the following statement on **P11**:

*RCaMP recordings showed that, with an extended light exposure, *cha*-IOLP only responded to the light onset with a fast calcium transient, demonstrating its specific*

response to light increments. In contrast, the light onset induced a small yet noticeable reduction of calcium signal in *glu-IOLP*, while the light offset produced an immediate calcium rise, suggesting that *glu-IOLP* is activated by the light decrements (Fig. 3c). This observation is consistent with light hyperpolarizing *glu-IOLP* at the onset of the light exposure and generating a sustained inhibition until the offset of the light exposure.

6. It will be very helpful if the authors provide a more detailed explanation of their proposed model, and how this model fits the data presented in this manuscript. For example, the On response of the *glu-IOLP* is altered by PTX in a very specific manner: from a biphasic response to a fast On response similar to that of *cha-IOLPs*. How does this manipulation impact VPN On/Off selectivity according to the model, and can this prediction be experimentally tested?

Response:

We greatly appreciate the reviewer's suggestions. In this revision, we provided the following statement to describe our proposed model (**P19-20**). In addition, we tested this model using behavioral analyses, which indicate that *glu-IOLP* is essential for dark-induced behavior responses in larvae and thus validated our model (**Fig. 7b-d**).

*To illustrate the potential roles for IOLPs in transmitting ON and OFF signals from the PRs to the VPNs and based on our findings and the connectivity map, we propose a model with three components. First, the pair of IOLPs exhibit distinct responses to light increments and decrements, thus acting as ON and OFF detectors. The sign-inversion required for OFF detection in *glu-IOLP* is mediated by the mAChR-B receptor. Second, while *cha-IOLP* displays clear ON selectivity, *glu-IOLP* has both ON and OFF responses. The OFF selectivity in *glu-IOLP* emerges from the temporal control of its activity by mAChR-B/Gαo signaling. Third, extending our findings in the LNvs and pOLP to the rest of the VPNs, we propose that, although downstream VPNs receive both cholinergic and glutamatergic inputs, there are specific groups of ON vs. OFF responsive VPNs that are functionally separated by their molecular compositions. ON-responsive VPNs (ON-VPNs), such as LNvs, are activated by cholinergic signaling and inhibited by glutamatergic signaling, while OFF-responsive VPNs (OFF-VPNs), such as pOLP, behave the opposite way. Although additional physiological studies on other VPNs are needed to validate this model, given the lack of anatomical segregation of ON and OFF pathways, this functional separation of VPNs is a plausible solution to ensure the preservation of the ON and OFF signals at the level of VPNs (Fig. 7a).*

*Predicated by this model, an ON response is dominated by cholinergic transmissions from cholinergic PRs and *cha-IOLP*, while the extended inhibition of *glu-IOLP* via mAChR-B/Gαo signaling ensures the activation of only the ON-VPNs. During an OFF response, with no cholinergic input, the *glu-IOLP* is solely responsible for activating OFF-VPNs. In terms of their roles in regulating behavioral output, *cha-IOLP* likely functions in modulating the strength and duration of the light-induced response, and *glu-IOLP* would be essential for initiating the dark-induced behavioral response (Fig. 7a).*

*.. Although further experiments are needed to address the behavioral relevance of *cha-* and *glu-IOLP* in regulating other visually-guided behaviors, our studies measuring dark-*

induced pausing behavior indicate that glu-IOLP has an essential role in mediating OFF detection in the larval visual circuit, consistent with the model developed based on the anatomical and physiological studies.

In addition, we also agree with the reviewer that additional clarifications on the effect of PTX expression will be helpful. To clearly demonstrate and describe the effect on the downstream VPNs produced by this manipulation, we reorganized **Fig. 6** and amended our description of the results in the main text (**P16, 18**), including the following statements:

...we found that PTX expression in glu-IOLP eliminates its OFF response while modifying the temporal profile of its ON response and thus effectively transforms glu-IOLP into an ON selective cell. Instead of transmitting light decrements, glu-IOLP expressing PTX transmits light increments to downstream VPNs and potentially disrupts the separation of the ON and OFF channels.

Together, our results show that altering the temporal kinetics of a single neuron, glu-IOLP, strongly influences light responses in both visual interneurons and projection neurons, supporting the functional significance of the temporal control of glutamatergic transmission in the larval visual circuit. In addition, our studies also validated reciprocal interactions between cha- and glu-IOLPs and demonstrated the ability of glu-IOLP to elicit distinct physiological responses in different types of VPNs.

Minor points:

Page 4: “Behavioral studies suggest that the spectral differentiation in PRs does not code for color discrimination, but rather generates independent processing for spatial and temporal information during larval navigation.” Please clarify this sentence. Do the authors mean that one PR type is responsible for spatial information, and the other type is responsible for temporal information?

What do the red arrows represent in Figure 1?

Response:

We thank the reviewer for pointing out the ambiguity in our descriptions. We amended the statement in the Introduction to clarify this statement (**P4**).

Behavioral studies suggest that the spectral differentiation in PRs does not code for color discrimination, but rather generates independent processing for spatial and temporal information during larval navigation. Specifically, while Rh5-PRs are essential for navigation based on spatial cues, Rh6-PRs are involved in perceiving temporal changes of light intensity during larval head casts and contribute to effective light avoidance behaviors^{7,9}.

The red arrow in the legend of **Fig. 1** was used to indicate the excitatory inputs from Rh5- and Rh6-PRs onto LNvs. However, since the effects of light inputs on other VPN are unclear, we changed the color to grey, indicating the anatomical connections between PRs and VPNs.

Reviewer #2 (Remarks to the Author):

In the present study, Qin et al study putative ON and OFF pathway neurons in the visual system of the Drosophila larvae and claim that a temporal control of inhibition generates this selectivity. Using voltage recordings, the authors show that there is a physiological split onto two neurons that are ON and OFF-selective, respectively, and that this is mediated through a metabotropic AChR receptor. If this split is indeed relevant for animal behavior remains unclear, as these neurons converge onto downstream neurons, and behavioral experiments lack specificity. There is no solid evidence for the claim that there is reciprocal interaction between these two neurons / pathways, or that there even is a temporal shift, as I will outline below. While the data suggest some interesting aspects of ON and OFF pathway selectivity in the Drosophila larvae, the authors fail to point these out and concentrate on likely very unspecific aspects of the calcium signal. The study in its current form should not be published.

Response:

We thank the reviewer for evaluating our work. The reviewer recognized our efforts in establishing the distinct ON and OFF responses within the two interneurons of the larval visual circuit and identifying mAChR-B as the receptor that mediates the light-induced inhibition in glu-IOLP. On the other hand, the reviewer disagreed with two of our main conclusions. Specifically, the reviewer labeled the slow calcium transient in glu-IOLP we described as “likely very unspecific aspects of the calcium signal”. The reviewer also indicated “There is no solid evidence for the claim that there is reciprocal interaction between these two neurons / pathways, or that there even is a temporal shift”. These views are at the center of most of the reviewer’s negative comments. Here, we would like to direct the reviewer’s attention to multiple pieces of experimental evidence we presented that support these conclusions. We will discuss published studies on mammalian and adult *Drosophila* visual interneurons, which we used as references for both experimental design and data interpretation. In addition, following the questions and suggestions from both reviewers, we modified our writings extensively and reorganized our data presentation to clarify the statements on our findings and proposed models. Please see below for the detailed description of changes we made to address specific comments.

1. In my view, the authors are dramatically misinterpreting their data. The switch in Arclight signal polarity between cha-IOLP and glu-IOLP is convincing (Fig. 3a). This indeed suggest that these neurons might ON and OFF signals respectively. The problem starts with the interpretation of calcium responses. To me, the RCaMP and GCaMP responses in general reflect the Arclight signal and show an increase in dF/F in cha-IOLP and a smaller initial decrease in glu-IOLP that is not discussed. This decrease likely reflects the hyperpolarization seen when using Arclight. This decrease in fluorescence appears to be specific, and is visible in Figures 3b, 3d, 4f, 5a, 5e, so every time the authors are measuring calcium responses in glu-IOLP.

Response:

We thank the reviewer for the positive comments on the Arclight experiments. Regarding the light-induced decrease of calcium signals induced in glu-IOLP, although we agree that we did not describe this observation adequately, we recognized the decrease as the result of light

induced inhibition and conducted the Arclight experiment to confirm this result. In this revision, we modified the description of the light-induced calcium response in the main text (**Page 7**):

Calcium imaging using IOLP^{sh}-Gal4 and IOLP-Gal4 revealed distinct light-elicited physiological responses in the two local OLPs. Upon light stimulation, a 100 ms light pulse delivered by the 561 nm laser, glu-IOLP exhibited a small reduction in the GCaMP signal followed by a slow calcium rise, whereas cha-IOLP responded to light with a large and immediate calcium rise (Fig. 1d, e). Calcium transients obtained from the terminal region of glu-IOLP displayed the same biphasic waveform as those in the somas, but with higher amplitudes and shorter latencies (Fig. 1d). In addition, terminal recordings of both IOLPs produced calcium transients with two distinct peaks that clearly reflect the temporal differences in the light-induced responses of the two IOLPs (Fig. 1e).

2. The fact that this decrease in calcium is smaller than the hyperpolarization as measured with Arclight is not surprising, because the calcium indicator can be rectifying. That is especially likely here because baseline (F0) appears to be very low, and will hardly allow measuring strong decreases in calcium.

Response:

As the reviewer pointed out, the initial decrease in calcium is small and it is likely due to the rectification of the intracellular calcium. However, in this case rectification is not caused by the GCaMP6f indicator. Our observations in IOLPs resembled findings in previous studies of adult visual interneurons which showed strong half-wave rectification in the calcium responses¹⁰. We included a statement regarding this observation on **P16**.

In addition, upon examining the voltage and calcium responses in the controls and PTX expression glu-IOLPs, we found that the intracellular calcium in glu-IOLP appears to show rectification that favors depolarization events and generates large increases in calcium concentration in contrast to hyperpolarization events that produce small reductions in the calcium signals. The half-wave rectification of intracellular calcium is also observed in adult visual interneurons in the ON and OFF pathways, where the voltage to calcium transformation generates sign inversion or preservation, which, together with the rectification, lead to the ON selectivity in Tm3 and Mi1 and OFF selectivity in Tm1 and Tm2^{13,29}.

To specifically address the reviewer's comments on the baseline (F0) level of the calcium indicator, our recording results are mostly presented as $\Delta F/F$, and thus the calcium transient start at a level near 0. The baseline of the original GCaMP signal in the glu-IOLP is, in fact, high, as shown in representative raw traces and raw images presented in **Fig. 2a** and **Fig. S3**.

3. In the calcium measurements in glu-IOLP, the author solely concentrate on a slow increase in calcium, which they interpret as light decrements activating glu-IOLP. This is happening at time scales that are far from physiological (several seconds after a 100 ms light flash, and after the above mentioned decrease in calcium that has the same timing as the increase in calcium in cha-IOLP). This could just be an unspecific rebound of the calcium signal, or if specific, be triggered

by network properties downstream of these neurons, but has nothing to do with the initial light response of these neurons. Furthermore, it is not visible in the response to a prolonged ON stimulus (Figure 3C), further arguing for an actual OFF response associated with the disappearance of the stimulus (even a 100 ms OFF flash has an ON and an OFF component!). All quantifications of the glu-IOLP signal are focusing on this, likely unspecific, signal component, and all claims on temporal differences are based on this.

Response:

The reviewer regarded the slow ON response in glu-IOLP as non-specific. We will present evidence and references to specifically address different aspects of the reviewer's comments:

This is happening at **time scales** that are far from physiological (several seconds after a 100 ms light flash, and after the above mentioned decrease in calcium that has the same timing as the increase in calcium in cha-IOLP). This could just be an unspecific rebound of the calcium signal, or if specific, be triggered by network properties downstream of these neurons, but has nothing to do with the initial light response of these neurons.

This slow calcium rise induced by light pulses is prominent in all glu-IOLP recordings, especially in the terminal regions (**Fig. 1-6**). This significant temporal delay can be measured using different indicators and paradigms consistently and can also be modified genetically (**Fig. 4d, e, 5a-f, Fig. S6**). In addition, changing the temporal profile of the glu-IOLP's ON response strongly impacts both downstream VPNs' light responses (**Fig. 6**) and dark induced behaviors (**Fig. 7**). Therefore, we believe we have strong evidence to support our claims. We acknowledge that the temporal scale of this ON response is surprisingly slow. However, it is part of the physiological property of glu-IOLP and appear to be consistent with the temporal scale of the dark-induced pausing behavior (**P20, Fig. 7b, c**).

Besides our own experimental evidence, the long-latency (1-2 sec) ON response in the OFF pathway has also been observed and investigated in mammalian retinae by electrophysiology¹¹. Studies indicate that this long-latency ON response is generated within the OFF pathway and is suppressed by the ON pathway, similar as the slow ON response we observed in glu-IOLP. Although there are clear differences in circuit architecture, intrinsic properties of neurons and experimental paradigms, similar findings in mammalian visual system indicate that the temporal scale should not be the reason to consider the slow ON response non-specific.

Furthermore, it is not visible in the response to a prolonged ON stimulus (Figure 3C), further arguing for an actual OFF response associated with the disappearance of the stimulus (even a 100 ms OFF flash has an ON and an OFF component!). All quantifications of the glu-IOLP signal are focusing on this, likely unspecific, signal component, and all claims on temporal differences are based on this.

We also recognize the differences in the glu-IOLP's response to a short light pulse and extended light exposure. This is in fact the difference between the ON and OFF response in glu-IOLP. We included the following statement to describe the differential responses (**P11-12**):

RCaMP recordings showed that, with an extended light exposure, cha-IOLP only responded to the light onset with a fast calcium transient, demonstrating its specific response to light increments. In contrast, the light onset induced a small yet noticeable reduction of calcium signal in glu-IOLP, while the light offset produced an immediate calcium rise, suggesting that glu-IOLP is activated by the light decrements (Fig. 3c). This observation is consistent with light hyperpolarizing glu-IOLP at the onset of the light exposure and generating a sustained inhibition until the offset of the light exposure.

We performed additional experiments to examine the differential responses of glu-IOLP toward light increments and decrements by subjecting the preparation to 100 ms light or dark pulses that are contrast matched. We found that a 100 ms dark pulse, or a brief reduction in light intensity following an extended light exposure, is sufficient to generate an immediate calcium rise in glu-IOLPs. Compared to the slow calcium responses induced by 100 ms light pulses, this dark-induced OFF response has a similar amplitude, but a significantly shorter latency (Fig. 3d). Similar recordings indicate that cha-IOLP does not respond to dark pulses and only generates the fast ON response to light pulses.

In short, our results indicated that glu-IOLP has both a slow ON response induced by light increments and an immediate OFF response induced by light decrements. These differential responses towards light increments and decrements indicate that glu-IOLP is responding to both ON and OFF signals and the temporal regulation of its activation is important for its OFF selectivity. This was evaluated by additional experiments and supported by evidence presented in **Fig. 4, 5** and **6**. In addition, light-induced calcium responses in IOLPs showed similar waveforms as the ones observed in the adult visual interneurons in the ON and OFF pathways. In this case, Tm3 neuron displays the ON selectivity and Tm1 displays OFF selectivity.

Figure 6, Yang, et al, 2016

4. interpretation of the data is further corroborated by the findings upon manipulation. For example, the increase in calcium in glu-IOLP upon knockdown Galphao has the same dynamics as the smaller decrease in calcium in the control, arguing that it is really just required to switch response polarity! Similarly (and very obvious) in Figure 5, PTX expression just switches the polarity but leaves the timing of the peak unaltered. Taken together, despite interesting aspects in the data, there is no evidence for a delay of light-induced responses!

Response:

We thank the reviewer for recognizing the specific phenotype produced by knocking down $G\alpha$ in glu-IOLP. However, the reviewer suggested that the PTX and $G\alpha$ knockdown manipulations “switch the response polarity” and disregard the changes in the calcium response. If we understand correctly, the reviewer is describing the voltage events, which changed from hyperpolarization to depolarization with the PTX expression and was fully acknowledged in our manuscript. We would like to point out that these manipulations undoubtedly modified the timing of light induced calcium response in glu-IOLP, and we believe our description of the phenotype is appropriate.

To improve the clarity of our data presentation, we modified **Fig. 6** and provided additional evidence in **Fig. S6**. We hope these modifications in both the text and figures will help convince the reviewer. Below is the summary of the section describing this set of experiments (**P16**).

In summary, our genetic studies confirm the role of mAChR-B/Gao signaling in mediating light-induced inhibition of glu-IOLP and reveal the complexity of glu-IOLP's light responses, which appear to contain multiple signaling events that cooperatively regulate the direction and timing of the neuron's physiological output. Although additional studies are needed to fully characterize these responses, we found that PTX expression in glu-IOLP eliminates its OFF response while modifying the temporal profile of its ON response and thus effectively transforms glu-IOLP into an ON selective cell. Instead of transmitting light decrements, glu-IOLP expressing PTX transmits light increments to downstream VPNS and potentially disrupts the separation of the ON and OFF channels

5. Calcium signals in the somata of monopolar neurons are not a good reflection of the real dynamics within the cell. This is not only known from the literature (see for example Yang et al (2016) Cell 166: 245-57, Figure 5C), but only well visible in the current manuscript: Figure 1e shows how different the temporal dynamics are at the terminal (which is where calcium matters as controls neurotransmitter release) compared to the cell body. This might be okay for statements about response polarity, but to draw conclusions about the temporal dynamics of a somatic calcium response is a big stretch. And even if the authors wanted to argue that this was meaningful, one can clearly see the initial fast decrease in calcium at the terminal.

Response:

We appreciate the reference provided by the reviewer, as we used the same reference in our studies. For the same reason as the reviewer mentioned in the comments, we collected data from both the soma and terminal region for all the glu-IOLP experiments and some of the IOLP experiments. The data are presented in **Fig. 1, 5, and S6**. In fact, the slow ON response in the terminal region is more prominent than the ones in the soma and has a significant delay as well. For the experiments examining two or three OLPs simultaneously, calcium signals collected from the terminal regions cannot be accurately quantified due to being mixed and thus were not included.

6. When looking at the contribution of Rh6-expressing neurons (Figure 2), the description of the results says that glu-IOLP “remained equally responsive” (figure legend). That is not correct. Glu-IOLP responses are gone when stimulated with 561 nm light (also well visible in their

quantification), and the response to 488 nm light triggers an initial decrease in calcium, that is well visible in the individual traces, and this component is also gone in rh6 mutants.
... That is actually fully in line with the idea that glu-IOLPs receive major input from Rh6, that this drives an inhibitory response in glu-IOLP, and that the later response is some overall network effect.

Response:

There appears to be some misunderstanding of our data. The results shown in **Fig. 2** provide critical evidence for IOLPs receiving Rh6-PR inputs and validate our data collection and stimulation protocol. We modified the description of this set of data in the revision to improve clarity (**P8-9**).

To establish the functional connectivity between subtypes of PRs and OLPs, we performed calcium imaging experiments with light stimulations delivered at 488 nm (blue) and 561 nm (green) wavelengths (Fig. S3). Previous studies indicated that Rh6 detects light within the 400-600 nm range and its maximal spectral sensitivity is ~437 nm, while Rh5 detects light from 350-500 nm and its maximal spectral sensitivity is ~508 nm. Therefore, Rh6-PRs are sensitive to light stimulations at both 488 nm and 561 nm, whereas Rh5-PRs only respond to blue light at 488 nm⁵. These features, in combination with a loss-of-function Rh6 mutant (rh6⁻)⁶, allowed us to examine the specific contributions of Rh5- and Rh6-PRs to the OLPs' light responses.

In wildtype larvae, 488 and 561 nm light stimulations elicit almost identical responses from the OLPs (Fig. 2a, c, d). In contrast, all responses to green light (561 nm) were eliminated in the Rh6 mutant (rh6⁻), demonstrating that green light-evoked responses in OLPs are solely generated by visual transduction in Rh6-PRs (Fig. 2b-d). To test the functional connectivity between Rh5-PRs and OLPs, we performed experiments using blue light (488nm) stimulations in the Rh6 mutant background, where blue light-elicited responses are exclusively generated by Rh5-PRs. Interestingly, compared to the wildtype control, blue light-induced calcium responses in cha-IOLP and pOLP were significantly reduced in Rh6 mutants, suggesting that Rh6-PR is the primary mediator of light inputs into these two neurons. On the other hand, the Rh6 mutation produced no significant difference in glu-IOLP's blue light response (Fig. 2b-d), indicating that glu-IOLP receives both Rh5- and Rh6-PR inputs.

7.The introduction fails to mention many important aspects of ON and OFF selectivity, and is full of errors. To just give a few examples:

Response:

Following the reviewer's suggestion, we modified the introduction to focus on the organization of the larval visual circuit and the potential functions of the IOLPs (**P3-5**).

- In a general introduction on ON and OFF selectivity, seminal studies such as the work on the split of ON and OFF channels in the olfactory system of *C. elegans* (Chalasanani et al. 2007), or the use of glutamate gated chloride channels in fish should be mentioned

We thank the reviewer for providing an additional reference regarding the ON and OFF selectivity in the olfactory system of *C. elegans*. However, our introduction focused on the studies related to ON and OFF pathways in vertebrate retina and adult *Drosophila* visual system, which we believe is appropriate given the topic of this study. In addition, molecular studies on the role of the glutamate receptor signaling in ON and OFF selectivity and the work on glutamate gated chloride channels are also cited in the manuscript.

- nicotinic and muscarinic acetylcholine receptors have not at all been implicated in ON and OFF detection in adult flies. Shinomiya et al. (cited as [10]) show that neurons of the OFF pathway (after the ON and OFF split happened!) are cholinergic, and that downstream direction-selective cell express nAChRs and mAChRs respectively. Both Shinomiya et al. and Strother et al. [11] argue that this could produce a time delay, and thus contributes to the extraction of direction-selective and not ON and OFF signals.

Response:

We thank the reviewer for the comments. As the reviewer pointed out, the proposed function of nAChR and mAChR signaling were discussed in the context of direction selectivity and not the ON and OFF selectivity. We amended our statement in the introduction (**P3**):

In the adult Drosophila visual system, although both functional and anatomical connectivity have been established for ON and OFF pathways, the molecular machinery mediating signal transductions within these pathways has yet to be identified¹²⁻¹⁵.

- It is unclear if ON and OFF pathways are conserved or convergent structure, and while some papers discuss that they might have a common origin, others do not use the word “conserved” at all (e.g. [6]). Together, this part of the paper is very unsatisfying.

Response:

We modified this statement in the introduction to avoid the misinterpretation (**P3**).

Extensive studies of adult Drosophila optic ganglia and vertebrate retinae suggest that the construction principles of ON and OFF selective pathways are shared among visual systems, albeit with circuit-specific implementations¹⁶⁻¹⁸.

8. The authors “conclude that calcium responses in pOLP are driven by the activities in glulOLP” (p.15). There is no evidence for that. In flies this can be tested by combining genetic silencing with physiological measurements. There is no evidence for reciprocal inhibitory interactions between cholinergic and glutamatergic transmission as claimed.

Response:

We would like to direct the reviewer’s attention to the experimental evidence we presented in **Fig. 6a** (old Fig. 5e-f), where we presented the results that lead us to this conclusion. We explained our rationale on **P17**:

Blocking Gao signaling in glu-IOLP also revealed close interactions between pOLP and glu-IOLP. PTX expression in glu-IOLP significantly reduced the latency of the light-induced response in pOLP without affecting its amplitude (Fig. 6a, b). Because of these matching temporal profiles, both with or without the PTX expression in glu-IOLP, we concluded that light-induced calcium responses in pOLP are driven by glu-IOLP's activities (Fig. 2a, b, 6a, b, Fig. S3b). Although the connectome study did not find direct synaptic interactions between the pair, it is possible that this effect is indirect. However, the close physical proximity between glu-IOLP and pOLP also suggests that they may interact through gap junctions⁸.

The reviewer suggested a genetic silencing experiment paired with physiological measurements. We indeed followed the exact same logic, leading us to perform the experiments shown in **Fig. 6**, in which the PTX manipulation in the glu-IOLP leads to changed calcium response kinetics in pOLP and dampened responses in cha-IOLP and LNvs.

We would also like to direct the reviewer's attention to evidence we presented for the reciprocal interactions between cha-IOLP and glu-IOLP, which is shown in **Fig. 6** and described on **P17**:

Using PTX expression to modify the temporal profile of glu-IOLP's activation, we examined how glu-IOLP interacts with cha-IOLP and two types of projection neurons, three components of the larval visual circuit. We expressed PTX in glu-IOLP using IOLPglu-Gal4 and monitored the light-induced calcium responses in all three OLPs through OLP-LexA driving expression of GCaMP6f. Consistent with our earlier observations, PTX expression accelerated the light response in glu-IOLP without affecting its amplitude. Importantly, this fast activation of glu-IOLP led to significant reductions in light-induced calcium responses in cha-IOLP (Fig. 6a, b), suggesting that glu-IOLP acts as an inhibitory input to cha-IOLP and that disrupting the temporal separation between the cha- and glu-IOLP's light responses affects the ability of cha-IOLP to respond to light. Collectively, the direct synaptic interactions between the two IOLPs demonstrated by the connectome study, the inhibitory effect of cholinergic inputs on glu-IOLP (Fig. 3a, b) and the dampened light response in cha-IOLP generated by early activation of glu-IOLP (Fig. 6a, b) support a model of reciprocal inhibitory interactions between glu-IOLP and cha-IOLP.

9. When recording in neurons downstream while expressing PTX in glu-IOLP, the authors interpret a drop in signal as “temporally delayed feedforward glutamatergic inhibition from glu-IOLP”. What argues against the more straightforward idea that glutamate is excitatory and they are just taking away an excitatory input?

Response:

Besides PTX expression in glu-IOLP dampening LNvs' light responses (**Fig. 6d-f**), there are two pieces of additional evidence:

- (1) Glutamatergic input is inhibitory to larval LNvs, which has been shown by previous studies and we included this statement in the manuscript on **P18**:

previous studies demonstrated that glutamatergic input inhibits larval LNvs through the action of a glutamate-gated chloride channel, *GluCl*¹⁹.

(2) We demonstrated that PTX expression eliminates the delay of glu-IOLP's calcium response and therefore transforms glu-IOLP into an ON responsive cell. We included this statement in the revision (**P16**):

Although additional studies are needed to fully characterize these responses, we found that PTX expression in glu-IOLP eliminates its OFF response while modifying the temporal profile of its ON response and thus effectively transforms glu-IOLP into an ON selective cell. Instead of transmitting light decrements, glu-IOLP expressing PTX transmits light increments to downstream VPNs and potentially disrupts the separation of the ON and OFF channels.

Taken together, our results are consistent with glutamate acting as an inhibitory input for LNvs. When PTX expression temporally advances the glu-IOLP's activation, the inhibitory glutamatergic input onto the LNvs reduces its light-induced calcium response (**Fig. 6d-f**).

10. The behavioral experiments lack specificity. That manipulation of this "OFF-neuron" not affect light induced behavior? Does cha-IOLP not affect this dark induced behavior? So far, nothing argues against the simple idea that there is no behaviorally relevant ON and OFF pathway split, but the neurons guiding these behaviors are just lacking visual drive.

Response:

It is unclear to us why the reviewer considered our behavior tests to "lack specificity". We used two assays to specifically test the larval response to either light increments or decrements. Predicated by our model, we expect to see differential response of cha- and glu-IOLP in these assays. Comparing results from light vs. dark induced behaviors (**Fig. 7b** vs. **7d**), we did observe a specific role for glu-IOLP in mediating the dark-induced response, consistent with its function in OFF-detection and supporting the idea of a split ON and OFF pathway. The rationale of the experiments is described on **P19-20**:

To illustrate the potential roles for IOLPs in transmitting ON and OFF signals from the PRs to the VPNs, and based on our findings and the connectivity map, we propose a model with three components. First, the pair of IOLPs exhibit distinct responses to light increments and decrements, thus acting as ON and OFF detectors. The sign-inversion required for OFF detection in glu-IOLP is mediated by the mAChR-B receptor. Second, while cha-IOLP displays clear ON selectivity, glu-IOLP has both ON and OFF responses. The OFF selectivity in glu-IOLP emerges from the temporal control of its activity by mAChR-B/G α o signaling. Third, extending our findings in the LNvs and pOLP to the rest of VPNs, we propose that, although downstream VPNs receive both cholinergic and glutamatergic inputs, there are specific groups of ON vs. OFF responsive VPNs that are functionally separated by their molecular compositions. ON-responsive VPNs (ON-VPNs), such as LNvs, are activated by cholinergic signaling and inhibited by glutamatergic signaling, while OFF-responsive VPNs (OFF-VPNs), such as

pOLP, behave the opposite way. Although additional physiological studies on other VPNs are needed to validate this model, given the lack of anatomical segregation of ON and OFF pathways, this functional separation of VPNs is a plausible solution to ensure the preservation of the ON and OFF signals at the level of VPNs (Fig. 7a).

Predicated by this model, an ON response is dominated by cholinergic transmissions from cholinergic PRs and cha-IOLP, while the extended inhibition of glu-IOLP via mAChR-B/Gao signaling ensures the activation of only the ON-VPNs. During an OFF response, with no cholinergic input, the glu-IOLP is solely responsible for activating OFF-VPNs. In terms of their roles in regulating behavioral output, cha-IOLP likely functions in modulating the strength and duration of the light-induced response, and glu-IOLP would be essential for initiating the dark-induced behavioral response (Fig. 7a).

11. this major criticism, it is interesting that the authors see different response polarities in two neurons downstream of photoreceptors, and there results on mAChRs are compelling. I would encourage the authors to follow up on this, using either Arclight recordings, calcium recordings at the terminal, paying attention to decreases in calcium, or using recently published indicators that are better at capturing decreases in calcium (Zhao et al 2018 Scientific Reports 8). Furthermore, a more ethological visual stimulus might be critical in allowing them to distinguish ON and OFF steps relative to an intermediate contrast.

Response:

We thank the reviewer for recognizing the interesting findings of our study. The reviewer suggested three additional approaches: (1) performing Arclight recordings (2) performing calcium recordings at the terminal (3) studying the decrease in calcium using a different indicator. In fact, we already present data on the first two points raised by the reviewer in this manuscript; Arclight recordings are shown in **Fig. 3a-b** and **5e-f**. Calcium recordings at terminal regions are shown in **Fig. 1d-e**, **5b, d** and **Fig. S6a-b**.

In terms of the decreased calcium signals, we quantified the trough of the biphasic calcium transient in glu-IOLP induced by light and presented the data in the new **Fig. S4**. In addition, we presented evidence for rectification of calcium signals in glu-IOLP in **Fig. 5e-f** (Please also see response to comment #2), which lead us to believe that the small decrease of calcium generated by hyperpolarization is a part of glu-IOLP's physiological properties and is unlikely to be changed by using another calcium indicator.

It is worth noting, however, that the results from all these experiments do not conflict with any of our current conclusions and, in fact, support the glu-IOLP's function in medicating OFF detection and that the temporal delay of glu-IOLP's light response likely contributes to ON and OFF selectivity in the larval visual circuit.

12. further comments:

*- It is not clear what a light pulse of “~30uW” means for the fly. Light intensities should be calibrated and given in photon / area*time. How do the authors know that what they are using is a physiologically relevant stimulus?*

Response:

Because we use a commercial system for image collection and light stimulation, we do not have access to the light source to measure the photon number emitted by the lasers. Therefore, we could not provide the measurement in the unit suggested by the reviewer. To provide additional information about the light stimulation we used, we included another measurement of the light intensity, $\mu\text{W}/\text{cm}^2$. We obtain the measurement using a calibrated slide power meter (Thorlab) that directly measures the power at sample plane. Detailed descriptions of the light intensity measurement and specific values for each experiment are included in the Methods.

The intensity of the light we used for stimulation is relatively low, in the range of 10-40 $\mu\text{W}/\text{cm}^2$. We measured ambient light level using the same equipment. During daylight hours, without an additional light source, the indoor light intensity at 488 nm is around 60-80 $\mu\text{W}/\text{cm}^2$. In addition, previous studies²⁰ indicated that light delivered at 470 nm elicits larval light avoidance at intensity level from 3.9 to 88 $\mu\text{W}/\text{cm}^2$. Based on this information, we concluded that the light intensity we used in this study is physiologically relevant.

13. - it is not mentioned in the text or the figure legend, how long the pulses are that are delivered in figures 1 and 2. The responses do not reflect a typical visual neuron's impulse response, as they last seconds!

Response:

The figure legends for **Fig. 1** and **2** clearly indicate that the light pulse is 100 ms and is additionally described in the main text. In this revision, we included this description in all figure legends.

The temporal profiles of the light induced calcium response resemble the ones observed in adult visual interneurons¹⁰. Please see the response to comment #3.

14. - Whenever calcium or voltage signals are shown, they should be shown as dF/F and not in arbitrary fluorescent units. This is not only standard, but also important to distinguish decreases and increases in the response (see for example *glu-IOLP* responses in Figure 2a,b)

Response:

We included representative raw data in **Fig. 2a, b, 3c, 6d** and **Fig. S3** to specifically illustrate the range of signal intensity we detected using different indicators. The rest of the data analyses and quantifications are performed using $\Delta F/F$. In addition, imaging data are also shown as $\Delta F/F$ in the average traces.

15. - P.5: The claim that glutamatergic transmission is "generally inhibitory" at *Drosophila* central synapses, is wrong. Although glutamate can be inhibitory, there are many ionotropic glutamate receptors, that are broadly expressed in the central nervous system, and pass excitatory signals (that would also explain there findings in Figure 6)

Response:

Please refer to our response to comment #9. To avoid confusion, we eliminated the statement from the introduction.

16. - if the Gal4 lines are used for behavior, it needs to be shown how specific the Gal4 line outside of the larval optic neuropil.

Response:

The expression patterns of the three Gal4 lines in the whole larval brain are presented in **Fig. S1**.

17. - In Figure 2, the pOLP response is happening ~5s after the visual stimulus in the individual trace, but ~2s after the stimulus in the averages.

Response:

The average trace is calculated by averaging the value of $\Delta F/F$ of different samples at each time point and therefore the onset of the calcium rise could be biased by individual samples. All our temporal quantifications are made using the peak time, which is less variable than the timing of the onset. In the specific case of Fig. 2, the peak times for the average and representative traces are both around 5 sec after light stimulation.

18. - The small circuit schematic in the imaging figures is very confusing. The green neuron appears to be Rh6-PRs. Then why do they talk to the terminal, and why are the cell bodies at the other end of the cell?

Response:

Please see **Fig. 1b** and **c** for confocal images illustrating the anatomical organization of the PR-OLP connections, which indicate that the PRs' axonal terminal and the IOLPs' processes make connections in the neuropil region (LON, grey circle). The somas of the IOLPs are on the lateral side of the brain lobe. Our schematics reflect this organization.

19. - It should probably be mentioned that the mAChR is not only expressed in glu-IOLP but also in another neuron in Figure 4C

Response:

The statement is included in the main text (**P13**) and in the figure legend (**Fig. 4b**):

The mAChR-B enhancer-driven EGFP expression revealed its extensive distribution in the 3rd instar larval brain and in neurons with soma positions and projection patterns that resemble the OLPs.

Reference:

1. Yuan, Q. *et al.* Light-induced structural and functional plasticity in Drosophila larval visual system. *Science* **333**, 1458-1462 (2011).
2. Chen, T.W. *et al.* Ultrasensitive fluorescent proteins for imaging neuronal activity. *Nature* **499**, 295-300 (2013).
3. Cao, G. *et al.* Genetically targeted optical electrophysiology in intact neural circuits. *Cell* **154**, 904-913 (2013).

4. Dana, H. *et al.* Sensitive red protein calcium indicators for imaging neural activity. *Elife* **5** (2016).
5. Salcedo, E. *et al.* Blue- and green-absorbing visual pigments of *Drosophila*: ectopic expression and physiological characterization of the R8 photoreceptor cell-specific Rh5 and Rh6 rhodopsins. *J Neurosci* **19**, 10716-10726 (1999).
6. Vasilaiuskas, D. *et al.* Feedback from rhodopsin controls rhodopsin exclusion in *Drosophila* photoreceptors. *Nature* **479**, 108-112 (2011).
7. Humberg, T.H. & Sprecher, S.G. Age- and Wavelength-Dependency of *Drosophila* Larval Phototaxis and Behavioral Responses to Natural Lighting Conditions. *Front Behav Neurosci* **11**, 66 (2017).
8. Larderet, I. *et al.* Organization of the *Drosophila* larval visual circuit. *Elife* **6** (2017).
9. Humberg, T.H. *et al.* Dedicated photoreceptor pathways in *Drosophila* larvae mediate navigation by processing either spatial or temporal cues. *Nat Commun* **9**, 1260 (2018).
10. Yang, H.H. *et al.* Subcellular Imaging of Voltage and Calcium Signals Reveals Neural Processing In Vivo. *Cell* **166**, 245-257 (2016).
11. Renteria, R.C. *et al.* Intrinsic ON responses of the retinal OFF pathway are suppressed by the ON pathway. *J Neurosci* **26**, 11857-11869 (2006).
12. Shinomiya, K. *et al.* Candidate neural substrates for off-edge motion detection in *Drosophila*. *Curr Biol* **24**, 1062-1070 (2014).
13. Strother, J.A. *et al.* The Emergence of Directional Selectivity in the Visual Motion Pathway of *Drosophila*. *Neuron* **94**, 168-182 e110 (2017).
14. Joesch, M., Schnell, B., Raghu, S.V., Reiff, D.F. & Borst, A. ON and OFF pathways in *Drosophila* motion vision. *Nature* **468**, 300-304 (2010).
15. Reiff, D.F., Plett, J., Mank, M., Griesbeck, O. & Borst, A. Visualizing retinotopic half-wave rectified input to the motion detection circuitry of *Drosophila*. *Nat Neurosci* **13**, 973-978 (2010).
16. Sanes, J.R. & Zipursky, S.L. Design principles of insect and vertebrate visual systems. *Neuron* **66**, 15-36 (2010).
17. Perry, M., Konstantinides, N., Pinto-Teixeira, F. & Desplan, C. Generation and Evolution of Neural Cell Types and Circuits: Insights from the *Drosophila* Visual System. *Annu Rev Genet* **51**, 501-527 (2017).
18. Ding, H., Smith, R.G., Poleg-Polsky, A., Diamond, J.S. & Briggman, K.L. Species-specific wiring for direction selectivity in the mammalian retina. *Nature* **535**, 105-110 (2016).
19. Collins, B., Kane, E.A., Reeves, D.C., Akabas, M.H. & Blau, J. Balance of activity between LN(v)s and glutamatergic dorsal clock neurons promotes robust circadian rhythms in *Drosophila*. *Neuron* **74**, 706-718 (2012).
20. Xiang, Y. *et al.* Light-avoidance-mediating photoreceptors tile the *Drosophila* larval body wall. *Nature* **468**, 921-926 (2010).

Reviewers' Comments:

Reviewer #1:

Remarks to the Author:

The authors have addressed my technical concerns and gave a more detailed explanation of the model.

Reviewer #2:

Remarks to the Author:

The authors edited the text, but failed to address a major flaw of the paper.

The data itself are convincing, and are very much arguing for a sign inversion happening downstream of PRs in that glu-IOLP has an opposite signed (OFF) response as compared to cha-IOLP (ON) and that this sign inversion is brought about by mAChRs and Gao. I find this part of the paper very convincing.

I however still cannot find evidence for the major claim that the authors are making about the temporal differences of the neurons' responses, nor any functional consequences arising from those. Instead, the response kinetics in glu-IOLP and cha-IOLP are very similar. The different response polarities can already be seen the neurons' initial, stimulus evoked, fast response. This is obvious when looking at:

- the Arclight data (Figure 3a): Both cell types peak at the same time, and have very similar decay kinetics, but opposite signed responses

- calcium imaging data (e.g. RCaMP data in Figure 3b): The increase in dF/F in RCaMP in cha-IOLP coincides in time with the (weaker) decrease of the RCaMP signal in glu-IOLP

Even within glu-IOLP, the increase in calcium signal in response to OFF and the decrease in response to light ON are very similar (Figure 3d)

(see attached the figures attached to this review, illustrating data from Figures 3.

While the mechanisms behind ON and OFF selectivity itself is interesting, the authors keep coming back to the slower increase in calcium signal that is following the initial decrease in calcium signal in glu-IOLP. This is very likely not a physiological response, and it will not come from a direct synaptic connection to photoreceptors, but might at best be feedback signals. This is for example true because

- it is not present in the Arclight data (it is true that voltage and calcium signals can differ from each other, as shown by Yang et al., but a calcium signal will not appear out of nowhere, if there is no voltage change),

- it is only seen as a rebound to decreases in calcium signal and is always gone when the initial decrease in signal is gone,

- it is happening on the order of seconds, which is too slow for synaptic transmission across one or two synapses and is also seen in pOLP which does not receive synaptic inputs based on connectomics.

Furthermore, genetic manipulations are striking in this context as they do NOT change the temporal dynamics of the initial response, but flip the sign:

- In Figure 5a and b, the initial decrease in GCaMP signal is transformed into an increase upon GalphaO KD

- In Figure 5c and d, the initial decrease in GCaMP signal is transformed into an increase upon PTX expression

- In Figure 5e, the same is true for Arclight data

Whenever this sign inversion is happening, the slow response is gone, further arguing that this is an unspecific side effect of a decrease in calcium.

(also see attached the figures attached to this review, illustrating data from Figures 5).

Some specific parts of the text that are drawing misleading conclusions are:

p. 11: "light hyperpolarizing glu-IOLP at the onset of the light exposure and generate sustained inhibition until the offset"

-> this is not at all visible in figure 3c

p. 11: "Compared to the slow calcium response induced by 100 ms light pulses, this dark induced OFF response has a similar amplitude, but a significantly shorter latency"

-> the latency is exactly the same, but the signal has the opposite polarity, this is neither mentioned nor quantified

p.11: "Similar recordings indicate that cha-IOLP does not respond to dark pulses and only generate the fast ON response"

-> where is the data for this?

p. 15: "we conclude that the extent and timing of glu-IOLP's light response"

-> the timing of the fast response is always the same, just the sign is switched (and voltage recordings further demonstrate that)

p.15: "The [Arclight] response is temporally correlated with the biphasic calcium transient

-> control Arclight data aren't biphasic at all, and very different from calcium data!! But the baseline might be shifted after stimulation (Fig. 5e)

p.17: "using PTX to modify the temporal profile of glu-IOLP's activation

-> this is not an accurate statement. First and foremost, the sign of the response is switched

p.17: "PTX expression accelerated the light response in glu-IOLP

-> No. The kinetics of the initial light response is the same. The sign is switched.

p.17: "that disrupting the temporal separation between the cha- and glu-IOLP's light response affects the ability of cha-IOLP to ..

-> this should read: "that disrupting normal glu-IOLP's light responses". There is no single experiment that specifically alters the temporal dynamics.

p.17: the inhibitory effect of cholinergic inputs on glu-IOLP (Fig. 3a,b)"

-> there is no data in there that shows this, that would in fact require silencing / blocking inhibition in cha-IOLP while measuring responses in glu-IOLP

p. 17: "PTX expression in glu-IOLP significantly reduces the latency of the light-induced response in pOLP without affecting its amplitude (Fig. 6a,b)

-> this should read "light induced calcium responses in pOLPs are driven by glu-IOLP's activity", but there is no evidence for the temporal aspect (latency) of the claim

In summary, the present work could be a solid study if the authors were concentrating on the initial responses of opposite sign, and describe the mechanisms that is establishing these ON and OFF responses (mAChR, Gao). The subsequent work on downstream circuitry (Figure 6) and behavior (Figure 7) further shows that glu-IOLP is important for downstream function. But I would encourage the authors to stop arguing about temporal differences. The authors' interpretations that temporal kinetics are altered cannot be hold, which is currently the major claim of the paper. Even the title suggests that this "temporal control of inhibition" generated ON & OFF selectivity, whereas the data show that ON and OFF selectivity are already generated with exactly the same kinetics.

If the authors indeed want to follow up on their hypothesis and keep this as a central claim in there paper, they needed to show that this late calcium response is specific, and they needed to provide further evidence for mutual inhibition. In fact, there would be ways to test this, by for example blocking the output of IOLPs and glulIOLP while recording their response properties, probing inhibition, etc..

Further comments:

The authors compare their measured responses to impulse responses measured by Yang et al., but these are happening on time scales that differ by an order of mangiuted

Fig 2a/B: there is still no reason not to plot calcium signals of individual neurons as dF/F

"pOLP is mainly driven by Rh6 input"

-> looking at the timing of this response (seconds after the stimulus) this is very likely no direct PR input, which further agrees w/ connectomics as even stated by the authors ("not indicated in the connectomics study")

p.14: mAChR-B knockdown also significantly dampened the light response in cha-IOLP suggesting that eliminating the inhibition of glu-IOLP could potentially impact cha-IOLP's light response

->here, the knockdown is done in both glu-IOLP and cha-IOLP (using GMR84D2), so the conclusions should be phrased more carefully

The work in its current form is below the quality of work suitable for Nature Communication.

Reviewer #3:

Remarks to the Author:

I have been asked to review this paper during this second round review process, though I was not one of the initial reviewers. This is not straightforward to do. I am therefore focusing on evaluating how first round concerns were addressed in this revised manuscript. As much as possible, I am not making any additional critiques of the paper or methods. I do have a few minor comments that should simple to address.

The authors present convincing data that two neurons postsynaptic to PRs in larval Drosophila respond

oppositely to light, with one being an ON cell and the other an OFF cell. The impulse responses in calcium measurements in these cells can be confusing, as others have noted (Yang et al., 2016), and the most convincing data are the responses to ON and OFF steps. The authors then present interesting genetic evidence indicating that metabotropic acetylcholine receptors are responsible for the inversion in one set of these interneurons. They hypothesize other interactions that could explain changes in the circuit when these receptors are disrupted in the glu-IOLPs. They finish by presenting evidence that this signaling difference matters to the larvae, since they observe light-OFF specific behavioral deficits when knocking down m-AchR-B with RNAi in the OFF cell.

The rectifying effects of calcium and/or calcium indicators make it appear as though there is a delayed response to light pulses, when this is in fact a rebound from hyperpolarization. I found this clear in the manuscript when I read it, so it appears to have been clarified from the first round. I think I was most convinced of the ON/OFF signals in these two interneurons by the steps up and down in light intensity, since pulses can be hard to interpret, due to the biphasic nature of the response.

I share the authors' interpretation of the delayed calcium signals in glu-IOLPs in response to light flashes. They quantify this rebound signal, but note the initial decrease as the initial part of a biphasic OFF filter. And this is what one might expect from an OFF cell, given Yang et al.'s data. As much as possible, I think it would be clearest to get to the steps as soon as possible, due to the non-intuitive nature of this 'delayed' response in calcium signals to light flashes.

Minor

p. 3: "molecular machinery not yet identified" in adult fly ON/OFF channels. This doesn't seem like quite the right assessment for the field. First, if not identified, then it's unclear what papers you're citing. Several of your citations are much more targeted at downstream motion computations rather than at the split into ON and OFF pathways. My reading of the field is as follows: L1 and L2 receive the majority of synaptic outputs from PRs, and both hyperpolarize in response to light, due to the histamine-gated chloride channels they express. That is, L1 and L2 are both OFF cells, and respond somewhat linearly to both contrast increments and decrements (Clark et al., 2011). The Reiff paper cited reports that L2 responds only to decrements, but this is difficult to reconcile with our current understanding of the circuit and the responses to light increments observed in Tm1 and Tm2 in both Behnia et al. and Yang et al. Strother et al., 2014 is also a nice characterization of lamina-specific ON/OFF segregation in flies. There is some work on L1 and L2 expression patterns, showing that they are glutamatergic and cholinergic, respectively. Gao et al., 2008 showed this with chat and vglut split-Gal4 drivers that labeled L2 and L1, while Davis et al., Bioarxiv 2018, found a similar result with single cell RNA sequencing. To explain the data, it may be that Tm1/Tm2 express nicotinic ach receptors and Mi1 and Tm3 express gluCl. However, it is not proven or shown at the level of detail and causality you show here. This issue comes up again on p. 16 with the discussion of Mi1, Tm1-3. It is interesting to note that the signal inversion appears to happen one synapse later in adult Dmel than in larvae and in mammalian retina. I think the summary of the data in the adult should be made a bit more specific, which would also serve to better emphasize this paper's contribution.

F1b: it might be useful to outline the regions of the brain described in 1a.

p. 11: "In contrast, the light onset induced a small yet noticeable reduction of calcium signal in glu-IOLP". I don't see this in Figure 3c, which is referenced for this effect. 3d, yes.

F3d: make it clear in main text that these ON and OFF pulses are on a background of light (the same background)? That's what I assumed from stating they were at the same contrast. What was the contrast?

p. 19: "predicated by" does not seem to be the right word, I think. Also p. 20.

p. 24: motion detection vs. direction-selectivity. The authors should take care with terminology here. Do they really want to imply that larvae can detect directional motion? I haven't seen evidence of that, but may have missed it. But this could be unpacked a bit. If it's not DS, then I'm less sure this is a worthy comparison, since 'motion detection' can in just be detecting local luminance changes, which generally occur only when there is motion in the field of view.

p. 25: I'm a bit skeptical of calling this $G_{\alpha o}$ a "convergence", if I'm interpreting this discussion point correctly. What seems common is using a GPCR and an ionotropic receptor to the same neurotransmitter. These are even used differently for ON and OFF in mammalian retina vs. here, since the PRs in the system respond with opposite polarities. There are only so many G-proteins, right?

Since the authors have proposed signs for most of the interactions in their diagram, it would be helpful to indicate excitatory vs. inhibitory synapses in a final model, including in the interactions between IOLPs.

I believe the authors could clear up considerable confusion by referring to the delayed calcium response to light flashes in OFF cells as 'delayed', rather than 'slow'. This effectively contrasts it with 'immediate' in the ON cell. They refer to this delayed calcium signal as an 'ON response' at several points; this characterization should be avoided, given their interpretation of the data elsewhere. An ON response corresponds to a positive response to a positive derivative, while this is a response to a flash of light, which contains positive and negative temporal derivatives (as R2 noted).

The authors should also not make claims about rectification of the signals in the IOLPs based on light flashes alone; light and dark flashes would have to be compared. That claim is not central and could be easily removed.

Fig 3

Fig 3: same kinetics of decrease in dF/F to OFF and increase of dF/F to ON

Fig 3: same kinetics of both RCaMP responses in cha-IOLP and glu-IOLP

Fig 5

Point-by-point response to reviewers' comments

NCOMMS-18-25662-C: "Temporal control of inhibition *via* muscarinic acetylcholine receptor signaling generates ON and OFF selectivity in a simple visual circuit" by Qin et al.

We thank the reviewers for their evaluations of our work. Both Reviewer 1 and 3 did not raise additional technical or conceptual issues and agreed with us on the main conclusions of our study. In addition, Reviewer 3 offered specific suggestions on how to modify the main text to improve the clarity and accuracy of our statements. We greatly appreciate these comments and made modifications to the main text following the reviewers' suggestions and the format requirements of Nature Communications research articles.

Besides editing the main text, to specifically address Reviewer 2's concerns about the physiological relevance of the delayed calcium response in glu-IOLP, we provided additional experimental evidence to illustrate the temporal profile of light-evoked glutamate release from glu-IOLP. Using iGluSnFR recordings on the LNV dendrites, we observed a slow glutamate transient that exhibits similar temporal kinetics to the delayed glu-IOLP calcium response and is also accelerated by PTX expression in glu-IOLP (**new Figure 7a-c**). In combination with the results in Figure 6 in which we show effects of accelerated activation of glu-IOLP on the light responses in cha-IOLP, pOLP, and LNVs, our recordings of glutamate transients strongly support the physiological function of the delayed calcium response and the role of temporal regulation in larval visual processing. We hope the additional data and the modifications we made to the main text will satisfy Reviewer 2.

Major changes we made in the revision:

1. To emphasize our contribution in identifying molecular and circuit mechanisms underlying ON vs. OFF discrimination in the *Drosophila* system, we changed the title to: "**Muscarinic acetylcholine receptor signaling generates OFF selectivity in a simple visual circuit**".
2. To comply with the format requirement of the journal, we edited the main text and reduced the length to ~5,500 words in total (Introduction, Results, Discussion). Please let us know if it is acceptable.
3. To address Reviewer 2's concern, we included results describing the light-evoked slow glutamate transients from glu-IOLP (**new Fig. 7a-c**).
4. To comply with the format requirement of figure legends, we changed **old Fig. 7a** to **new Fig. 7d**, and **old Fig. 7b-e** to **new Fig. 8a-d**.
5. To improve clarity and accuracy of our statements following Reviewer 3's suggestions: we eliminated the text related to rectification of calcium signals; modified the description of "the **Slow** calcium transients" to "the **Delayed** calcium transients"; eliminated the use of "ON response" in glu-IOLP; changed "**calcium responses**" to "**calcium rises**" when we describe the glu-IOLP activation; and modified the Introduction and Discussion sections.

We believe we have addressed all comments from the reviewers. We hope the editor and reviewers find this revised version now suitable for publication in Nature Communications.

Reviewer #1 (Remarks to the Author):

The authors have addressed my technical concerns and gave a more detailed explanation of the model.

Reviewer #2 (Remarks to the Author):

The authors edited the text, but failed to address a major flaw of the paper.

The data itself are convincing, and are very much arguing for a sign inversion happening downstream of PRs in that glu-IOLP has an opposite signed (OFF) response as compared to cha-IOLP (ON) and that this sign inversion is brought about by mAChRs and Gao. I find this part of the paper very convincing.

I however still cannot find evidence for the major claim that the authors are making about the temporal differences of the neurons' responses, nor any functional consequences arising from those. Instead, the response kinetics in glu-IOLP and cha-IOLP are very similar.

Response:

We thank Reviewer 2 for evaluating our manuscript and the positive comments on our findings related to the light-evoked ON and OFF responses in the IOLPs. However, Reviewer 2 remained skeptical about our interpretations of the delayed calcium response in glu-IOLP and challenged the proposed function of the temporally-regulated inhibition in larval visual computation. As detailed in our response below, we believe that **the delayed increase in glu-IOLP's intracellular calcium is functionally important for glutamate release and signal transduction within the visual circuit**. In this revision, we modified main text to clarify our claims and emphasize our main findings and provided additional experimental evidence to support the physiological relevance of the delayed calcium responses and the temporal regulation of glu-IOLP's activation.

1. The different response polarities can already be seen the neurons' initial, stimulus evoked, fast response. This is obvious when looking at:

- the Arclight data (Figure 3a): Both cell types peak at the same time, and have very similar decay kinetics, but opposite signed responses

- calcium imaging data (e.g. RCaMP data in Figure 3b): The increase in dF/F in RCaMP in cha-IOLP coincides in time with the (weaker) decrease of the RCaMP signal in glu-IOLP

Even within glu-IOLP, the increase in calcium signal in response to OFF and the decrease in response to light ON are very similar (Figure 3d)

(see attached the figures attached to this review, illustrating data from Figures 3.)

Response:

We thank the reviewer for evaluate our data and we agree with the reviewer that **there is no difference in the temporal kinetics of the initial light-induced depolarization in cha-IOLP and hyperpolarization in glu-IOLP**, nor the accompanying changes in the calcium signals. This is reflected in our manuscript (Page 7-8). However, our results also indicate that glu-IOLP exhibits **a biphasic response** towards light stimulation and that **the delayed calcium transient is also a part of the light-evoked physiological response**. In this revision, to improve clarity, we modified

the description of “the **Slow** calcium transients” to “the **Delayed** calcium transients”, eliminated the use of “ON response” in glu-IOLP, and changed “**calcium responses**” to “**calcium rises**” when we describe the activation of glu-IOLP.

2. While the mechanisms behind ON and OFF selectivity itself is interesting, the authors keep coming back to the slower increase in calcium signal that is following the initial decrease in calcium signal in glu-IOLP. This is very likely not a physiological response, and it will not come from a direct synaptic connection to photoreceptors, but might at best be feedback signals. This is for example true because

- it is not present in the Arlight data (it is true that voltage and calcium signals can differ from each other, as shown by Yang et al., but a calcium signal will not appear out of nowhere, if there is no voltage change).

- it is only seen as a rebound to decreases in calcium signal and is always gone when the initial decrease in signal is gone.

- it is happening on the order of seconds, which is too slow for synaptic transmission across one or two synapses and is also seen in pOLP which does not receive synaptic inputs based on connectomics.

Response:

We thank the reviewer for recognizing our findings related to the ON and OFF responses in the IOLPs, but we disagree with the reviewer’s comments on the delayed calcium response of glu-IOLP being “**not a physiological response**”, which is at the center of the negative comments from the reviewer. In fact, this delayed calcium increase is much more robust and consistent than the initial reduction in calcium signals in all experiments and greatly facilitated our genetic analyses on mAChR-B signaling. In addition, in this revision, we included glutamate release measurements (**new Fig. 7a-c**), which indicate that the delayed calcium response in glu-IOLP leads to a slow glutamate release onto its target VPNs.

To specifically address reviewer’s comments,

1. In Figure 5e-f, we show a biphasic Arlight trace that exhibits similar temporal dynamics to the calcium transient. We acknowledge that the Arlight trace typically presents with a low amplitude that can be obscured by noise, making the depolarization is less obvious in the average trace. Given the poor signal to noise ratio, we were able to quantify the peak value and latency of the depolarization event. We attached a figure here to illustrate the temporal correlations among the light-induced depolarization, the delayed calcium response and the slow glutamate release in glu-IOLP.

Temporal correlation of the light induced physiological events in glu-IOLP

(Red dashed lines indicate ~3.5s after the light pulse)

2. As part of the light-evoked response, the delayed calcium response can be manipulated by mAChR-B knock-down and G α o inhibitors, which is not in conflict with our interpretations. In addition, we used four different types of genetic manipulations, including two different RNAi lines targeting mAChR-B, the RNAi knock-down of G α o and the PTX expression, all of which showed a varying degree of effects on the waveform of light-induced calcium responses in glu-IOLP, supporting our model of glu-IOLP activation can be temporally-regulated by the light induced inhibition through mAChR-B/G α o signaling.
3. The slow kinetics of the delayed calcium transient and its relationship with the voltage changes measured by Arlight recordings have been addressed previously and we would like to direct the reviewer's attention to the following statement from our previous response: "This slow calcium rise induced by light pulses is prominent in all glu-IOLP recordings, especially in the terminal regions (**Fig. 1-6**). This significant temporal delay can be measured using different indicators and paradigms consistently and can also be modified genetically (**Fig. 4d, e, 5a-f, Fig. S6**). In addition, changing the temporal profile of the glu-IOLP's calcium response strongly impact both downstream VPNS' light responses (**Fig. 6**) and dark induced behaviors (**Fig. 7**). Therefore, we believe we have strong evidence to support our claims. We acknowledge that the temporal scale of this ON response is surprisingly slow. However, it is part of the physiological properties of glu-IOLP and appears to be consistent with the temporal scale of the dark-induced pausing behavior (**P20, Fig. 7b-c**). **Besides our own experimental evidence, the long-latency (1-2 sec) ON response in the OFF pathway has also been observed and investigated in mammalian retinae by electrophysiology**". Studies indicated that this long-latency ON response is generated within the OFF pathway and is suppressed by the ON pathways, similar as the delayed calcium response we observed in glu-IOLP. Although there are clear differences in circuit architecture, intrinsic properties of neurons, and experimental paradigms, studies in the mammalian visual system indicate that the temporal scale is not a reason to consider the delayed response non-specific."
4. Our data in Figure 6 and Page 16 indicate that the activity of pOLP is driven by glu-IOLP. Their light responses have similar kinetics. Although there is no evidence of a direct synaptic interaction, glu-IOLP and pOLP may interact through gap junctions.

Following Review 3's suggestion, to avoid confusion, we modified the main text to clarify the type of calcium responses we are quantifying. Please also see our response to comment #1.

3. Furthermore, genetic manipulations are striking in this context as they do NOT change the temporal dynamics of the initial response, but flip the sign:

- In Figure 5a and b, the initial decrease in GCaMP signal is transformed into an increase upon GalphaO KD

- In Figure 5c and d, the initial decrease in GCaMP signal is transformed into an increase upon PTX expression

- In Figure 5e, the same is true for Arlight data

Whenever this sign inversion is happening, the slow response is gone, further arguing that this is an unspecific side effect of a decrease in calcium.

(also see attached the figures attached to this review, illustrating data from Figures 5).

Response:

We thank the reviewer for evaluate our data and provide his interpretation. However, while the reviewer focused on the initial light-induced voltage and calcium responses, **we also evaluated the activation of glu-IOLP, which is reflected by the increase of intracellular calcium signals that lead to the neurotransmitter release.** The consistency and robustness of the delayed calcium rise and its acceleration produced by our genetic manipulations are evident in Figure 5 marked by the reviewer.

The mAChR-B/G α o manipulations not only eliminated the initial reduction in calcium signal, but also changed the temporal kinetics of the calcium rise in glu-IOLP (**Fig. 5, Fig. S6**) and the glutamate release from glu-IOLP (**new Fig. 7a, b**), which modified the light-induced responses in cha-IOLP, pIOLP, and LNvs (**Fig. 6**). To clarify this point, we modified the text to reflect that we measured the latency of **light-induced calcium increases** in our experiments on Page 13:

G α o knock-down eliminated the initial reduction of calcium signal and produced a fast calcium rise instead of the typical delayed response (Fig. 5a, b). Blocking Gao activity in glu-IOLP by Pertussis toxin (PTX) expression, which specifically inhibits Gao in Drosophila³, also eliminated the initial calcium reduction and accelerated the light-induced calcium rise without significantly affecting its amplitude, an effect observable in both soma and terminal regions of glu-IOLP (Fig. 5c, d).

4. Some specific parts of the text that are drawing misleading conclusions are:

p. 11: “light hyperpolarizing glu-IOLP at the onset of the light exposure and generate sustained inhibition until the offset”

-> this is not at all visible in figure 3c

Response:

We amended our statement to improve accuracy on Page 10:

RCaMP recordings showed that, with an extended light exposure, cha-IOLP only responded to the light onset with a fast calcium transient, demonstrating its specific response to light increments. In contrast, glu-IOLP responded to the light offset with an immediate calcium rise, suggesting that glu-IOLP is activated by the light decrements (Fig. 3c).

p. 11: “Compared to the slow calcium response induced by 100 ms light pulses, this dark induced OFF response has a similar amplitude, but a significantly shorter latency”

-> the latency is exactly the same, but the signal has the opposite polarity, this is neither mentioned nor quantified.

Response:

Here we quantified the calcium rise. The initial reduction in calcium signal induced by light, although visible in the average trace, is only around 1%, preventing us from quantifying the response accurately. This is likely due to the low sensitivity associated with the RCaMP imaging

(The comparison between RCaMP and GCaMP recording is shown in Supplemental Fig. 8). To avoid confusion, we modified our statement on Page 10-11:

Compared to the delayed calcium rise induced by light pulses, this dark-induced OFF response has a similar amplitude, but a significantly shorter latency (Fig. 3d). Similar recordings indicate that cha-IOLP does not respond to dark pulses and only generates the fast ON response to light pulses (Fig. 4d-e).

p.11: “Similar recordings indicate that cha-IOLP does not respond to dark pulses and only generate the fast ON response”
-> where is the data for this?

Response:

The result for cha-IOLP’s negative OFF response is shown in Figure 4d-e. We modified the text accordingly.

p. 15: “we conclude that the extent and timing of glu-IOLP’s light response”
-> the timing of the fast response is always the same, just the sign is switched (and voltage recordings further demonstrate that)

Response:

Please see our response to comment #3. Here we described the light-induced calcium rise in glu-IOLP. To avoid confusion, we modified the text on Page 14:

By comparing the outcomes generated by different manipulations that potentially block mAChR-B/G α signaling to varying degrees (Fig. 5a-d, Supplementary Figure 6), we conclude that the extent and timing of glu-IOLP’s activation is regulated by mAChR-B/G α signaling.

p.15: “The [Arclight] response is temporally correlated with the biphasic calcium transient -> control Arclight data aren’t biphasic at all, and very different from calcium data!! But the baseline might be shifted after stimulation (Fig. 5e)

Response:

Please see our response to comment #2 and the attached figure.

p.17: “using PTX to modify the temporal profile of glu-IOLPL’s activation -> this is not an accurate statement. First and foremost, the sign of the response is switched

Response:

Please see our response to comment #3.

p.17: “PTX expression accelerated the light response in glu-IOLP -> No. The kinetics of the initial light response is the same. The sign is switched.

Response:

Please see our response to comment #3.

p.17: “that disrupting the temporal separation between the cha- and glu-IOLP’s light response affects the ability of cha-IOLP to ..

-> this should read: “that disrupting normal glu-IOLP’s light responses”. There is no single experiment that specifically alters the temporal dynamics.

Response:

Please see our response to comment #3.

p.17: the inhibitory effect of cholinergic inputs on glu-IOLP (Fig. 3a,b)”

-> there is no data in there that shows this, that would in fact require silencing / blocking inhibition in cha-IOLP while measuring responses in glu-IOLP

Response:

Glu-IOLP receives cholinergic inputs from both photoreceptors and cha-IOLP. Currently, we do not have specific tools to separate these two sources. However, considering the light-induced hyperpolarization in glu-IOLP and depolarization in cha-IOLP, it is reasonable to conclude that the combined cholinergic inputs from both photoreceptors and cha-IOLP is inhibitory to glu-IOLP. We modified the main text on Page 15 to improve accuracy.

Collectively, the direct synaptic interactions between the two IOLPs demonstrated by the connectome study ¹⁷, the inhibitory effect of cholinergic inputs from both photoreceptors and cha-IOLP on glu-IOLP (Fig. 3a, b), and the dampened light response in cha-IOLP generated by accelerated activation of glu-IOLP (Fig. 6a, b) support a model of reciprocal inhibitory interactions between glu-IOLP and cha-IOLP.

p. 17: “PTX expression in glu-IOLP significantly reduces the latency of the light-induced response in pOLP without affecting its amplitude (Fig. 6a,b)

-> this should read “light induced calcium responses in pOLPs are driven by glu-IOLP’s activity”, but there is no evidence for the temporal aspect (latency) of the claim

Response:

Here, we described the calcium increases in pOLP. We modified the text on Page 16:

PTX expression in glu-IOLP significantly reduced the latency of the light-induced calcium rises in pOLP without affecting its amplitude (Fig. 6a, b). Because of these matching temporal profiles, both with and without the PTX expression in glu-IOLP, we concluded that the light-induced calcium increase in pOLP is driven by glu-IOLP’s activities (Fig. 2a, b, 6a, b, Supplementary Figure 3b).

In summary, the present work could be a solid study if the authors were concentrating on the initial responses of opposite sign, and describe the mechanisms that is establishing these ON and OFF responses (mAChR, Gao). The subsequent work on downstream circuitry (Figure 6) and behavior (Figure 7) further shows that glu-IOLP is important for downstream function. But I would encourage the authors to stop arguing about temporal differences. The authors’

interpretations that temporal kinetics are altered cannot be hold, which is currently the major claim of the paper. Even the title suggests that this “temporal control of inhibition” generated ON & OFF selectivity, whereas the data show that ON and OFF selectivity are already generated with exactly the same kinetics.

Response:

We thank the reviewer for recognizing the significance of our findings. We hope the additional experimental evidence and the detailed responses we provided here will convince the reviewer that the delayed calcium response in glu-IOLP is specific and physiologically relevant.

The set of results we presented in Figure 5 to 7, which described the mAChR-B signaling-mediated temporal regulation of glu-IOLP and its impacts on circuit function, are important for us to understand not only how the ON and OFF signals are detected, but also how they are preserved and transmitted to downstream VPNs. The temporal regulation is supported by our genetic and imaging studies and provides a plausible explanation about how activation of downstream ON- and OFF-VPNs could be segregated and reinforced through temporal regulations of glutamatergic neurotransmission. We described our model **on Page 18, Fig. 7d**, and discussed its implications on **Page 21**.

Although our experimental evidence clearly demonstrates the unique temporal kinetics of light-induced responses in glu-IOLP, we acknowledge that additional behavior experiments are needed to fully support our claim on the role of temporal controlled inhibition in generating ON and OFF selectivity. To avoid overstatement, in this revision, we changed our title to “**Muscarinic acetylcholine receptor signaling generates OFF selectivity in a simple visual circuit**” and discussed the role of temporal regulation in visual processing mainly in the model we proposed (Page 18) and Discussion (Page 21).

If the authors indeed want to follow up on their hypothesis and keep this as a central claim in there paper, they needed to **show that this late calcium response is specific**, and they needed to provide further **evidence for mutual inhibition**. In fact, there would be ways to test this, by for example blocking the output of IOLPs and glulOLP while recording their response properties, probing inhibition, etc...

Response:

We thank the reviewer for the suggestions. As mentioned above, in the revised manuscript, we modified the title and main text to emphasize our findings on the role of mAChR-B receptor signaling in mediating the OFF selectivity in larval visual circuit. However, these changes are not in conflict with the results and conclusions we presented in the current version of the manuscript, which contains evidence supporting the specificity of the delayed calcium response and the mutual inhibition between cha- and glu-IOLP.

To specifically address Reviewer 2’s concerns about the specificity and physiological relevance of the delayed calcium response in glu-IOLP, we provide additional experimental evidence to illustrate the temporal profile of light-evoked glutamate release from glu-IOLP. Using iGluSnFR recordings on the LNv dendrites, we observed a slow glutamate transient that exhibits similar temporal kinetics to the delayed glu-IOLP calcium response and is also accelerated by PTX

expression in glu-IOLP (**new Figure 7a-c**). In terms of evidence for mutual inhibition, we have addressed this comment previously. The evidence we presented for the reciprocal interactions between cha-IOLP and glu-IOLP is shown in **Figure 6** and described in **Page 15**. In short, our study clearly demonstrated the role of mAChR-B signaling in generating OFF selectivity, at the same time, also revealed the circuit mechanism mediating ON vs. OFF discrimination in the larval visual circuit.

5. Further comments:

The authors compare their measured responses to impulse responses measured by Yang et al., but these are happening on time scales that differ by an order of magnitude

Response:

We would like to point out that the temporal kinetics of the light evoked calcium response in the OFF neurons in adult visual systems (marked by red arrows in the picture below), although faster than what we observed in larval IOLPs, do not differ by an order of magnitude. The peak time of the calcium transient is also in the order of seconds, and the waveforms for calcium and voltage responses are almost identical to our results. Please see the image below.

Figure 6, Yang, et al, 2016

Fig 2a/B: there is still no reason not to plot calcium signals of individual neurons as dF/F

Response:

As mentioned in our previous response, we would like to show the representative raw data in our figures without changing the measurements. The $\Delta F/F$ values have been shown and quantified for average traces.

“pOLP is mainly driven by Rh6 input”

-> looking at the timing of this response (seconds after the stimulus) this is very likely no direct PR input, which further agrees w/ connectomics as even stated by the authors (“not indicated in the connectomics study”)

Response:

Please see our response to comment #2. Based on our observations and the connectome data, we propose that the glu-IOLP and pOLP may interact through gap junctions.

PTX expression in glu-IOLP significantly reduced the latency of the light-induced calcium rises in pOLP without affecting its amplitude (Fig. 6a, b). Because of these matching temporal profiles, both with and without the PTX expression in glu-IOLP, we concluded that the light-induced calcium increase in pOLP is driven by glu-IOLP's activities (Fig. 2a, b, 6a, b, Supplementary Figure 3b). Although the connectome study did not find direct synaptic interactions between the pair, it is possible that this effect is indirect. However, the close physical proximity between glu-IOLP and pOLP also suggests that they may interact through gap junctions¹⁷.

*p.14: mAChR-B knockdown also significantly dampened the light response in cha-IOLP suggesting that eliminating the inhibition of glu-IOLP could potentially impact cha-IOLP's light response
->here, the knockdown is done in both glu-IOLP and cha-IOLP (using GMR84D2), so the conclusions should be phrased more carefully*

Response:

We modified our statement on Page 13:

Intriguingly, the mAChR-B knock-down also significantly dampened the light responses in cha-IOLP (Fig. 4d, e), suggesting that eliminating the inhibition of glu-IOLP potentially impacts cha-IOLP's light response. However, because the knock-down of mAChR-B was performed in both IOLPs, further evidence is needed to support the interaction between the IOLPs.

The work in its current form is below the quality of work suitable for Nature Communication.

Response:

We believe we have addressed all concerns from the reviewer with new experimental evidence and modification of the text.

Reviewer #3 (Remarks to the Author):

I have been asked to review this paper during this second round review process, though I was not one of the initial reviewers. This is not straightforward to do. I am therefore focusing on evaluating how first round concerns were addressed in this revised manuscript. As much as possible, I am not making any additional critiques of the paper or methods. I do have a few minor comments that should be simple to address.

The authors present convincing data that two neurons postsynaptic to PRs in larval Drosophila respond oppositely to light, with one being an ON cell and the other an OFF cell. The impulse responses in calcium measurements in these cells can be confusing, as others have noted (Yang et al., 2016), and the most convincing data are the responses to ON and OFF steps. The authors then present interesting genetic evidence indicating that metabotropic acetylcholine receptors are responsible for the inversion in one set of these interneurons. They hypothesize other interactions that could explain changes in the circuit when these receptors are disrupted in the glu-IOLPs. They finish by presenting evidence that this signaling difference matters to the larvae, since they

observe light-OFF specific behavioral deficits when knocking down m-AchR-B with RNAi in the OFF cell.

The rectifying effects of calcium and/or calcium indicators make it appear as though there is a delayed response to light pulses, when this is in fact a rebound from hyperpolarization. I found this clear in the manuscript when I read it, so it appears to have been clarified from the first round. I think I was most convinced of the ON/OFF signals in these two interneurons by the steps up and down in light intensity, since pulses can be hard to interpret, due to the biphasic nature of the response.

I share the authors' interpretation of the delayed calcium signals in glu-IOLPs in response to light flashes. They quantify this rebound signal, but note the initial decrease as the initial part of a biphasic OFF filter. And this is what one might expect from an OFF cell, given Yang et al.'s data. As much as possible, I think it would be clearest to get to the steps as soon as possible, due to the non-intuitive nature of this 'delayed' response in calcium signals to light flashes.

Response:

We greatly appreciate the reviewer's constructive comments. Reviewer 3 evaluated our revised manuscript thoroughly and agreed with our interpretations of the results. Without raising major technical or conceptual issues, the reviewer also provided insightful suggestions about how to modify the manuscript to improve clarity and emphases of our main findings.

Minor

p. 3: "molecular machinery not yet identified" in adult fly ON/OFF channels. This doesn't seem like quite the right assessment for the field. First, if not identified, then it's unclear what papers you're citing. Several of your citations are much more targeted at downstream motion computations rather than at the split into ON and OFF pathways. My reading of the field is as follows: L1 and L2 receive the majority of synaptic outputs from PRs, and both hyperpolarize in response to light, due to the histamine-gated chloride channels they express. That is, L1 and L2 are both OFF cells, and respond somewhat linearly to both contrast increments and decrements (Clark et al., 2011). The Reiff paper cited reports that L2 responds only to decrements, but this is difficult to reconcile with our current understanding of the circuit and the responses to light increments observed in Tm1 and Tm2 in both Behnia et al. and Yang et al. Strother et al., 2014 is also a nice characterization of lamina-specific ON/OFF segregation in flies. There is some work on L1 and L2 expression patterns, showing that they are glutamatergic and cholinergic, respectively. Gao et al., 2008 showed this with chat and vglut split-Gal4 drivers that labeled L2 and L1, while Davis et al., Bioarxiv 2018, found a similar result with single cell RNA sequencing. To explain the data, it may be that Tm1/Tm2 express nicotinic ach receptors and Mi1 and Tm3 express gluCl. **However, it is not proven or shown at the level of detail and causality you show here.** This issue comes up again on p. 16 with the discussion of Mi1, Tm1-3. It is interesting to note that the signal inversion appears to happen one synapse later in adult Dmel than in larvae and in mammalian retina. **I think the summary of the data in the adult should be made a bit more specific, which would also serve to better emphasize this paper's contribution.**

Response:

We thank the reviewer for the helpful suggestions. In this revision, we modified the introduction to include the information and citations on molecular studies related to ON and OFF selectivity in the adult *Drosophila* visual system on Page 3.

In the adult Drosophila visual system, both functional and anatomical connectivity have been established for ON and OFF pathways. Functional imaging studies indicate that the ON vs. OFF selectivity emerges in the visual interneurons in the medulla, where the voltage to calcium transformation generates sign-inversion or preservation and leads to ON selectivity in Tm3 and Mi1 and OFF selectivity in Tm1 and Tm2¹⁰⁻¹³. However, despite recent efforts in transcriptome profiling and genetic analyses^{14, 15}, the molecular machinery mediating signal transformation within these pathways has not yet been clearly identified.

F1b: it might be useful to outline the regions of the brain described in 1a.

Response:

We modified Fig. 1b with the larval optic neuropil (LON) region outlined.

p. 11: “In contrast, the light onset induced a small yet noticeable reduction of calcium signal in glu-LOLP”. I don’t see this in Figure 3c, which is referenced for this effect. 3d, yes.

Response:

We thank the reviewer for the raising this valid point. The reduction of the RCaMP signal in this set of experiments is difficult to observe, likely due to the poor signal to noise ratio associated with RCaMP recordings using a confocal laser. To avoid misinterpretation, we amended our statement in the main text on Page 10:

RCaMP recordings showed that, with an extended light exposure, cha-LOLP only responded to the light onset with a fast calcium transient, demonstrating its specific response to light increments. In contrast, glu-LOLP responded to the light offset with an immediate calcium rise, suggesting that glu-LOLP is activated by the light decrements (Fig. 3c).

F3d: make it clear in main text that these ON and OFF pulses are on a background of light (the same background)? That’s what I assumed from stating they were at the same contrast. What was the contrast?

Response:

We included the light measurements and the methods we used for light stimulation in the Methods section. Following the reviewer’s suggestion, we included the measurement of the level of contrast in the main text on Page 10.

p. 19: “predicated by” does not seem to be the right word, I think. Also p. 20.

Response:

Following the reviewer’s suggestion, we modified our statements to “The model suggested that…” on both Page 18 and 19.

p. 24: motion detection vs. direction-selectivity. The authors should take care with terminology here. Do they really want to imply that larvae can detect directional motion? I haven't seen evidence of that, but may have missed it. But this could be unpacked a bit. If it's not DS, then I'm less sure this is a worthy comparison, since 'motion detection' can in just be detecting local luminance changes, which generally occur only when there is motion in the field of view.

Response:

We appreciate the reviewer's comments. Although our recent studies indicate that the Rh6-PR/IOLP pathway is involved in the larval motion detection (Dombrowski, M. et al. 2019), we did not perform functional studies on the VPNs and do not have evidence for larvae detecting directional motion. To avoid overstatement, we removed the discussion related to motion detection.

p. 25: I'm a bit skeptical of calling this G alpha o a "convergence", if I'm interpreting this discussion point correctly. What seems common is using a GPCR and an ionotropic receptor to the same neurotransmitter. These are even used differently for ON and OFF in mammalian retina vs. here, since the PRs in the system respond with opposite polarities. There are only so many G-proteins, right?

Response:

To avoid overstatement and improve accuracy, we amended the text on Page 23:

Although the two visual systems are constructed using different neurochemicals, Gao signaling is responsible for producing sign-inversion in both glu-IOLP and the ON bipolar cell 51. In mGluR6-expressing ON bipolar cells, light increments trigger Gao deactivation, the opening of TrpM1 channels, and depolarization. In larval glu-IOLP, how light induces voltage and calcium responses via mAChR-B signaling has yet to be determined.

Since the authors have proposed signs for most of the interactions in their diagram, it would be helpful to indicate excitatory vs. inhibitory synapses in a final model, including in the interactions between IOLPs.

Response:

We thank the reviewer for the suggestion. The diagram shown in **Fig. 7d** (old Fig. 7a) illustrated our model with the inhibitory and excitatory synaptic connections marked in different colors. We defined the colors in the figure legend: "blue as inhibitory and red as excitatory".

I believe the authors could clear up considerable confusion by referring to the delayed calcium response to light flashes in OFF cells as 'delayed', rather than 'slow'. This effectively contrasts it with 'immediate' in the ON cell. They refer to this delayed calcium signal as an 'ON response' at several points; this characterization should be avoided, given their interpretation of the data elsewhere. An ON response corresponds to a positive response to a positive derivative, while this is a response to a flash of light, which contains positive and negative temporal derivatives (as R2 noted).

The authors should also not make claims about rectification of the signals in the IOLPs based on light flashes alone; light and dark flashes would have to be compared. That claim is not central and could be easily removed.

Response:

We greatly appreciate the reviewer's suggestions. We modified the main text and used '**delayed**' rather than '**slow**' response to describe **the calcium rise** in glu-IOLP. We also replaced the '**ON response**' of glu-IOLP to '**light-induced response**' in the text. Lastly, we **removed the statement of rectification of calcium signals in the IOLPs** (Page 14) following the reviewer's suggestions.

Reviewers' Comments:

Reviewer #1:

Remarks to the Author:

I have no additional comments.

Reviewer #2:

Remarks to the Author:

When I received the latest version of the manuscript and first read the title and abstract, I was at first pleased to see that the authors did no longer make strong claims about temporal dynamics, and instead seemed to focus on the mechanisms of ON and OFF selectivity.

However, I was surprised when I looked at the figures, and read the description of the results, and the rebuttal letter. The authors still fail to recognize and focus on the key results that are visible in their data. Astonishingly, they still keep quantifying the slow increase in calcium (instead of the initial, fast decrease, which is still omitted from most bar plots). This is true for various figures.

The core question and source of disagreement deals with the nature of the slow calcium response and the question of whether this is physiologically meaningful.

In their rebuttal the authors state that the slow calcium response is more robust and "bigger". This unfortunately does not mean that it is meaningful. First, there is no single manipulation where the fast and slow component of the calcium response are manipulated independently from each other. So there is no reason to believe that this isn't just a secondary consequence of the decrease in calcium signal, that is for some reason produced under their imaging / stimulation conditions. Second, the authors themselves basically acknowledge that it is not visible in Arlight data. In the figure that they provided in their rebuttal, they point out a random peak does not obscure the fact that this is just the highest point in a shifted baseline!

And even if the response WAS physiological, it will still at best be a feedback signal. This signal happens several seconds (!!) after the light stimulus and thus cannot be an immediate neuronal response to a photoreceptor signal that is just one or two synapses away. Therefore, it is not helpful in elucidating the synaptic and circuit mechanisms that the authors are aiming to address in their manuscript.

This is also fundamentally different to the temporal dynamics of visual neurons measured by Yang et al (although the manuscript keeps highlighting similarities to adult visual system neurons), which are completed in a few hundred milliseconds. The fact that larval behavior is slow is not a relevant argument in this context, because the authors use their data to make claims about circuit organization and connectivity.

Pointing out long latency responses that have been measured in other systems (e.g. in mGluR6 mutants in the retina) are not a good comparison in this context, because this manuscript here aims towards identifying the synaptic mechanisms that is transforming a fast photoreceptor input.

The paper published in its current version is much below the quality expected for Nature Communication.

Reviewer #3:

Remarks to the Author:

The authors have addressed most of my concerns from the last round. The addition of the gluSNFR data is nice, since it is consistent with a physiological role for the delayed calcium transient they observe in response to light flashes in OFF-IOLPs cells. Whether this sort of light flash would occur naturally remains unclear and is beyond the scope of this study. I think the title change is also welcome, and correctly highlights the most important point of the paper. The changes in immediate vs. delayed responses under some of these manipulations are fascinating but can be difficult to interpret.

I was a little dismayed by the new introductory prose on the adult *Drosophila* ON-OFF split, since it mis-states the primary results in the field (see below). However, a more careful read through the cited papers followed by a revision should be able to correct this prose. Overall, I think the new first Intro paragraph is made more confusing than it needs to be by interleaving results from mouse retina and adult *Drosophila*, in ways that are at times unclear. The authors would be better served by first discussing unifying principles, then what is known about each system in turn.

Line 34: Paragraph 1 hops back and forth a lot between *Drosophila* adults and mammalian retina. It might be better to have one paragraph of general principles (ON OFF segregation, separate lamina), then go into each in turn.

Line 42: "Molecularly..." Makes it sound like this is general, while it in fact refers to vertebrate retina. (And what will be found in this paper.)

Line 50: "Functional imaging..." states incorrectly that voltage-to-calcium transformations lead to sign inversion or not in medulla neurons. This is not true. The sign inversion happens at the L1-Mi1/Tm3 synapse and L2-Tm1/Tm2. L1 and L2 are both off cells. L1 inverts by using glu where Mi1 and Tm3 (likely) express gluCl channels. L2 is likely excitatory with ACh. This is why the voltage responses in Mi1 and Tm1, for instance, are already inverted (Yang et al.). Some degree of *rectification* occurs in the voltage-calcium transformation in these cells. This distinction is critical to state correctly when summarizing this literature, which is adjacent to this study.

Line 290: "fast" calcium rise. Here and elsewhere, I would advocate referring to "immediate" and "delayed" calcium signals. This is still compatible with your interpretations. The results in Figure 5 are really quite striking.

Line 457: typo "produces"

Figure 7c: Fast and slow have a weird underscore between them. Perhaps just write it out?

Point-by-point response to reviewers' comments

NCOMMS-18-25662-D: "Muscarinic acetylcholine receptor signaling generates OFF selectivity in a simple visual circuit" by Qin et al.

We thank the editor and reviewers for their evaluations of our work. Both Reviewer 1 and 3 did not raise additional technical or conceptual issues. In this revision, we made textual changes to address the editorial requests and reviewers' comments. We hope the editor find this revised version now suitable for publication in Nature Communications.

Reviewer #1 (Remarks to the Author):

I have no additional comments.

Reviewer #2 (Remarks to the Author):

When I received the latest version of the manuscript and first read the title and abstract, I was at first pleased to see that the authors did no longer make strong claims about temporal dynamics, and instead seemed to focus on the mechanisms of ON and OFF selectivity. However, I was surprised when I looked at the figures, and read the description of the results, and the rebuttal letter. The authors still fail to recognize and focus on the key results that are visible in their data. Astonishingly, they still keep quantifying the slow increase in calcium (instead of the initial, fast decrease, which is still omitted from most bar plots). This is true for various figures.

Response:

We thank Reviewer 2 for evaluating our revised manuscript. As detailed in our revised manuscript and previous responses, our functional imaging studies revealed light-evoked biphasic physiological responses in glu-IOLP. We quantified both components of this response for the initial set of experiments but focused on the calcium increases in our genetic studies. To specifically address the reviewer's concern, we added the following statement in the Discussion section to clarify this point (P10):

“our optical recording approaches have certain technical limitations, including the kinetics and sensitivities of the voltage and calcium sensors as well as our imaging and visual stimulation protocols. In addition, although glu-IOLP displays a biphasic response towards the light stimulation, we quantified calcium reductions and increases for only the initial set of physiological characterizations (Supplementary Figure 4). Compared to the delayed calcium rise, the light-induced calcium reductions have low amplitudes and high variabilities, possibly due to the half-wave rectification of the intra-cellular calcium previously described in adult visual interneurons^{13,29}. For the genetic experiments, we then focused on evaluating the activation of glu-IOLP, which is reflected by the increase of intracellular calcium signals that lead to neurotransmitter release.”

The core question and source of disagreement deals with the nature of the slow calcium response and the question of whether this is physiologically meaningful.

In their rebuttal the authors state that the slow calcium response is more robust and "bigger". This unfortunately does not mean that it is meaningful. First, there is no single manipulation where the fast and slow component of the calcium response are manipulated independently from each other. So there is no reason to believe that this isn't just a secondary consequence of the decrease in calcium signal, that is for some reason produced under their imaging / stimulation conditions. Second, the authors themselves basically acknowledge that it is not visible in Arclight data. In the figure that they provided in their rebuttal, they point out a random peak does not obscure the fact that this is just the highest point in a shifted baseline!

Response:

We disagree with the reviewer on this assessment of our work. We quantify the delayed calcium rise not just because its robustness and consistency. As illustrated in our data presented throughout the manuscript, light-induced biphasic responses in glu-IOLP are clearly demonstrated by four different types of indicators, GCaMP, RCaMP, Arclight and iGluSnFR, in over one hundred independent recordings. Our data strongly support the conclusion that the delayed calcium response is physiological and functionally important. If we were to focus only on the initial reduction of the calcium signal, the features of light-induced responses in glu-IOLP and downstream VPNs would not be faithfully represented. We have clarified this point further in this revision and included additional statements in the Discussion on P10. Please see the response above.

And even if the response WAS physiological, it will still at best be a feedback signal. This signal happens several seconds (!) after the light stimulus and thus cannot be an immediate neuronal response to a photoreceptor signal that is just one or two synapses away. Therefore, it is not helpful in elucidating the synaptic and circuit mechanisms that the authors are aiming to address in their manuscript.

This is also fundamentally different to the temporal dynamics of visual neurons measured by Yang et al (although the manuscript keeps highlighting similarities to adult visual system neurons), which are completed in a few hundred milliseconds. The fact that larval behavior is slow is not a relevant argument in this context, because the authors use their data to make claims about circuit organization and connectivity.

Pointing out long latency responses that have been measured in other systems (e.g. in mGluR6 mutants in the retina) are not a good comparison in this context, because this manuscript here aims towards identifying the synaptic mechanisms that is transforming a fast photoreceptor input.

The paper published in its current version is much below the quality expected for Nature Communication.

Response:

Our results shown in Figure 6 and 7 indicate that the delayed increase in glu-IOLP's intracellular calcium is functionally important for glutamate release and signal transduction within the visual circuit. As mentioned in our previous response, although we acknowledge that the temporal scale of this ON response is surprisingly slow, light-induced slow physiological responses, in the range of hundreds of milliseconds to seconds, have been observed in visual interneurons in the

adult *Drosophila* visual system and mammalian retinae. Given their similar functional roles with glu-IOLP, we believe those comparisons are appropriate and support our conclusions.

Reviewer #3 (Remarks to the Author):

The authors have addressed most of my concerns from the last round. The addition of the gluSNFR data is nice, since it is consistent with a physiological role for the delayed calcium transient they observe in response to light flashes in OFF-IOLPs cells. Whether this sort of light flash would occur naturally remains unclear and is beyond the scope this study. I think the title change is also welcome, and correctly highlights the most important point of the paper. The changes in immediate vs. delayed responses under some of these manipulations are fascinating but can be difficult to interpret.

I was a little dismayed by the new introductory prose on the adult *Drosophila* ON-OFF split, since it mis-states the primary results in the field (see below). However, a more careful read through the cited papers followed by a revision should be able to correct this prose. Overall, I think the new first Intro paragraph is made more confusing that it needs to be by interleaving results from mouse retina and adult *Drosophila*, in ways that are at times unclear. The authors would be better served by first discussing unifying principles, then what is known about each system in turn.

Response:

We thank the reviewer for evaluating our revised manuscript and the constructive comments. We modified the introduction following the reviewer's suggestions (P3).

Line 34: Paragraph 1 hops back and forth a lot between *Drosophila* adults and mammalian retina. It might be better to have one paragraph of general principles (ON OFF segregation, separate lamina), then go into each in turn.

Line 42: "Molecularly..." Makes it sound like this is general, while it in fact refers to vertebrate retina. (And what will be found in this paper.)

Line 50: "Functional imaging..." states incorrectly that voltage-to-calcium transformations lead to sign inversion or not in medulla neurons. This is not true. The sign inversion happens at the L1-Mi1/Tm3 synapse and L2-Tm1/Tm2. L1 and L2 are both off cells. L1 inverts by using glu where Mi1 and Tm3 (likely) express gluCl channels. L2 is likely excitatory with ACh. This is why the voltage responses in Mi1 and Tm1, for instance, are already inverted (Yang et al.). Some degree of *rectification* occurs in the voltage-calcium transformation in these cells. This distinction is critical to state correctly when summarizing this literature, which is adjacent to this study.

Response:

We thank the reviewer for these specific suggestions and modified the introduction accordingly.

Line 290: “fast” calcium rise. Here and elsewhere, I would advocate referring to “immediate” and “delayed” calcium signals. This is still compatible with your interpretations. The results in Figure 5 are really quite striking.

Line 457: typo “produces”

Figure 7c: Fast and slow have a weird underscore between them. Perhaps just write it out?

Response:

We modified the text and figure following the reviewer’s suggestions.